**US surface ozone trends and extremes from 1980-2014: Quantifying the roles of rising Asian emissions, domestic controls, wildfires, and climate**

Meiyun Lin[1,2*], Larry W. Horowitz[2], Richard Payton[3], Arlene M. Fiore[4], Gail Tonnesen[3]

[1]Atmospheric and Oceanic Sciences, Princeton University, Princeton, NJ 08540, USA
[2]NOAA Geophysical Fluid Dynamics Laboratory, Princeton, NJ 08540, USA
[3]U.S. Environmental Protection Agency, Region 8, Denver, CO 80202, USA
[4]Lamont-Doherty Earth-Observatory and Department of Earth and Environmental Sciences, Columbia University, Palisades, NY 10964, USA

**\*Corresponding Author** (Meiyun.Lin@noaa.gov; Phone: 1-609 452-6551)

**Abstract.** US surface $O_3$ responds to varying global-to-regional precursor emissions, climate, and extreme weather, with implications for designing effective air quality control policies. We examine these conjoined processes with observations and global chemistry-climate model (GFDL-AM3) hindcasts over 1980-2014. The model captures the salient features of observed trends in daily maximum 8-hour average $O_3$: (1) increases over East Asia (up to 2 ppb yr$^{-1}$), (2) springtime increases at western US (WUS) rural sites (0.2-0.5 ppb yr$^{-1}$) with a baseline sampling approach, (3) summertime decreases, largest at the 95$^{th}$ percentile, and wintertime increases in the 50$^{th}$ to 5$^{th}$ percentiles over the eastern US (EUS). Asian $NO_x$ emissions tripled since 1990, contributing as much as 65% to modeled springtime background $O_3$ increases (0.3-0.5 ppb yr$^{-1}$) over the WUS, outpacing $O_3$ decreases attained via 50% US $NO_x$ emission controls. Methane increases over this period contribute only 15% of the WUS background $O_3$ increase. Springtime $O_3$ observed in Denver has increased at a rate similar to remote rural sites. During summer, increasing Asian emissions approximately offset the benefits of US emission reductions, leading to weak or insignificant observed $O_3$ trends at WUS rural sites. Mean springtime WUS $O_3$ is projected to increase by ~10 ppb from 2010 to 2030 under the RCP8.5 global change scenario. While historical wildfire emissions can enhance summertime monthly mean $O_3$ at individual sites by 2-8 ppb, high temperatures and the associated buildup of $O_3$ produced from regional anthropogenic emissions contribute most to elevating observed summertime $O_3$ throughout the USA. GFDL-AM3 captures the observed interannual variability of summertime EUS $O_3$. However, $O_3$ deposition sink to vegetation must be reduced by 35% for the model to accurately simulate observed high-$O_3$ anomalies during the severe drought of 1988. Regional $NO_x$ reductions alleviated the $O_3$ buildup during the recent heat waves of 2011 and 2012 relative to earlier heat waves (e.g., 1988; 1999). The $O_3$ decreases driven by $NO_x$ controls were more pronounced in the Southeast US, where the seasonal onset of biogenic isoprene emissions and $NO_x$-sensitive $O_3$ production occurs earlier than in the Northeast. Without emission controls, the 95$^{th}$ percentile summertime $O_3$ in the EUS would have increased by 0.2-0.4 ppb yr$^{-1}$ over 1988-2014 due to more frequent hot extremes and rising biogenic isoprene emissions.

## 1. Introduction

Within the United States, ground-level $O_3$ has been recognized since the 1940s and 1950s as an air pollutant detrimental to public health. Decreases in summertime $O_3$ were observed in parts of California and throughout the EUS (e.g., Cooper et al., 2012; Simon et al., 2015), following regional $NO_x$ controls after the lowering of the US National Ambient Air Quality Standard (NAAQS) for $O_3$ in 1997 to 84 ppb. On the basis of health evidence, the NAAQS level for $O_3$ has been further lowered to 75 ppb in 2008 and to 70 ppb in 2015 (Federal Register, 2015). There are concerns that rising Asian emissions and global methane (Jacob et al., 1999; Lin et al., 2015b), more frequent large wildfires in summer (e.g., Jaffe, 2011; Yang et al., 2015; Abatzoglou et al., 2016), and late spring deep stratospheric $O_3$ intrusions (Lin et al., 2012a; Langford et al., 2014; Lin et al., 2015a) may pose challenges in attaining more stringent $O_3$ standards at high-elevation WUS regions. A warming climate would also offset some of the air quality improvements gained from regional emission controls (e.g., Fiore et al., 2015). Quantitative understanding on sources of $O_3$ variability on daily to multi-decadal time scales can provide valuable information to air quality control managers as they develop $O_3$ abatement strategies under the NAAQS. Here we systemically investigate the response of US surface $O_3$ means and extremes to changes in Asian and North American anthropogenic emissions, global methane, regional heat waves and wildfires over the course of 35 years from 1980 to 2014, using observations and chemistry-climate model (GFDL-AM3) hindcasts (Lin et al., 2014; 2015a; 2015b).

Rapid economic growth has led to a tripling of $O_3$ precursor emissions from Asia in the past 25 years (e.g., Granier et al., 2011; Hillboll et al., 2013). Observed 1-hour $O_3$ mixing ratios can frequently reach 200-400 ppb during regional pollution episodes in eastern China (Wang T. et al., 2006; Li et al., 2016), with a seasonal peak in the late spring to early summer (Wang Y. et al., 2008; Lin et al., 2009). A synthesis of available observations from the mid-1990s to the 2000s indicates increases of 1-2 ppb $yr^{-1}$ in spring to summer $O_3$ in China (Ding et al., 2008; Ma et al., 2015; Sun et al., 2015). Long-range transport of Asian pollution plumes towards western North America has been identified by aircraft and satellite measurements and in chemical transport models (e.g., Jaffe et al., 1999; Fiore et al., 2009; Brown-Steiner and Hess, 2011; Lin et al., 2012b; Huang et al., 2013; Verstraeten et al., 2015). Systematic comparison of observed and modeled long-term $O_3$ trends over Asia is lacking in the published literature but is needed to establish confidence in models used to assess the global impacts of rising Asian emissions.

Model simulations indicate that import of Asian pollution enhances mean WUS surface $O_3$ in spring by ~5 ppb (Zhang et al., 2008; Lin et al., 2012b), and occasionally contributes 8-15 ppb during springtime pollution episodes observed at rural sites (Lin et al., 2012b) as supported by in situ aerosol composition analysis (VanCuren and Gustin 2015). Stratospheric intrusions can episodically increase daily 8-hour average surface $O_3$ by 20-40 ppb, contributing to the highest observed $O_3$ events at high-elevation WUS sites (Lin et al., 2012a; Lin et al., 2015), in addition to pollution transport from California (e.g., Langford et al., 2010). In the densely populated EUS, both changes in regional anthropogenic emissions and air pollution meteorology have the greatest impacts on summer surface $O_3$ during pollution episodes (e.g., Jacob and Winner 2009;

Rieder et al., 2015; Porter et al., 2015; Pusede et al., 2015). Discerning directly the effect of climate change on air quality from long-term observation records of $O_3$ would be ideal, but concurrent trends in precursor emissions and large internal variability in regional climate impede such an effort. It is difficult to separate the impacts of changes in global-to-regional precursor emissions and different meteorological factors on $O_3$ at given locations without the benefit of multiple sensitivity experiments afforded by models.

On the other hand, process-oriented assessments of the models are needed to build confidence in their utility for assessing pollution control strategies, estimating tropospheric $O_3$ radiative forcing and projecting pollution extremes under future climate scenarios (e.g., Monks et al., 2015). A number of studies show that global models capture observed decreases in summertime $O_3$ over the EUS during 1990-2010, but have difficulty simulating $O_3$ increases measured at remote high-elevation sites that are believed to represent hemispheric-scale conditions with little influence from fresh local pollution (hereafter referred to as "baseline") (e.g., Lamarque et al., 2010; Koumoutsaris and Bey, 2012; Parrish et al., 2014; Brown-Steiner et al., 2015; Strode et al., 2015). Recently, Lin et al. (2015b) examined the representativeness of $O_3$ trends derived from sparse measurements in the free troposphere over the WUS, originally reported by Cooper et al. (2010) and used in prior model evaluations. They found that discrepancies between observed and simulated $O_3$ trends reflect measurement sampling biases. Here we seek additional insights into the causes of the model-observation disagreement at the WUS rural sites with continuous, high-frequency measurements. Notably, we reconcile observed and simulated $O_3$ trends at these sites with a baseline sampling approach in the model.

Our goal in this paper is twofold: first, to systematically evaluate how well the GFDL-AM3 model represents trends and variability of surface $O_3$ observed at rural sites across the US; second, to examine changes in US surface $O_3$ means and extremes in a suite of multi-decadal hindcast simulations designed to isolate the response of $O_3$ to increases in Asian anthropogenic emissions, North American emission controls, rising global methane, wildfires, and interannual variability in meteorology. We examine trends across the entire probability distribution of $O_3$ concentration, which is crucial to assessing the ability of models to simulate the surface $O_3$ response under different temperature and chemical regimes depending on seasons, geographical location, and regional transport patterns. Specifically, we evaluate the trends separately for the 5[th], 50[th] and 95[th] percentiles of the $O_3$ concentration distribution in spring (MAM), summer (JJA), autumn (SON), and winter (DJF).

Section 2 briefly describes the observational records, model experiments, and analysis approach. As a first step towards assessing our understanding of the impacts of rising Asian emissions, we briefly review Asian $O_3$ trends from observations in recent publications and evaluate modeled trends (Sect. 3). We then focus our analysis on the US, using both observations and models to assess the response of US surface $O_3$ to changes in background $O_3$, regional anthropogenic emissions and meteorology (Sect. 4). In Section 5, we further separate the influence of background on WUS $O_3$ into components driven by rising Asian anthropogenic emissions, global methane, and wildfires. We quantify the contribution of these factors to surface $O_3$ in both rural areas such as national parks (Sect. 5.1 to 5.3) and in densely populated regions

such as the Denver Metropolitan area (Sect. 5.4). After evaluating historical trends, we
additionally draw upon two simulations following the 21st century RCP4.5 versus RCP8.5 global
change scenarios to project WUS $O_3$ through 2050 (Sect. 5.2). Section 6 examines how the EUS
summertime $O_3$ probability distribution and pollution extremes respond to large-scale heat waves,
droughts, and regional $NO_x$ reductions over the past decade, and how well our model simulates
the observed features. Finally, we summarize in Section 7 the key drivers of US surface $O_3$
trends and extremes and discuss the implications of this study.
**2. Model and Observations**
**2.1 Chemistry-Climate Model Experiments.**
**(Table 1 about here: Model Experiments)**
The GFDL-AM3 model includes interactive stratosphere-troposphere chemistry and
aerosols on a cubed sphere grid with a resolution of approximately 200x200 $km^2$ (Donner et al.,
2011). **Table 1** summarizes meteorology, radiative forcing agents, and emissions used in each
experiment. The hindcast simulations (1979-2014) are nudged to the NCEP/NCAR reanalysis
zonal and meridional winds using a height-dependent nudging technique (Lin et al., 2012b).
Biogenic isoprene emissions and lightning $NO_x$ are tied to model meteorology (Guenther et al.,
2006; Rasmussen et al., 2012) and thus can respond to changes in climate, whereas soil $NO_x$ and
chemical dry deposition velocities are set to a monthly climatology (Naik et al., 2013), with a
diurnal cycle applied for $O_3$ dry deposition. To investigate the possible influence of drought on
$O_3$ removal (e.g., Emberson et al., 2013), we additionally conduct a sensitivity simulation for
1988 with reduced $O_3$ deposition velocity (see Sect.6). Our **BASE** simulation and two additional
simulations with modified emissions (**FIXEMIS** and **IAVFIRE**) were previously used to
interpret the causes of increasing autumnal $O_3$ measured at Mauna Loa Observatory in Hawaii
since 1974 (Lin et al., 2014), interannual variability of springtime $O_3$ (Lin et al., 2015a) and the
representativeness of free tropospheric $O_3$ measurements over the WUS (Lin et al., 2015b).
With anthropogenic emissions and methane held constant (**Table 1**), the **FIXEMIS** and
**IAVFIRE** simulations isolate the influence from meteorology and wildfire emissions,
respectively. In **IAVASIA**, anthropogenic emissions from East Asia (15ºN-50ºN, 95ºE-160ºE) and
South Asia (5ºN-35ºN, 50ºE-95ºE) are allowed to vary from year to year as in **BASE**, while
anthropogenic emissions in the other regions of the world, global methane and wildfire emissions
are held constant as in **FIXEMIS**. In **IAVCH₄**, global methane is allowed to vary over time as in
**BASE**, but with anthropogenic and wildfire emissions held constant as in **FIXEMIS**. The
**IAVASIA** and **IAVCH₄** simulations thus isolate the role of rising Asian anthropogenic emissions
and global methane, respectively, by contrasting with the **FIXEMIS** simulation. Both **BASE** and
**IAVCH₄** simulations apply observed time-varying methane concentrations as a lower boundary
condition for chemistry **(Fig.S1)**. Thus, underestimates in historical methane emissions reported
recently by Schwietzke et al. (2016) do not affect our results. We quantify the total contributions
to surface $O_3$ from meteorological variability, stratosphere-to-troposphere transport, pollution
from foreign continents and $O_3$ produced by global methane, lightning $NO_x$, wildfires and
biogenic emissions with the **Background** simulation, in which North American anthropogenic

emissions are zeroed out relative to **BASE**. We additionally draw upon two simulations with the GFDL Coupled Model CM3 following the 21$^{st}$ century RCP global change scenarios to project changes in WUS $O_3$ through 2050. Details of these CM3 simulations were described in John et al. (2012).

**2.2 Anthropogenic and Biomass Burning Emissions**

**(Figure 1 about here: Changes in $NO_x$ emissions)**

We first examine how well the emission inventories in AM3 BASE represent changes in regional $NO_x$ emissions over recent decades inferred from satellite measurements of tropospheric vertical column density ($VCD_{trop}$) of $NO_2$. The combined record of GOME and SCIAMACHY shows that $VCD_{trop}$ $NO_2$ over the highly polluted region of eastern China almost tripled during 1996-2011 (**Fig.1a**). In contrast, $VCD_{trop}$ $NO_2$ over the EUS decreased by ~50% in the 2000s (**Fig.1b**) due to $NO_x$ State Implementation Plans (commonly known as the $NO_x$ SIP Call) and many rules that tighten emission standards for mobile sources (McDonald et al., 2012). Similar decreases occurred in WUS cities, resulting from the $NO_x$ control programs to achieve $O_3$ and regional haze planning goals. These trends are consistent with those reported by a few recent studies (e.g., Hilboll et al., 2013), including those using OMI $NO_2$ data (Russell et al., 2012; Duncan et al., 2016). For comparison with satellite data, we sample the model archived every three hours closest to the time of satellite overpass for the SCIAMACHY and GOME products we use in Figure 1 (10:00-10:30 am local time). Trends in $VCD_{trop}$ $NO_2$ are similar to those in $NO_x$ emissions (orange lines versus red triangles in **Fig.1a-1b**), indicating that any changes in $NO_x$ chemical lifetime or partitioning have negligible influence in our model, consistent with $NO_2$ loss against OH being minor during the morning overpasses of GOME and SCIAMACHY. The emission inventory used in BASE, from Lamarque et al. (2010) with annual interpolation after 2000 to RCP8.5 (Lamarque et al., 2012), mimics the opposing changes in $NO_x$ emissions over eastern China versus the EUS during 1996-2011, consistent with changes in $VCD_{trop}$ $NO_2$ retrieved from the satellite instruments. For comparison, the RCP4.5 interpolation for 2001-2010 in CMIP5 historical simulations analyzed by Parrish et al. (2014) underestimates the increase in Chinese $NO_x$ emissions by a factor of two (**Fig.1a**). Recent reductions in Chinese $NO_x$ emissions after 2011 (Duncan et al., 2016) are not represented in the inventories used in AM3.

Our BASE model applies interannually-varying monthly mean emissions from biomass burning based on the RETRO inventory (Schultz et al., 2008) for 1970 to 1996 and GFEDv3 (van der Werf et al., 2010) for 1997 onwards, distributed vertically as recommend by Dentener et al. (2006). **Fig. S2** illustrates the interannual variability of biomass burning CO emissions from the main source regions of the Northern Hemisphere over the period 1980-2014. Boreal fire emissions in Eurasia almost doubled from 1980-1995 to 1996-2014, with large fires occurring more frequently in the recent decade, as found for the WUS (Dennison et al., 2014; Yang et al., 2015).

**2.3 Ozone Observation Records and Uncertainties**

Long-term surface $O_3$ observation records were obtained at 70 selected rural monitoring sites with 20 (1995-2014) to 27 (1988-2014) years of continuous hourly measurements from the

US National Park Services, the US Clean Air Status and Trends Network (CASTNet), and the US EPA Air Quality System. Cooper et al. (2012) reported trends in daytime (11am-4pm) $O_3$ over 1990-2010 at 53 rural sites. We investigate trends in daily maximum 8-hour averaged (MDA8) $O_3$ and expand the analysis of Cooper et al. using additional data to 2014 and including 17 additional sites with measurements begun in 1991-1995. All sites have at least 20 years of data. If a site has less than 50% data availability in any season then that particular season is discarded. The trend is calculated separately for the 5[th], 50[th] and 95[th] percentiles of daily MDA8 $O_3$ for each season through ordinary linear least-square regression. Statistics are derived for the slope of the linear regression in units of ppb yr[-1], the range of the slope with a 95% confidence limit (not adjusted for sample autocorrelation), and the p-value indicating the statistical significance of the trend based on a two-tailed t-test.

**(Figure 2 about here: Measurement uncertainties)**

A cross-site consistency analysis was performed to determine robust changes in the time evolution of $O_3$ over the WUS during 1988-2014 (**Fig.2**). The monitor at Yellowstone National Park was moved 1.5 km from the Lake Yellowstone site to the Water Tank site in 1996. While the local transport patterns are slightly different for the two sites, using MDA8 data from the well-mixed midday period minimizes the differences (Jaffe and Ray, 2007). Observed $O_3$ interannual variations show large-scale similarity across sites over the Intermountain West except for the earlier period 1989-1990. During this period, observations at Yellowstone and Rocky Mountain National Parks show low-$O_3$ anomalies that do not appear at other sites but there is no change in measurement technique. Jaffe and Ray (2007) suggest this represents large-scale variations in background $O_3$ that are seen in common at these two parks. However, analysis of meteorological fields and model diagnostics does not reveal any obvious transport anomaly influencing $O_3$ variations at these sites in 1990 (Lin et al., 2015a). Observations at Pinedale in January-February 1990 are also anomalously low relative to Grand Canyon (GRC474), Centennial (CNT169), and Gothic (GTH161). These anomalous data at the beginning of measurement records can substantially influence trends calculated from short records. For example, Cooper et al. (2012) found a summer $O_3$ increase of 0.42±0.30 ppb yr[-1] at Yellowstone over 1990-2010. Removing 1990, we find a weaker increase of 0.28±0.27 ppb yr[-1] (**Fig.2b**). Removing 1990 at Rocky Mountain resulted in a weaker springtime $O_3$ increase of 0.29±0.17 ppb yr[-1] compared to 0.43±0.23 ppb yr[-1] over 1990-2010 (**Fig.2c**). To assess robust $O_3$ changes, we thus remove these apparently uncertain measurements in 1990 from the subsequent analysis.

**2.4 Model Baseline Sampling Approach**

**(Figure 3 about here: Influence of baseline sampling)**

Springtime $O_3$ observations at WUS high-elevation sites ($\geq$ 1.5 km a.s.l.) typically represent baseline conditions with little influence from fresh local pollution. In a global model with ~200x200 km$^2$ horizontal resolution, however, these remote sites can reside in the same grid cell that contains urban cities where $NO_x$ emissions decreased over the analysis period. For example, Rocky Mountain National Park (2.7 km a.s.l.) is less than 100 km from the Denver Metropolitan area in Colorado. This limitation of large-scale models in resolving urban-to-rural

gradients and sharp topography results in an artificial offset of increased baseline $O_3$ at remote sites by decreased urban pollution within the same model grid cell. Thus, coarse-resolution models are often unable to reproduce observed $O_3$ increases at the high-elevation sites representative of remote baseline conditions (**Figs. 3a vs. 3b**), as found in many prior modeling analyses (e.g., Parrish et al., 2014; Strode et al., 2015 and references therein). This limitation can be addressed by using a baseline selection procedure to identify conditions for sampling the model to avoid model artifacts caused by poor spatial resolution, as described below.

All measurements presented in this study are unfiltered. We implement a set of regional CO-like tracers (COt), with a 50-day exponential decay lifetime and surface emissions constant in time from each of four northern mid-latitude source regions (Lin et al., 2014). We use these COt tracers to bin modeled $O_3$ according to the dominant influence of different continental air regimes. To represent observed baseline conditions at WUS sites, we sample AM3 at 700 hPa (~3 km a.s.l.) and filter the $O_3$ data in the BASE simulation to remove the influence from fresh local pollution. Specifically, our filter excludes days when North American COt (NACOt) exceeds the 67[th] percentile for each season. This procedure yields higher calculated baseline $O_3$ increases (**Fig.3c**), bringing it closer to observations (**Fig.3a**). When sampled at 700 hPa without filtering (**Fig.3d**), BASE gives statistically significant $O_3$ increases but the rate of increase is ~0.1 ppb yr$^{-1}$ weaker than with filtering. With North American anthropogenic emissions shut off, the model simulates significant $O_3$ increases that are similar at the surface (**Fig.3e**) and at 700 hPa (**Fig.3f**). This finding indicates that the underestimate of $O_3$ increases in BASE, when sampled at the surface (**Fig.3b**), reflects an excessive offset from domestic pollution decreases in the model relative to observed conditions, as opposed to the insufficient mixing of free tropospheric $O_3$ to the surface. As individual sites display observed trends falling in between the filtered model, and those sampled at the surface versus aloft, we can use the model to interpret which sites are most frequently sampling baseline versus influenced by North American anthropogenic emissions. For consistency, in the subsequent analysis we apply model baseline filtering to all WUS sites with elevations greater than 1.5 km altitude. In the EUS, where the terrain and monitor elevations are much lower than in the west and observed $O_3$ trends are largely controlled by regional emission changes, we always sample the model at the surface without filtering.

## 3. Global Distribution of Lower Tropospheric $O_3$ Trends
### 3.1 Global $O_3$ Burden and Distribution of Trends

**(Figure 4 about here: Global distribution)**

We begin by examining the global distribution of lower tropospheric $O_3$ trends over 1988-2014 from the BASE simulation (**Fig.4**) and focus on the differences between the surface and free troposphere (~700 hPa), with implications for understanding the impact of trends in hemispheric baseline $O_3$ on surface air quality. The model indicates that surface MDA8 $O_3$ levels in Asia have increased significantly by 1.5-2.5 ppb yr$^{-1}$ in the 95[th] percentile (**Fig.4a-b**) and by 1-2 ppb yr$^{-1}$ in the median values (**Fig.4c-d**), with the largest increases occurring in South Asia during spring and over Eastern China during summer. In contrast, there is a marked decrease in

surface MDA8 $O_3$ in WUS cities, throughout the EUS and in central Europe, particularly at the high percentiles and during summer. The increase in surface $O_3$ over Asia and decreases over the US and Europe are consistent with changes in regional emissions of $O_3$ precursors over this period (**Fig.1**).

Over Southeast Asia (south of 30ºN) during spring, earlier springtime $O_3$ photochemical production at lower latitudes coupled with active frontal transport (Liu et al., 2002; Carmichael et al., 2003; Lin et al., 2010) leads to a comparable or even greater increase of $O_3$ in the free troposphere than at the surface **(Figs. 4c vs. 4e).** In contrast, over Central East China during summer the simulated trends of $O_3$ in the free troposphere are at least a factor of three weaker than in surface air **(Figs.4d vs. 4f)**, consistent with the analysis of MOZAIC aircraft data over Beijing in 1995-1999 versus 2003-2005 (Ding et al., 2008). Mean $O_3$ at 700 hPa above parts of North America and Europe show little change in summer or even increase during spring in the model, similar to the trends at 500 hPa (**Fig.S3**), despite the significant decreases in surface air. The global tropospheric $O_3$ burden in the BASE simulation increases by approximately 30 Tg over the past 35 years (**Fig.5a**), attributed mainly to changes in anthropogenic emissions. Over the 2004-2015 OMI/MLS satellite era, however, meteorological variability contributes approximately half to the total simulated decadal trends of $O_3$ burden (**Fig.5a**), indicating that attribution of the satellite-derived decadal trends of global tropospheric $O_3$ burden requires consideration of internal climate variability.

**3.2 Comparison of observed and simulated $O_3$ trends in Asia**

**(Figures 5 and 6 about here)**

Long-term $O_3$ observations are very sparse in Asia, making it difficult to evaluate modeled $O_3$ trends. We compile available measurements from the published literature; including ozonesonde profiles at Hong Kong (2000-2014; http:/woudc.org) and Hanoi (2005-2015; SHADOZ, Thompson et al., 2007), MOZAIC aircraft profiles collected on summer afternoons in the boundary layer (below 1250 m altitude) over Beijing for 1995-2005 (Ding et al., 2008), ground-based measurements at Mt. Tai (1.5 km a.s.l.) in Central Eastern China for July-August 2003-2015 (Sun et al., 2016), at the GAW stations - Shangdianzi north of Beijing for 2004-2014 (Ma et al., 2016) and Mt. Waliguan (3.8 km a.s.l.) in the Tibetan Plateau for 1994-2013 (Xu et al., 2016), at Taiwan for 1994-2007 (Y-K Lin et al., 2010), South Korea for 1990-2010 (Lee et al., 2014), Mt. Happo (1.9 km a.s.l.) in Japan for 1991-2011 (Tanimoto, 2009; Parrish et al., 2014), and a coastal site at Hong Kong in Southern China for 1994-2007 (T Wang et al., 2009).

Recently, Zhang et al. (2016) compiled sparse $O_3$ profiles above Southeast Asia from IAGOS commercial aircraft and ozonesondes from Hanoi for 1994-2004 versus 2005-2014 and found a total springtime $O_3$ increase of 20-25 ppb between the two periods (~2 ppb yr$^{-1}$). However, our model indicates an increase of up to 1 ppb yr$^{-1}$ for springtime free tropospheric $O_3$ over Southeast Asia (Fig.4e). We illustrate the possible influence of sampling deficiencies on the $O_3$ trends inferred from sparse observations (**Fig.5**). The ozonesonde frequency is 4 profiles per month at Hong Kong and only 1-2 profiles per month at Hanoi. To determine the representativeness of $O_3$ trends derived from these sparse measurements, we compare

observations and model results co-sampled on sonde launch days with the 'true average' determined from $O_3$ fields archived every three hours from the model, as in our prior work for WUS sites (Lin et al., 2015a; 2015b). **Figures 5b** and **5c** show the comparisons for the annual trends of $O_3$ over 900-600 hPa. The trends are generally consistent across the sonde data, model co-sampled and 'true average' results for Hong Kong, with an increase of 0.5±0.1 ppb yr$^{-1}$ over 2000-2014. Observations at Hanoi show an apparently rapid $O_3$ increase of 1.1±0.2 ppb yr$^{-1}$ over 2005-2014. AM3 BASE sampled sparsely as in the ozonesondes captures the observed variability ($r^2 = 0.7$), whereas the 'true average' over this period indicates the trend (0.7±0.1 ppb yr$^{-1}$) is only 63% of that inferred from observations. Moreover, interannual variability of $O_3$ resulting from wildfire emissions and meteorology in IAVFIRE is as large as the total $O_3$ change in BASE over the short period 2005-2014. We conclude that measurement sampling artifacts influence the $O_3$ trends reported by Zhang et al. (2016).

Expanding the comparison to a suite of sites across East Asia (**Fig. 6),** we find that AM3 captures the key features of observed $O_3$ trends in Asia, including their seasonal to regional variations, summertime increases (1-2 ppb yr$^{-1}$) in Central Eastern China where $NO_x$ emissions have approximately tripled since 1990 (**Fig.1a**), and springtime increases (0.5 ppb yr$^{-1}$) at Taiwan and Mt. Happo that are driven by pollution outflow from the Asian continent. Note that to place the trends derived from the short observational records into a broader context we show the 20-year trends over 1995-2014 from the model, except for South Korea (1990-2010) and Happo Japan (1991-2011). We match the time period in the model with observations at these two sites because AM3 shows weaker $O_3$ increases when data for the recent years are included, which likely reflects the offsetting effects of regional emission reductions in South Korea and Japan.

Parrish et al. (2014) show that three CMIP5-like models underestimate the observed springtime $O_3$ increase at Mt. Happo by a factor of four. This discrepancy may reflect a combination of factors: (1) underestimates of Asian emission growth in the RCP4.5 interpolation after 2000 used in CMIP5 historical simulations (**Fig.1a**), (2) trends driven by interannual meteorological variability that free-running CMIP5 models are not expected to reproduce exactly, (3) an excessive offset from Japanese pollution decreases in the models owing to their coarse resolution and limitation in resolving observed baseline conditions at Mt. Happo. Sampling our BASE model at 700 hPa above Happo, we find an $O_3$ increase of 0.35±0.13 ppb yr$^{-1}$. When focusing on days strongly influenced by outflow from the East Asian continent (Chinese COt $\geq$ 67$^{th}$), the model $O_3$ trend increases to 0.48±0.13 ppb yr$^{-1}$, approximating the observed increase of 0.76±0.35 ppb yr$^{-1}$ at Mt. Happo (**Fig.6b**). The observed and simulated trends are not statistically different given the overlapping confidence limits. The larger confidence limit (uncertainty) derived from the Happo observations reflects the measurement inconsistency before 1998 and instrumental problems after 2007 (Tanimoto et al., 2016). We conclude that GFDL-AM3 captures 65-90% of the observed $O_3$ increases in Asia, lending confidence in its application to assess the global impacts of rising Asian emissions.

**4. Regional and Seasonal Variability of US Surface $O_3$ Trends**

We next focus our analysis on the US where dense, high-frequency, long-term, reliable

measurements of surface $O_3$ facilitate process-oriented model evaluation. Comparisons of surface $O_3$ trends over 1988-2014 at 70 rural monitoring sites across the US as observed and simulated in AM3 BASE are shown in **Figure 7** for spring, **Figure 8** for summer, **Figure 9** for winter, and in Supplementary **Fig.S4** for autumn. The trends are calculated separately for the 5th, 50th and 95th percentiles of the daily MDA8 $O_3$ concentration distribution, with larger circles on the maps indicating sites with statistically significant trends (p<0.05). We first discuss observations (Sect. 4.1), followed by model evaluation and trend attribution (Sect. 4.2).

## 4.1 Observations

**(Figure 7 about here)**

In spring (**Figure 7**), observations indicate spatial heterogeneity in $O_3$ trends across the Intermountain West, Northeast (north of 38°N), and Southeast US. At the 95th percentile (**Fig.7a**) the pattern of observed trends is homogeneous across the Northeast and Southeast US, with approximately 85% of the sites having statistically significant $O_3$ decreases of 0.4-0.8 ppb yr$^{-1}$ and no sites showing a significant increase. In contrast, significant increases occur at 25% of the sites in the Intermountain West. Only Joshua Tree National Park located downwind of the Los Angeles Basin shows a significant decrease at the 95th percentile. At the 50th percentile (**Fig.7b**) there are significant $O_3$ decreases of 0.2-0.4 ppb yr$^{-1}$ in the Southeast and little overall change in the Northeast, while significant increases of 0.2-0.4 ppb yr$^{-1}$ occur at 50% of the sites in the Intermountain West. Significant springtime $O_3$ increases occur at all observed percentiles at Lassen Volcanic National Park in California, Great Basin National Park in Nevada, Rocky Mountain National Park and US Air Force Academy in Colorado. At the 5th percentile (**Fig.7c**) significant $O_3$ increases occur at most sites in the Northeast while little change and some negative trends are found in the Southeast. The occurrence of the greatest observed $O_3$ decreases for the highest percentiles are consistent with high-temperature $O_3$ production being more $NO_x$-limited (Pusede et al., 2015), and thus more responsive to decreases in $NO_x$ emissions.

The north-to-south gradient in springtime $O_3$ trends over the EUS reflects the earlier seasonal transition from $NO_x$-saturated to $NO_x$-sensitive $O_3$ production regimes in the Southeast, where plentiful radiation in spring enhances $HO_x$ supply and biogenic isoprene emissions begin earlier than in the Northeast. The different response of springtime $O_3$ to $NO_x$ controls in the Southeast versus Northeast noticed in this work is not present in prior analyses for shorter time periods (1990-2010 in Cooper et al. 2012 and 1998-2013 in Simon et al. 2015). We find 72% of the Southeast sites experiencing significant median $O_3$ decreases in spring over 1988-2014, while Cooper et al. found only 8%. Sites with significant 95th percentile springtime $O_3$ decreases in the EUS are also much more common in our study (85% versus 43% in Cooper et al.). In the 5th percentile, 45% of the Northeast sites in our analysis have significant spring $O_3$ increases, whereas only 15% in Cooper et al. Stronger $O_3$ reductions in the Southeast than the Northeast also occur during autumn (**Fig.S4**), reflecting an extension of biogenic isoprene emissions and $NO_x$-sensitive $O_3$ production in the Southeast to autumn.

**(Figure 8 about here)**

In summer (**Figure 8**), as radiation intensifies and isoprene emissions peak seasonally, the $O_3$ production becomes more $NO_x$-limited across both the Southeast and Northeast US where $NO_x$ emission controls have led to significant $O_3$ decreases of 0.8-1.8 ppb yr$^{-1}$ in the 95$^{th}$ percentile and 0.4-0.8 ppb yr$^{-1}$ in the median value (**Fig.8a-8b**). In the Southeast, significant decreases have also occurred at the lowest percentiles during summer (**Fig.8c**), in contrast to the weak response during spring (**Fig.7c**). Many northeast states in the late 1990s and early 2000s did not turn on power plant $NO_x$ emission controls until the $O_3$ season (May-September), which may contribute to observed differences between spring and summer $O_3$ trends. Compared to the 1990-2010 trends reported in Cooper et al., the EUS summer $O_3$ decreases reported here with additional data to 2014 are 33% stronger. Despite reductions in precursor emissions in the WUS cities (**Fig.1d**), there are no significant summer $O_3$ decreases at the intermountain sites except in Yosemite and Joshua Tree National Parks for the 95$^{th}$ percentile. Instead, a significant summer increase of ~0.3 ppb yr$^{-1}$ occurs across the entire $O_3$ distribution at Yellowstone. Significant summer increases are found in the 5$^{th}$ percentile for Lassen, Mesa Verde, and Rocky Mountain National Parks.

**(Figure 9 about here)**

In winter (**Figure 9**), observed $O_3$ increases are more common than in spring and summer across the US. The wintertime $O_3$ increases are strongest in the lowest percentiles over the EUS, indicating the influence from weakened $NO_x$ titration as a result of regional $NO_x$ emission controls (see also Gao et al., 2013; Clifton et al., 2014; Simon et al., 2015). Even during winter, some decreasing $O_3$ trends are found in the highest percentiles over the Southeast (**Fig.9a**), most prominent in Texas (Dallas and Houston), where tropical climate and year-round active photochemistry makes $O_3$ most responsive to regional $NO_x$ emission controls. Despite the greatest $NO_x$ emission reductions over the past decade in the central and northeast US regions, observed $O_3$ reductions have been most pronounced in the Southeast, particularly in spring and autumn.

**4.2 Model Evaluation and Attribution of Observed $O_3$ Trends**

The BASE simulation with GFDL-AM3 captures the salient features of observed $O_3$ trends over 1988-2014 at rural sites across the US: (1) the overall springtime increases and the lack of significant trends in summer over the Intermountain West, (2) the north-to-south gradients in $O_3$ trends during spring and the largest decreases in the 95$^{th}$ percentile during summer over the EUS, (3) wintertime increases in the 5$^{th}$ and 50$^{th}$ percentiles (left vs right panels in **Figs. 7 to 9**). AM3 also simulates a median springtime $O_3$ increase of 0.32±0.11 ppb yr$^{-1}$ over 1988-2014 (0.64±0.50 ppb yr$^{-1}$ over 2004-2014) at Mount Bachelor Observatory in Oregon, consistent with the positive trend (0.63±0.41 ppb yr$^{-1}$) observed over the shorter 2004-2015 period (Gratz et al., 2014). These analyses imply that GFDL-AM3 represents the underlying chemical and physical processes controlling the response of US surface $O_3$ means and extremes to changes in global-to-regional precursor emissions and climate, despite mean state biases (**Figs. S5-S6**).

The filtered model shows greater 95$^{th}$ percentile $O_3$ increases than observed at some WUS sites (e.g., Yosemite; Grand Canyon; Canyonlands) for both spring and summer (**Figs.7a,d and**

**Fig.8a,d**), reflecting that observations at these sites sometimes can be influenced by transport of photochemically aged plumes from nearby urban areas and from southern California during late spring and summer. When sampled at the surface, AM3 simulates small summertime $O_3$ decreases in the 95[th] and 50[th] percentiles over the Intermountain West (**Fig.4b,d),** consistent with observations at Yosemite, Grand Canyon, and Canyonlands (**Fig.8a,b**). As illustrated in **Fig.3** for spring and discussed in Sect. 2.4, individual sites in the west display observed trends falling in between the filtered model and those sampled at the surface versus aloft.

**(Figures 10 and 11 about here)**

We examine how US surface $O_3$ responds to changes in regional anthropogenic emissions, hemispheric background, and meteorology by comparing $O_3$ trends in the BASE, Background, and FIXEMIS experiments (**Figs. 10-11**). With North American anthropogenic emissions shut off in the Background simulation, little difference is discernable from the BASE simulation for WUS $O_3$ trends during spring (first vs. second rows in **Fig.10**), indicating the key role of hemispheric background driving increases in springtime $O_3$ over the WUS. With anthropogenic emissions held constant in time, FIXEMIS still shows statistically significant spring $O_3$ increases in the 95[th] percentile (**Fig.10c**), approximately half of the trends simulated in BASE, for Grand Canyon, Canyonlands, Mesa Verde and Rocky Mountain National Parks. Prior work shows that deep stratospheric intrusions contribute to the highest observed and simulated surface $O_3$ events at these sites (Langford et al., 2009; Lin et al., 2012a). Strong year-to-year variability of such intrusion events (Lin et al., 2015a) can confound the attribution of springtime $O_3$ changes over the WUS to anthropogenic emission trends, particularly in the highest percentile and over a short record length. Summer avoids this confounding influence when stratospheric intrusions are at their seasonal minimum, as evidenced by little $O_3$ change in FIXEMIS over the WUS (**Figs. 11c,f**). In contrast to spring, the model shows larger differences in WUS $O_3$ trends between BASE and Background for summer when North American pollution peaks seasonally (**Figs.10a,d vs. 10b,e compared to Figs.11a,d vs. 11b,e**). There are significant increases of 0.2-0.5 ppb yr$^{-1}$ in the 95[th] and 50[th] percentile summer background $O_3$ at more than 50% of the western sites (**Fig.11b,e**), offsetting the $O_3$ decreases resulting from US $NO_x$ reductions and leading to little overall change in total observed and simulated $O_3$ at WUS rural sites during summer (**Fig.8**).

Over the EUS, AM3 also simulates background $O_3$ increases, occurring in both the 95[th] and 50[th] percentiles, with a rate of 0.1-0.3 ppb yr$^{-1}$ during spring (**Fig.10b,e**) and 0.2-0.5 ppb yr$^{-1}$ during summer (**Fig.11b,e**). Based on prior model estimates that springtime background $O_3$ is greater in the Northeast than the Southeast (Lin et al., 2012a; Lin et al., 2012b; Fiore et al., 2014), one might assume that the springtime $O_3$ increases in the 5[th] percentile observed over the Northeast (**Fig.7c**) have been influenced by a rising background. However, AM3 simulates homogeneous background $O_3$ trends across the entire EUS (**Fig.10b,e**), indicating that the observed north-to-south gradient in $O_3$ trends reflects an earlier seasonal onset of $NO_x$-sensitive photochemistry in the Southeast as opposed to the background influence.

**(Figure 12 about here).**

A warming climate is most likely to worsen the highest $O_3$ events in polluted regions (e.g., Schnell et al., 2016; Shen et al., 2016). With anthropogenic emissions held constant in time over 1988-2014, FIXEMIS suggests significant increases of 0.2-0.4 ppb $yr^{-1}$ in the 95th percentile summertime $O_3$ over the EUS (**Fig.11c**). Using self-organizing map cluster analysis, Horton et al. (2015) identified robust increases in the occurrence of summer anticyclonic circulations over eastern North America since 1990. We find that biogenic isoprene emissions over this period increased significantly by 1-2% $yr^{-1}$ (10 to 20 mg C $m^{-2}$ $summer^{-1}$) throughout the EUS in the model, consistent with simulated increases in the 90th percentile JJA daily maximum temperature (**Fig. 12a-12b**). Increases in isoprene emissions contribute to raising EUS background $O_3$ in summer (**Fig.11b,e**). Using the Global Land-Based Datasets for Monitoring Climate Extremes (GHCNDEX; Donat et al., 2013), we find increases in the number of warm days above the 90th percentile and maximum temperature over the southeast US in August (**Fig.12c-12d**). The trends in temperature extremes are similar between June and August, but there is no significant trend in July (not shown). While changes in regional temperature extremes on 20 to 30-year time series may reflect internal climate variability (Shepherd, 2015), we suggest that increasing hot extremes and biogenic isoprene emissions over the last two decades may have offset some of the benefits of regional $NO_x$ reductions in the EUS.

**5. Impacts of rising Asian emissions, methane and wildfires on western US $O_3$**
**5.1 Historical western US $O_3$ trends in spring**

**(Figure 13 about here: Time series analysis)**

Further indications of the factors driving baseline $O_3$ changes over the WUS can be inferred by examining the time series at several high-elevation sites, which are most frequently sampling baseline $O_3$ in the free troposphere during spring (Sect. 2.4). **Figure 13** shows the results, both observed and simulated, for six such monitoring sites: Great Basin National Park in Nevada (2.1 km a.s.l.), Rocky Mountain National Park (2.7 km a.s.l.) in Colorado, US Air Force Academy (1.9 km a.s.l.) in Colorado Springs, Yellowstone National Park (2.4 km a.s.l.) and Pinedale (2.4 km a.s.l.) in Wyoming, and Mesa Verde National Park (2.2 km a.s.l.) in the Colorado-New Mexico-Arizona-Utah four corner region. The observed median values of springtime MDA8 $O_3$ have increased significantly at a rate of 0.2-0.5 ppb $yr^{-1}$ over the past 20-27 years at these sites, except Pinedale, where the increase in background $O_3$ is likely offset by the $O_3$ decrease due to recent emission control for the large oil and gas production fields in this area (http://deq.wyoming.gov/aqd/winter-ozone/resources/technical-documents/). When filtered to remove the influence from fresh local pollution (Sect.2.4), AM3 BASE captures the long-term trends of $O_3$ observed at these sites.

Correlating AM3 Background with observed $O_3$ indicates that most of the observed variability reflects changes in the background, with fluctuations in stratospheric influence contributing to anomalies on interannual time scales (e.g., the 1999 anomaly, Lin et al., 2015a), whereas Asian influence dominates the decadal trends as discussed below. The $O_3$ reduction resulting from US anthropogenic emission controls is less than 0.1 ppb $yr^{-1}$ (BASE minus Background) at these baseline sites. We show model results for the entire 1980-2014 period for

Great Basin, Rocky Mountain, and US Air Force Academy to provide context for observed trends in the two most recent decades (**Fig.13a**). In the 1980s when Chinese $NO_x$ emissions (~4 Tg/yr NO) were much lower than US $NO_x$ emissions (~15 Tg/yr NO) (Granier et al., 2011), there was little overall $O_3$ change over the WUS in the model. From the mid-1990s onwards, with $NO_x$ emissions in China rising steeply (**Fig.1a**) and surpassing US emissions in the 2000s, the $O_3$ trends at remote WUS sites appear to be dominated by trends of background, reflecting rising emissions outside the US. The largest spring $O_3$ increases from 1981-1990 to 2003-2012 at 700 hPa extend from Southeast Asia to the subtropical North Pacific Ocean to the southwestern US (**Fig.S7a**), consistent with the influence of rising Asian precursor emissions.

**(Table 2 about here: Trend attribution)**

**Table 2** contains a summary of the drivers of $O_3$ trends in the model at seven CASTNet sites that exhibit a significant spring $O_3$ increase observed over 1988-2012. Here we focus our attribution analysis on the period 1988-2012 (instead of 1988-2014) because the IAVASIA and IAVCH$_4$ simulations only extend to 2012. Meteorology varies from year to year in all experiments. Thus, we quantify the contributions of rising Asian emissions in IAVASIA, global methane in IAVCH$_4$, and wildfire emissions in IAVFIRE by subtracting out the slope of the linear regression of seasonal $O_3$ means in FIXEMIS. Simulated $O_3$ with anthropogenic emissions varying in both South and East Asia but held constant elsewhere shows statistically significant increases of 0.1-0.2 ppb yr$^{-1}$ (p≤0.01; IAVASIA minus FIXEMIS in **Table 2**), consistent with trends of 0.2 ppb yr$^{-1}$ estimated by scaling results from HTAP phase 1 multi-model sensitivity experiments with Asian emissions reduced by 20% (Riedmiller et al., 2009). This Asian influence can explain 50-65% of the modeled background $O_3$ increase in spring (**Table 2**).

With only methane varying, the model trends are less than 0.1 ppb yr$^{-1}$ (IAVCH$_4$ minus FIXEMIS), accounting for an average of 15% of the background increase. The contribution from wildfire emissions during spring is of minor importance (IAVFIRE minus FIXEMIS, **Table 2**). A stratospheric $O_3$ tracer (O$_3$Strat) in AM3 (Lin et al., 2012a; Lin et al., 2015a) demonstrates a positive but insignificant trend in stratospheric $O_3$ transport to the sites. We examine the trends of lower tropospheric $O_3$ at these sites when transport conditions favor the import of Asian pollution into western North America, as diagnosed by East Asian CO tracer (EACOt) exceeding the 67[th] percentile for each spring. Similar to the conclusion of Lin et al., (2015b), we find that the rate of $O_3$ increase in the Background simulation is greater by 0.05-0.1 ppb yr$^{-1}$ under strong transport from Asia than without filtering. Filtering the IAVASIA simulation for Asian influence also results in greater $O_3$ increases than filtering for baseline conditions (**Table 2**).

Rising Asian emissions even influence trends of $O_3$ downwind of the Los Angeles Basin during spring. $O_3$ measured in Joshua Tree National Park shows an increase of 0.31±0.25 ppb yr$^{-1}$ in spring over 1990-2010 (Cooper et al., 2012), despite significant improvements in $O_3$ air quality in the Los Angeles Basin (Warneke et al., 2012). The $O_3$ record extended to 2014 shows a decline in the 95[th] percentile $O_3$ in Joshua Tree National Park for both spring and summer (**Figs. 7-8**), whereas the 5[th] percentile continues to increase in spring and there is no significant trend in the median. Sampling the AM3 Background simulation at this site indicates rising background (0.31±0.14 ppb yr$^{-1}$). Aircraft measurements in May-June 2010 indicate the presence of Asian

pollution layers 2 km above southern California with distinct sulfate enhancements coincident with low organic mass (Lin et al., 2012b), supporting the conclusion that rising Asian emissions can contribute to trends of $O_3$ observed in this region. Yosemite National Park (1.6 km a.s.l.) and Chiricahua National Monument (1.5 km a.s.l.) are also influenced by increases in Asian emissions and concurrent decreases in local pollution in California. $O_3$ observed at Yosemite shows an increase from 1995 to around 2012 ($0.37\pm0.32$ ppb yr$^{-1}$; **Fig.S8**), which the model attributes primarily to rising Asian emissions (**Table 2**), but observations have remained constant since then, reflecting an offset by $O_3$ decreases in California (**Fig.4**).

**5.2 Projecting western US springtime $O_3$ for the 21$^{st}$ Century**

**(Figure 14 about here: Future Projections).**

Under the RCP8.5 scenario, Chinese $NO_x$ emissions are projected to peak in 2020-2030, reflecting an increase of ~50% from 2010 (**Fig.1a),** followed by a sharp decrease reaching 1990 levels by 2050. Global methane increases by ~60% from 2010 to 2050 under RCP8.5 (**Fig.S1**). Under the RCP4.5 scenario, in contrast, $NO_x$ emissions in China change little over 2010-2030 and global methane remains almost constant from 2010 to 2050. $NO_x$ emissions in the US decrease through 2050 under both scenarios, by ~40% from 2010. A number of studies have examined future US $O_3$ changes under the RCPs (e.g., Gao et al., 2013; Clifton et al., 2014; Pfister et al., 2014; Fiore et al., 2015; Barnes et al., 2016). However, as discussed earlier, the trends of $O_3$ in the model when sampled near the surface are overwhelmingly dominated by US anthropogenic emission trends. Thus, the future $O_3$ changes estimated by these prior studies do not represent baseline conditions, particularly the response to rising Asian emissions. In **Fig. 14** we show changes of WUS free tropospheric (700 hPa) $O_3$ relative to 2010 in the CM3 future simulations under RCP8.5 versus RCP4.5. Historical hindcasts and observations are also shown for context. Under RCP4.5, springtime $O_3$ over the WUS shows little overall change over 2010-2050. Under RCP8.5, in contrast, springtime WUS $O_3$ increases by ~10 ppb from 2010 to 2030 and remains almost constant from 2030 to 2050, consistent with the projected trends in Asian emissions and global methane.

**5.3 Trends and variability of western US $O_3$ in summer**

**(Figure 15 about here: Yellowstone)**

Yellowstone National Park is the only site with statistically significant summer $O_3$ increases observed across all percentiles (**Fig.8a-8c**). The 1988-2012 trends for the median observed and simulated $O_3$ are summarized in **Figure 15a**. Observations show an increase of $0.32\pm0.18$ ppb yr$^{-1}$ for JJA, with a greater rate of increase in June ($0.38\pm0.25$ ppb yr$^{-1}$) than in July-August ($0.26\pm0.18$ ppb yr$^{-1}$). AM3 BASE sampled at 700 hPa and filtered for baseline conditions (hatched pink bar in **Fig.15a**) captures the observed increase. Without baseline filtering (solid pink bar), North American emission reductions offset almost 50% of the simulated $O_3$ increase at Yellowstone, causing the model to underestimate the observed $O_3$ trend. The model attributes much of the observed summer $O_3$ increase at Yellowstone to rising Asian

emissions, with IAVASIA simulating an $O_3$ increase of 0.31±0.19 ppb $yr^{-1}$ under baseline
conditions, increasing to 0.42±0.23 ppb $yr^{-1}$ under conditions of Asian influence (EACOt ≥ 67[th]
percentile). The stronger increase measured in June than in July-August is consistent with the
influence of the Asian summer monsoon producing a surface $O_3$ minimum in July-August in East
Asia (e.g., Lin et al., 2009), as well as the seasonality of intercontinental pollution transport.
Changes in methane, wildfires, and meteorology over this period are of minor importance for the
JJA $O_3$ trends at Yellowstone.
Enhanced wildfire activity in hot and dry weather is thought to be a key driver of
interannual variability of surface $O_3$ in the Intermountain West in summer (Jaffe et al., 2008;
Jaffe, 2011). However, hot and dry conditions also facilitate the buildup of $O_3$ produced from
regional anthropogenic emissions, which can complicate the unambiguous attribution of
observed $O_3$ enhancements. Using August data at Yellowstone as an example, we isolate the
relative contribution of these two processes to observed $O_3$ with the IAVFIRE versus FIXEMIS
experiments (**Fig.15b**). Here we sample AM3 at the surface to account for any influence of
varying boundary layer mixing depths. Even without interannual variations of wildfire emissions,
FIXEMIS captures much of the observed year-to-year variability of August mean $O_3$ at
Yellowstone (r=0.67). IAVFIRE with interannually varying fire emissions only moderately
improves the correlations (r = 0.75). FIXEMIS also captures the observed $O_3$ increase from the
early 1990s to around 2002, likely reflecting warmer temperatures and deeper mixing depths
allowing more baseline $O_3$ to mix down to the surface. Over the entire 1988-2014 (or 1980-2014)
period, IAVFIRE gives ~0.1 ppb $yr^{-1}$ greater $O_3$ increases in August than FIXEMIS, consistent
with an overall increase in boreal wildfire activity **(Fig.S2 and Fig.S7b).**
**(Figure 16 about here: Wildfires)**
**Figure 16** shows year-to-year variability in surface MDA8 $O_3$ enhancements from wildfires
during summer, as diagnosed by the differences between IAVFIRE and FIXEMIS. The results
are shown for individual months since fires are highly episodic. During the summers of 1998,
2002, and 2003, biomass fires burned a large area of Siberia and parts of the North American
boreal forests, raising carbon monoxide across the Northern Hemisphere as detected from space
(Yurganov et al., 2005; van der Werf et al., 2010). Long-range transport of Siberian fire plumes
resulted in 2-6 ppb enhancements in surface MDA8 $O_3$ at the US west coast and in parts of the
Intermountain West in AM3. The model calculates enhancements in monthly mean MDA8 $O_3$ of
up to 8 ppb from the intense wildfire events in Northern California during July 2008 (Huang et
al., 2013; Pfister et al., 2013), over Texas-Mexico during June 2011 (Y Wang et al., 2015), and in
Wyoming-Utah during August 2012 (Jaffe et al., 2013). The AM3 estimates are roughly
consistent with a previous analysis of boundary layer aircraft data with and without fire
influences (as diagnosed by $CH_3CN$) during June 2008 over California (Pfister et al., 2013).
While fires during hot and dry summers clearly result in enhanced $O_3$ at individual sites
for some summers, the ability of AM3 with constant fire emissions to simulate variability of $O_3$
for a high (e.g., 1988; 2002; 2006) versus low (e.g., 1997; 2009) fire year (**Fig.15b**) indicates
that biomass burning is not the primary driver of observed $O_3$ interannual variability.
Year-to-year variability of JJA mean MDA8 $O_3$ observed at Yellowstone is strongly correlated
(r > 0.6) with observed large-scale variations in JJA mean daily maximum temperature across the
Intermountain West (**Fig.15c**). Correlations for other ground stations show a similar large-scale
feature. Similar to the conclusion from Zhang L. et al. (2014), our analysis indicates that the
correlation between $O_3$ and biomass burning reported by Jaffe et al. (2008, 2011) at rural sites
reflects common underlying correlations with temperature rather than a causal relationship of fire
on $O_3$. At remote mountain sites (e.g., Yellowstone), warmer surface temperatures lead to deeper
mixed layers that facilitate mixing of free tropospheric $O_3$-rich air down to the surface. At sites
near sources of air pollution, hot conditions enhance regional $O_3$ production and orographic
lifting of urban pollution to mountaintop sites during daytime, as occurs at Rocky Mountain
National Park located downwind of the Denver Metropolitan area during summer (**Sect. 5.4**).
Reactive volatile organic compound (VOC) emissions from fires may enhance $O_3$ production in
$NO_x$-rich urban areas (Baker et al., 2016), although evaluating these impacts needs
high-resolution models and better treatment of sub-grid scale fire plumes.
**5.4 Ozone Trends in the Denver Metropolitan Area**
**(Figure 17 about here: Denver)**
Efforts to improve air quality have led to a marked decrease in high-$O_3$ events in the Los
Angeles Basin as illustrated by the annual 4[th] highest MDA8 $O_3$ at Crestline – a regionally
representative monitor operated continuously from 1980 to present (**Fig.17a**). In striking contrast,
the 4[th] highest MDA8 $O_3$ in the Denver Metropolitan area shows little change over the past
decades, despite significant reductions in $NO_x$ (**Fig.1**) and CO emissions (-80% from 1990-2010;
Cooper et al., 2012). Recent field measurements indicate that increased VOC emissions from oil
and natural gas operations are an important source of $O_3$ precursors in the Denver-Julesberg
Basin (Gilman et al., 2013; Halliday et al., 2016; McDuffie et al., 2016). However, total VOC
emissions in Denver may not be increasing over time due to the marked reductions in VOC
emissions from vehicles (Bishop and Stedman, 2008; 2015). We seek insights into the causes of
the lack of significant $O_3$ responses to emission controls in Denver by separately analyzing
trends in spring and summer (**Fig.17b-17c**).
The ~200x200 km$^2$ AM3 model is not expected to resolve the urban-to-rural differences
between Rocky Mountain National Park and the Denver Metropolitan area. However, if observed
$O_3$ variability in Denver correlates with that at remote sites in the Intermountain West, then
model attribution for the remote sites can be used to infer sources of observed $O_3$ in Denver. This
is demonstrated in **Fig.17b** for spring using data at three representative sites in Denver: Rocky
Flats North, National Renewable Energy Lab (NREL), and Welby with continuous
measurements since the early 1990s. Year-to-year variability of median MDA8 $O_3$ at these sites
during spring correlates strongly with that in Great Basin National Park (r = 0.7), a fairly remote
site in Nevada not influenced by urban emissions from Denver. Median spring $O_3$ observations in
Denver increased significantly by ~0.3 ppb yr$^{-1}$ similar to the rate of increase in Great Basin
National Park which the model attributes to rising background (**Fig.13a**), implying that the
tripling of Asian emissions since 1990 also raised mean springtime $O_3$ in the Denver
Metropolitan area. Trends in the 95[th] percentile are statistically insignificant.

During summer, changes in regional emissions and temperature have the greatest impacts on the highest observed $O_3$ concentrations in polluted environments. **Fig.17c** shows times series of July-August 95[th] percentile MDA8 $O_3$ in Denver, together with the distribution of daily maximum temperature. In every year since 1993, the highest summer MDA8 $O_3$ observed at these sites exceeds the 70 ppb NAAQS level. There is a small negative trend that is swamped by large interannual variability. The summers with the highest observed $O_3$ coincide with those with the highest observed temperatures, such as 1998, 2003, 2007, 2011 and 2012. During these summers, enhancements of MDA8 $O_3$ were also recorded in Rocky Mountain National Park, reflecting enhanced lifting of pollution from Denver under warmer conditions (Brodin et al., 2010). Appling quantile regression (e.g., Porter et al., 2015) to daily observations at Rocky Flats North over 1993-2015, we find a 2 ppb °C[-1] sensitivity of 95[th] percentile July-August $O_3$ to changes in maximum daily temperature. We suggest that the substantial increases in extreme heat occurrence over central North America over the last two decades, as found by Horton et al. (2015), contribute to raising summer $O_3$ in Denver, which offsets $O_3$ reductions that otherwise would have occurred due to emission controls in Denver. Potential shifts in the $O_3$ photochemistry regime can also contribute to trends of summer $O_3$ in Denver, although advancing this knowledge would require a high-resolution air quality model.

## 6. Impacts of heat waves and droughts on eastern US summer $O_3$

**(Figure 18 about here: Interannual Variability)**

We discuss in this section interannual variability and long-term changes in summer $O_3$ over the EUS, where air stagnation and high temperatures typically yield the highest $O_3$ observed in surface air (e.g., Jacob and Winner 2009). Evaluating the ability of models to simulate the high-$O_3$ anomalies during historical heat waves and droughts is crucial to establishing confidence in the model projection of pollution extremes under a warming climate. **Figure 18a** shows comparisons of July mean MDA8 $O_3$ at one regionally representative site, the Pennsylvania State University (PSU) CASTNet site, from observations and model simulations. With time-varying emissions, the BASE model simulates an $O_3$ decrease (-0.45±0.32 ppb yr[-1]) consistent with observations (-0.67±0.33 ppb yr[-1]), and captures the observed July mean $O_3$ interannual variability (r = 0.82) that is correlated with large-scale variations in daily maximum temperature (r = 0.57). In particular, $O_3$ pollution extremes are successfully simulated during the EUS summer heat waves of 1988, 1995, 1999, 2002, 2011 and 2012 (Leibensperger et al., 2008; Fiore et al., 2015; Jia et al., 2016). Year-to-year variations in meteorology can explain 30% of the total observed $O_3$ variability (r = 0.55), as inferred by FIXEMIS with constant anthropogenic emissions. If US anthropogenic emissions remained at 1990s levels (as in FIXEMIS), then anomalies in July mean MDA8 $O_3$ would have been 10 ppb greater during the 2011 and 2012 heat waves. Loughner et al. (2014) found that half of the days in July 2011 would have been classified as $O_3$ exceedance days for much of the mid-Atlantic region if emissions had not declined.

**(Figure 19 about here: Changes in $O_3$ distribution)**

**Figure 19a** compares the probability density functions of MDA8 $O_3$ at 40 EUS surface

sites for JJA in the pre-NO$_x$ SIP Call (1988-2002) versus post-NO$_x$ SIP Call (2003-2014) periods
and during the extreme heat waves of 1988 versus 2012. Following the NO$_x$ SIP Call, the
probability distribution of observed JJA MDA8 O$_3$ over the EUS shifted downward (solid black
vs. dotted gray lines in **Fig.19a**). The median value declined by 9 ppb and the largest decreases
occurred in the upper tails, leading to weaker day-to-day O$_3$ variability and a narrower O$_3$ range
(standard deviation σ decreased from 16.4 to 12.9 ppb). These observed O$_3$ changes driven by
regional NO$_x$ reductions are even more prominent when comparing the heat waves of 1988
versus 2012 (solid purple vs. dotted brown lines in **Fig.19a**): σ = 22.3 vs. 13.4 ppb and median
value μ= 68.6 vs. 52.2 ppb.
**Fig.19b** shows the corresponding comparisons using the results from AM3 BASE.
Despite the high mean model bias (~20 ppb), AM3 captures the overall structure of the changes
in the surface O$_3$ distributions and thus the response of surface O$_3$ to the NO$_x$ SIP Call, including
the reductions of high-O$_3$ events during the heat wave of 2012 compared to 1988. Nevertheless,
there is a noticeable difference between the observations and simulations in the shape of MDA8
O$_3$ probability distributions for summer 1988, particularly in the upper tail of the distribution
above 110 ppb (purple lines in **Figs.19a vs. 19b**). The BASE model also underestimates the
observed July mean O$_3$ anomaly at PSU in 1988 by ~10 ppb (purple versus black dots in
**Fig.18a**). One possible explanation for these biases is that drought stress can effectively reduce
the O$_3$ deposition sink to vegetation, leading to an increase in surface O$_3$ concentrations as found
during the 2003 European heat wave (Solberg et al., 2008), whereas AM3 does not include
interannually varying dry deposition velocities.
The North American drought of 1988 ranks among the worst episodes of drought in the
US (e.g., Seager and Hoerling, 2014), with JJA soil moisture deficits occurring over the northern
Great Plains − Midwest region with magnitudes of 1-2.5 mm standardized departures from the
1979-2010 climatology (**Fig.19c**). Huang et al. (2016) found that monthly mean O$_3$ dry
deposition velocities (V$_{d,O3}$) for forests decreased by 33% over Texas during the dry summer of
2011. Based on this estimate, we conduct a sensitivity simulation for 1988 using BASE
emissions but decreasing monthly mean V$_{d,O3}$ from May to August by 35% in the areas over
North America (20°N-60°N) where soil moisture deficits in 1988 exceed -1.0σ mm (**Fig.19c**).
This experiment (hereafter referred to as IAVDEP) simulates ~10 ppb higher July mean MDA8
O$_3$ at PSU CASTNet site than the BASE model and matches the observed O$_3$ anomaly in 1988
relative to the record mean (green symbol in **Fig.18a**). The impact is largest (up to 15 ppb) on
days when observed MDA8 O$_3$ exceeds 100 ppb (**Fig.18b**; T$_{max}$ ≥30 °C). Simulated JJA MDA8
O$_3$ at EUS sites in IAVDEP shows an upward shift in the probability distribution, particularly in
the upper tail above 110 ppb (green vs. purple lines in **Fig.19b**), bringing it closer to observations
in 1988 (**Fig.19a**). The O$_3$ standard deviation in IAVDEP (σ = 18 ppb) shifts towards that in
observations (σ = 22 ppb) relative to the BASE model (σ = 16 ppb).
Quantile mapping can be applied to correct systematic distributional biases in surface O$_3$
compared to observations (Rieder et al., 2015), but this approach has limitations if there are
structural biases in the O$_3$ distribution due to missing physical processes in the model (e.g.,
variations of V$_{d,O3}$ with droughts). Travis et al. (2016) suggest that the National Emission

Inventory (NEI) for $NO_x$ from the US EPA is too high nationally by 50%. Decreasing US $NO_x$ emissions by this amount corrects their model bias for boundary layer $O_3$ by 12 ppb in the Southeast for summer 2013, while surface MDA8 $O_3$ in their model is still biased high by 6±14 ppb, which the authors attribute to excessive boundary layer mixing. US $NO_x$ emissions in the emission inventory used in AM3 (Sect. 2.2) are approximately 15% lower than those from the NEI. The 35% decrease in $NO_x$ emissions from the pre-$NO_x$ SIP Call to the post-$NO_x$ SIP Call in the model reduces mean $O_3$ by 8 ppb in the EUS, implying that the $NO_x$ emission bias could correct 40% of our model mean bias of ~20 ppb. These estimates support the idea that the common model biases in simulating surface $O_3$ over the Southeast US (e.g., Fiore et al., 2009) may partly reflect excessive $NO_x$ emissions. Some of the positive $O_3$ biases could be also due to the averaging over a deep vertical box in the model surface layer (~60 m in AM3) that can't resolve near-surface gradients (Travis et al., 2016).

**7. Conclusions and Recommendations**

Through an observational and modeling analysis of interannual variability and long-term trends in sources of $O_3$ over the past 35 years, we have identified the key drivers of $O_3$ pollution over the US. We initially evaluated the trends of $O_3$ in Asia resulting from rising Asian precursor emissions (**Figs.4-6**). Our synthesis of available observations and simulations indicates that surface and free tropospheric $O_3$ over East Asia has increased by 1-2 ppb yr$^{-1}$ since 1990 (i.e., 25-50 ppb over 25 years), with significant implications for regional air quality and global tropospheric $O_3$ burden. Shifting next to the US, we find 0.2-0.5 ppb yr$^{-1}$ increases in median springtime MDA8 $O_3$ measured at 50% of sixteen WUS rural sites, with 25% of the sites showing increases across the entire $O_3$ concentration distribution, despite stringent US domestic emission controls (**Fig. 7**). While many prior studies show that global models have difficulty simulating $O_3$ increases observed at rural baseline sites (e.g., Parrish et al., 2014; Strode et al., 2015), we reconcile observed and simulated $O_3$ trends in GFDL-AM3 with a novel baseline sampling approach (**Figs.3 and 13**). We suggest that the common model-observation disagreement in baseline $O_3$ trends reflects limitations of coarse-resolution global models in resolving observed baseline conditions. This representativeness problem can be addressed by filtering model $O_3$ for hemispheric-scale baseline conditions using the easy-to-implement, low-cost regional CO-like tracers. This approach allows trends of $O_3$ measured at baseline sites to be compared directly with multi-decadal global model hindcasts, such as those being conducted for the Chemistry-Climate Model Initiative (CCMI; Morgenstern et al., 2016).

The ability of the GFDL-AM3 model to reproduce observed US surface $O_3$ trends lends confidence in its application to attribute these observed trends to specific processes (**Figs.7 to 11**). We summarize the overall statistics in **Fig.20**, drawing upon the decadal mean $O_3$ changes from 1981-1990 to 2003-2012 in the BASE and sensitivity simulations. The changes in BASE are: over the WUS 4.3±1.8 ppb for spring and 1.6±1.2 ppb for summer; over the Northeast -1.8±1.7 ppb for spring and -6.0±2.0 ppb for summer; over the Southeast -3.9±1.4 ppb for spring and -7.5±1.6 ppb for summer. Increasing $O_3$ in the WUS under BASE coincides with an increase of background $O_3$ by 6.3±1.9 ppb for spring and 4.2±2.0 ppb for summer. Under conditions of

strong transport from Asia (East Asian COt $\geq 67^{th}$), the background trend rose to 7.6±2.2 ppb for spring and 6.0±2.1 ppb for summer (green dots in **Fig.20**). The WUS background $O_3$ increase reflects contributions from: increases in Asian anthropogenic emissions (accounting for 50% of background increase in spring; 52% in summer), rising global methane (13% in spring; 23% in summer), and variability in biomass burning (6% in spring; 12% in summer; excluding the meteorological influence).

We conclude that the increase in Asian anthropogenic emissions is the major driver of rising background $O_3$ over the WUS for both spring and summer in the past decades, with a lesser contribution from methane increases over this period. The tripling of Asian $NO_x$ emissions since 1990 contributes up to 65% of modeled springtime background $O_3$ increases (0.3-0.5 ppb $yr^{-1}$) over the WUS, outpacing $O_3$ decreases resulting from 50% US $NO_x$ emission controls ($\leq 0.1$ ppb $yr^{-1}$; **Table 2 and Fig.10**). Springtime $O_3$ observed in the Denver metropolitan area has increased at a rate similar to remote rural sites (**Fig. 17b**). Mean springtime $O_3$ above the WUS is projected to increase by ~10 ppb from 2010 to 2030 under the RCP8.5 global change scenario but to remain constant throughout 2010 to 2050 under the RCP4.5 scenario (**Fig.14**). As $NO_x$ emissions in China continue to decline in response to efforts to improve air quality (Krotkov et al., 2016; Liu et al., 2016), rising global methane and $NO_x$ emissions in the tropical countries (e.g., India) in Asia, where $O_3$ production is more efficient, may become more important in the coming decades. A global perspective is necessary when designing a strategy to meet US $O_3$ air quality objectives.

During summer, a tripling of Asian anthropogenic emissions from 1988 to 2014 approximately offsets the benefits of 50% reductions in US domestic emissions, leading to weak or insignificant $O_3$ trends observed at most WUS rural sites (**Figs.8 and 11**). Rising Asian emissions contribute to observed summertime $O_3$ increases (0.3 ppb $yr^{-1}$) at Yellowstone National Park. Our findings confirm the earliest projection of Jacob et al. (1999) with a tripling of Asian emissions. While wildfire emissions can result in 2-8 ppb enhancements to monthly mean $O_3$ at individual sites in some summers, they are not the primary driver of observed $O_3$ interannual variability over the Intermountain West (**Figs.15 and 16**). Instead, boundary layer depth, high temperatures and the associated buildup of $O_3$ produced from regional anthropogenic emissions contribute most to the observed interannual variability of $O_3$ in summer. Summertime $O_3$ measured in Denver during pollution episodes frequently exceeds the 70 ppb NAAQS level, with little overall trend despite stringent precursor emission controls (**Fig.17c**), likely due to the effects of more frequent occurrences of hot extremes in the last decade.

In the eastern US, if emissions had not declined, the $95^{th}$ percentile summertime $O_3$ would have increased by 0.2-0.4 ppb $yr^{-1}$ over 1988-2014 (**Fig.11c**), due to more frequent hot summer extremes and increases in biogenic isoprene emissions (1-2% $yr^{-1}$) over this period (**Fig.12**). Regional $NO_x$ reductions alleviated the $O_3$ buildup during the recent heat waves of 2011 and 2012 relative to earlier heat waves (e.g., 1988; 1995; 1999). GFDL-AM3 captures year-to-year variability in monthly mean $O_3$ enhancements associated with large-scale variations in temperatures (**Figs. 18 and 19**). However, there is a need to improve the model representation of $O_3$ deposition sink to vegetation, in particular its reduced efficiency under drought stress, as

we demonstrated for the severe North American drought of 1988. Such land-biosphere couplings are poorly represented in current models and further work is needed to examine their impacts on $O_3$ pollution extremes in a warming climate.

Following the $NO_x$ SIP Call, surface $O_3$ in the eastern US declined throughout its probability distribution, with the largest decreases occurring in the highest percentiles during summer (-0.8 to -1.8 ppb $yr^{-1}$; **Fig.8**). Spatially, historical $O_3$ decreases during non-summer seasons were more pronounced in the Southeast, where the seasonal onset of biogenic isoprene emissions and $NO_x$-sensitive $O_3$ production occurs earlier than in the Northeast (**Figs.7, 9** and **S4**). The $95^{th}$ percentile $O_3$ concentration in the Southeast has even decreased during winter. Despite high mean-state biases, GFDL-AM3 captures the salient features of observed $O_3$ trends over the eastern US, including wintertime increases in the $5^{th}$ and $50^{th}$ percentiles in the Northeast, greater springtime decreases in the Southeast than the Northeast, and summertime decreases throughout the $O_3$ concentration distribution. These results suggest that $NO_x$ emission controls will continue to provide long-term $O_3$ air quality benefits in the Southeast US during all seasons.

**Acknowledgments**. This work was supported by funding from the NASA grants NNH13ZDA001N-AURAST and NNX14AR47G to M.Y. Lin. We thank O. Cooper, S. Fan and J. Schnell for helpful comments on the manuscript. We acknowledge the free use of ozonesonde data at Hong Kong available on woudc.org and GOME-SCIAMACHY tropospheric $NO_2$ column data available on www.temis.nl. AMF acknowledges support under EPA Assistance Agreement No. 83587801. The views expressed in this document are solely those of the authors and do not necessarily reflect those of the Agency. M.Y. Lin devotes this article to her father Tianci Lin who is the motivation of her life and research career.

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

3 **Table 1** Summary of forcings and emissions used in AM3 hindcasts and CM3 projections

| Experiment | Time Periods | Meteorology | Radiative forcings | $CH_4$ (chemistry) | Anthropogenic emissions | Fire Emissions |
|---|---|---|---|---|---|---|
| BASE | 1979-2014 | Nudged to NCEP | Historical | Historical | Historical | Historical |
| Background | 1979-2014 | as BASE | Historical | Historical | Zeroed out in N. America; As BASE elsewhere | Historical |
| FIXEMIS | 1979-2014 | as BASE | Historical | 2000 | Constant* | Constant* |
| IAVFIRE | 1979-2014 | as BASE | Historical | 2000 | Constant* | Historical |
| IAVASIA | 1979-2012[+] | as BASE | Historical | 2000 | Varying in Asia as BASE; as in FIXEMIS elsewhere | Constant* |
| $IAVCH_4$ | 1979-2012[+] | as BASE | Historical | Historical | Constant* | Constant* |
| CM3_RCP4.5 | 2005-2050 | Free running | RCP4.5 | RCP4.5 | RCP4.5 | RCP4.5 |
| CM3_RCP8.5 | 2005-2050 | Free running | RCP8.5 | RCP8.5 | RCP8.5 | RCP8.5 |

5    *Averaged over the whole 1970-2010 period.

6    +Note that the IAVASIA and $IAVCH_4$ simulations only extend to 2012.

**Table 2.** Summary of springtime median MDA8 $O_3$ trends (in ppb $yr^{-1}$) over 1988-2012 at WUS sites from observations and AM3 simulations. Trends with the 95% confidence intervals and levels of significance (**bold**: <1%; *italic*, 1-5%; ▯, ≥5%) were estimated by the two-tailed *t*-test.

| Experiment[a] | Lassen | Great Basin | Rocky Mountain | Mesa Verde | Yellowstone | Yosemite | Chiricahua |
|---|---|---|---|---|---|---|---|
| Observed | **0.38±0.14** | **0.38±0.26** | **0.37±0.18** | **0.30±0.18** | *0.21±0.19* | *0.37±0.32* | **0.17±0.10** |
| BASE* | **0.33±0.11** | **0.34±0.12** | **0.32±0.13** | **0.37±0.14** | **0.21±0.11** | **0.35±0.17** | *0.25±0.19* |
| Background | **0.31±0.12** | **0.40±0.13** | **0.45±0.13** | **0.43±0.17** | **0.30±0.11** | **0.41±0.16** | **0.32±0.21** |
| Background$_{EA}$ | **0.41±0.12** | **0.39±0.18** | **0.50±0.15** | **0.52±0.20** | **0.40±0.16** | **0.47±0.17** | **0.47±0.21** |
| IAVASIA* | **0.29±0.13** | **0.31±0.11** | **0.25±0.11** | **0.27±0.11** | **0.19±0.11** | **0.24±0.14** | 0.15±0.15 |
| IAVASIA$_{EA}$ | **0.26±0.16** | **0.26±0.16** | **0.35±0.13** | **0.32±0.13** | **0.27±0.16** | **0.31±0.18** | **0.25±0.15** |
| IAVCH$_4$* | *0.18±0.12* | **0.20±0.11** | *0.12±0.09* | *0.16±0.12* | 0.09±0.12 | *0.15±0.16* | *0.04±0.15* |
| IAVFIRE | 0.10±0.12 | *0.14±0.12* | *0.17±0.14* | *0.16±0.14* | *0.11±0.13* | *0.15±0.16* | 0.08±0.17 |
| FIXEMIS | 0.08±0.12 | *0.12±0.12* | *0.16±0.12* | *0.13±0.12* | 0.09±0.13 | *0.12±0.16* | *0.04±0.16* |
| O$_3$Strat | 0.18±0.18 | 0.20±0.25 | 0.18±0.18 | *0.25±0.23* | 0.15±0.18 | *0.27±0.30* | 0.07±0.24 |

a. The * mask indicates data filtered to represent baseline conditions (NACOt ≤67[th]). The EA subscript indicates that data were filtered to represent transport conditions favoring the import of Asian pollution (EACOt ≥ 67[th]).

**Figure captions**

**Figure 1. Changes in NO$_x$ emissions**. (a-b) Mean annual vertical column densities of tropospheric (VCDtrop) NO$_2$ normalized to the year 2000 for the Eastern China and Eastern US domains (black boxes on map) from GOME (1996-2002, open circles) and SCIAMACHY (2003-2011, closed circles) measurements and AM3 BASE simulations (orange lines). Triangles indicate trends in NO$_x$ emissions (normalized to 2000) from Lamarque et al. (2010) with annual interpolation after 2000 to RCP8.5 (red) versus RCP4.5 (blue). (c-d) Differences in annual mean SCIAMACHY VCDtrop NO$_2$ from 2003-2005 to 2009-2011. The red boxes denote the regions where emissions vary over time in the IAVASIA simulation (Table 1). Satellite NO$_2$ data are from www.temis.nl, with retrieval technique described in Boersma et al. (2004).

**Figure 2. Measurement uncertainties**. (a) Comparison of observed monthly mean MDA8 O$_3$ at WUS CASTNet sites. All sites have more than 90% data availability in every month shown. The grey shading denotes the period when data at Yellowstone (red) and Rocky Mountain (black) were inconsistent with the other sites. (b-c) The 1990-2010 trends of median JJA MDA8 O$_3$ at Yellowstone and median MAM MDA8 O$_3$ at Rocky Mountain with and without data in 1990.

**Figure 3. Influence of baseline sampling**.   Median spring MDA8 O$_3$ trends over 1988-2014 at WUS sites from: (a) Observations; (b) BASE model sampled at the surface; (c) BASE sampled at 700 hPa and filtered to remove the influence from fresh local pollution (see Sect. 2.4); (d) BASE sampled at 700 hPa without filtering; and (e-f) Background (with North American anthropogenic emissions shut off) sampled at the surface versus at 700 hPa. Note that three low-elevation (<1.5 km) sites Joshua Tree, Big Bend and Glacier National Parks are always sampled at the surface. Larger circles indicate sites with statistically significant trends (p<0.05).

**Figure 4.** Global distribution of MDA8 O$_3$ trends from AM3 BASE over 1988-2014 for boreal spring (left) and summer (right) for the 95th percentile at the surface (a-b), median at the surface (c-d), and median in the free troposphere (700 hPa; e-f). Stippling indicates areas where the trend is statistically significant (p<0.05). The color scale is designed to resolve regional features rather than extreme values and saturates. The range of trends is -1 to +2.5 ppb yr$^{-1}$.

**Figure 5.** (a) Time series of changes in global tropospheric O$_3$ burden relative to the 1981-1990

mean from BASE and FIXEMIS simulations (Table 1). (b) Time series of 12-month running mean anomalies (relative to the 2005-2014 mean) of $O_3$ averaged over 900-600 hPa at Hong Kong from: the averages of ozonesonde samples (black circles) and BASE model co-sampled on sonde launch days (orange circles) versus the true average from BASE and IAVFIRE with continuous daily sampling (solid lines). (c) Same as (b) but for Hanoi.

**Figure 6. Surface $O_3$ trends in Asia.** (a) Observation sites superimposed on a map of the 95th percentile summer MDA8 $O_3$ trends over 1995-2014 from AM3 BASE. (b) Comparison of median $O_3$ trends from AM3 (1995-2014) with observations (see text for periods): in Central Eastern China at Mt. Tai (July-August, Sun et al. 2016), Beijing (May-June-July, Ding et al. 2008) and Shangdianzi (SDZ) (JJA, Ma et al. 2016); in South China at Hong Kong (HK) (annual average, Wang et al. 2009) and Taiwan (MAM, Lin YK et al. 2010); at Mt. Waliguan (WLG) in western China (MAM, Xu et al. 2016); at South Korea (JJA, Lee et al. 2014) and Mt. Happo Japan (MAM, Tanimoto 2009). For Mt. Happo (triangle on map) AM3 is sampled at 700 hPa and filtered for the influence from Asian continental air - more representative of observed baseline conditions in spring.

**Figure 7**. Linear trends in spring (MAM) MDA8 $O_3$ over 1988-2014 at US rural sites for the $95^{th}$, $50^{th}$, and $5^{th}$ percentiles as observed (left) and simulated (right) in AM3 BASE. Larger circles indicate sites with statistically significant trends ($p<0.05$). For WUS high-elevation sites, the model is sampled at 700 hPa and filtered to remove local influence (see text in Sect. 2.4).

**Figure 8**. As in Figure 7, but for summer (JJA). Note that the color scale saturates at ±0.8.

**Figure 9**. As in Figure 7, but for winter (DJF). Large squares in (a) denote AQS sites with significant $O_3$ decreases in the $95^{th}$ percentile.

**Figure 10**. Linear trends in the $95^{th}$ (left) and $50^{th}$ (right) percentile springtime MDA8 $O_3$ over 1988-2014 at US rural sites from BASE (top), Background (middle) and FIXEMIS simulations (bottom). Larger circles indicate sites with statistically significant trends ($p<0.05$). Top panels are repeated from Fig.7d,e. Note that the $95^{th}$ ($50^{th}$) percentile is sampled separately from the Background and FIXEMIS simulations without depending on the times when the BASE simulation is experiencing the $95^{th}$ ($50^{th}$) percentile days.

**Figure 11**. As in Figure 10, but for summer. Top panels are repeated from Fig. 8d,e.

**Figure 12**. The 1990-2012 trends in: (a) model JJA total biogenic isoprene emissions; (b) model 90th percentile JJA daily maximum temperature; (c) the warmest daily maximum temperature and (d) the frequency of warm days (i.e., those above the 90th percentile for the base period 1961-90) for August obtained from GHCNDEX dataset (Donat et al., 2013; available at http://www.climdex.org/view download.html). Stippling denotes areas where the change is statistically significant ($p<0.05$). Note that the trends are calculated for the 1990-2012 period, instead of 1988-2014, to avoid the influence from hot extremes in 1988 and cold conditions in 2014 (Sect. 6). When these years are included, the trends in (c) and (d) are swamped by the anomalies. The trends in (a) and (b) are similar between 1990-2012 and 1988-2014.

**Figure 13a**. Time series of median spring MDA8 $O_3$ anomalies (relative to the 1995-2014 mean) at Great Basin, Rocky Mountain, and US Air Force Academy as observed (black) and simulated in AM3 BASE filtered for baseline conditions (red, see Sect.2.4) and in Background with North American anthropogenic emissions zeroed out (NAB; green). Presented on the top of the graph are statistics from the linear fit and correlations between observations and simulations. Numbers on the bottom of the graph denote the sample size of observations for each year. Grey dots indicate uncertain observations that are removed from the linear fit (see Sect. 2.3).

**Figure 13b**. Same as Figure 13a, but for Yellowstone, Pinedale, and Mesa Verde over the period 1988-2012.

**Figure 14. Future projections.** Time series of median springtime $O_3$ changes relative to 2010 in GFDL AM3 hindcast (orange circles) and CM3 future simulations for RCP8.5 (red) versus RCP4.5 (blue; shading represents the range of three ensemble members), sampled at 700 hPa over the WUS (35-45N,120-105W). Black circles indicate observed changes averaged from Lassen, Great Basin, and Rocky Mountain National Parks.

**Figure 15. Summertime $O_3$ in Yellowstone National Park.** (a) Median JJA MDA8 $O_3$ trends over 1988-2012 at Yellowstone from observations (black) and simulations sampled at 700 hPa

for BASE without filtering (pink), BASE filtered for baseline conditions (hatched pink),
IAVASIA (solid purple, baseline), IAVASIA filtered for Asian influence (EACOt≥67th, hatched
purple), IAVCH4 (cyan), IAVFIRE (orange) and FIXEMIS (red). (b) Time series of anomalies in
August median MDA8 $O_3$ at Yellowstone as observed (black) and simulated by the model
sampled at the surface, with constant (red) and time-varying wildfire emissions (orange). Trends
over 1988-2014 are reported. (c) Interannual correlations of JJA mean MDA8 $O_3$ observed at
Yellowstone with JJA mean daily maximum temperature from observations (Harris et al., 2014).
**Figure 16.** Surface MDA8 $O_3$ enhancements from wildfire emissions for individual months in
the years with large biomass burning in boreal regions (1998, 2002, 2003) and over the WUS
(2008, 2011, 2012), as diagnosed by the differences between IAVFIRE and FIXEMIS. The black
circle denotes the location of Yellowstone National Park.
**Figure 17. Surface $O_3$ trends in Denver.** (a) Comparison of observed trends in annual 4th
highest MDA8 $O_3$ at Crestline Los Angeles (brown) and in Denver (blue, computed from all
monitors available in Denver non-attainment counties). (b) Time series of observed median
MAM MDA8 $O_3$ at Great Basin National Park (red), in comparison with three monitors in
Denver. (c) Time series of observed 95th percentile July-August MDA8 $O_3$ in Denver, together
with statistics ($25^{th}$, $50^{th}$, $75^{th}$, $95^{th}$) of observed July-August daily maximum temperature at
Rocky Flats (red, right axis).
**Figure 18.** (a) Time series of July mean MDA8 $O_3$ anomalies (relative to 1988-2014) at the
Pennsylvania State University (PSU) CASTNET site as observed (black) and simulated by the
GFDL-AM3 model with time-varying (purple) and constant anthropogenic emissions (red),
along with observed anomalies in July mean daily max temperature (gray lines; right axis). The
green triangle denotes the 1988 $O_3$ anomaly from a sensitivity simulation using BASE emissions
but with 35% decreases in $V_{d,O3}$ (IAVDEP). (b) Time series of daily MDA8 $O_3$ at PSU from June 1
to July 16 in 1988 from observations (black), BASE (purple), and IAVDEP simulations (green).
**Figure 19.** (a) Comparisons of probability distributions of summertime MDA8 $O_3$ from 40 EUS
CASTNet sites for the pre-NOx SIP Call (1988-2002; solid black) versus post-NOx SIP Call
(2003-2014; dashed gray) periods and during the extreme heat waves of 1988 (solid purple)
versus 2012 (dashed brown). The median (μ) and standard deviation (σ) are shown (ppb). (b)

Same as (a) but from AM3 BASE. Also shown is the $O_3$ distribution in 1988 from a sensitivity simulation with 35% decreases in $V_{d,O3}$ in drought areas (green). (c) Standardized soil moisture departures for JJA 1988 (calculated by dividing anomalies by the 1979-2010 climatological standard deviation, using data from NOAA Climate Prediction Center).

**Figure 20. Summary of US surface $O_3$ trends and drivers.** Changes in decadal mean MDA8 $O_3$ from 1981-1990 to 2003-2012 simulated in a suite of GFDL-AM3 experiments for spring and summer for the western (32N-46N and 123W-102W), Northeast (37N-45N and 90W-65W) and Southeast (30N-36N and 95W-77W) US domains. Observations are not shown because limited data are available during 1981-1990. Experiments are color-coded with the error bars indicating the range of the mean change at the 95% confidence level. Filled circles represent the changes under Background (green) and IAVASIA (purple) when filtered for Asian influence (EACOt $\geq$ 67[th]), while other results are from the unfiltered models. The text near the bottom of the plot provides the change in $NO_x$ emissions over the same period for each region.

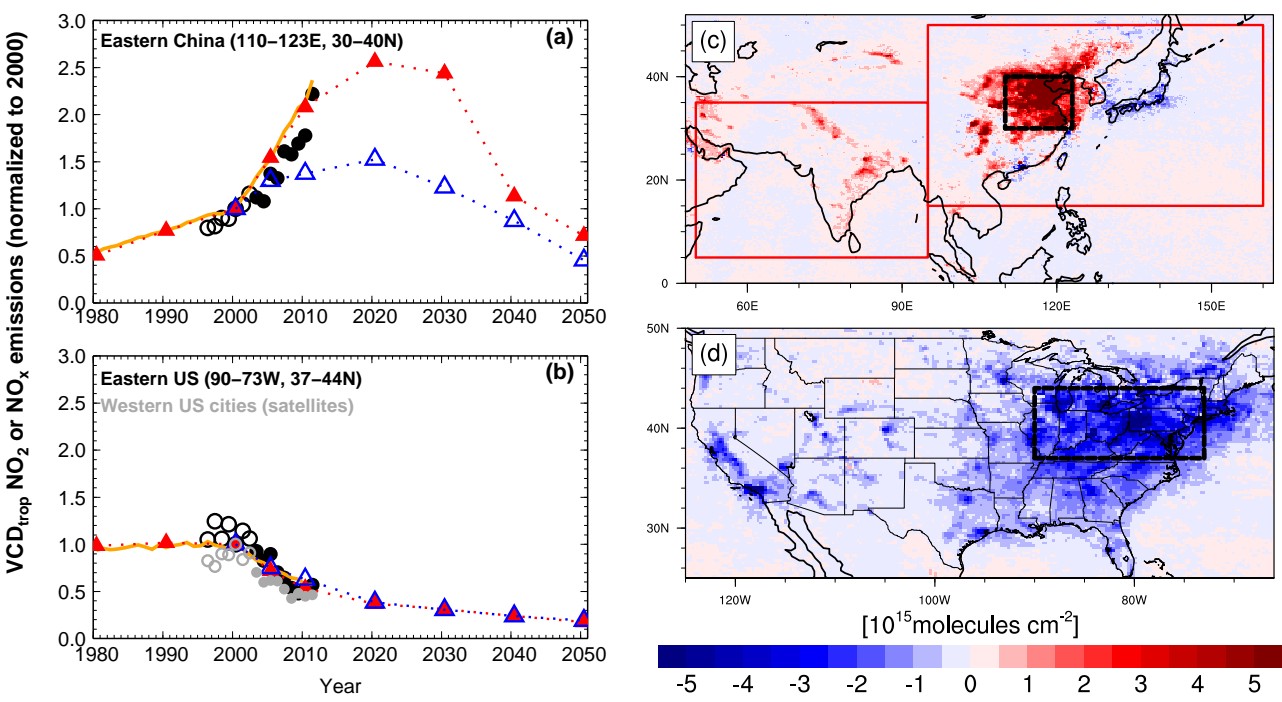

Figure 1. **Changes in $NO_x$ emissions**. (a-b) Mean annual vertical column densities of tropospheric ($VCD_{trop}$) $NO_2$ normalized to year 2000 for the Eastern China and Eastern US domains (black boxes on map) from GOME (1996-2002, open circles) and SCIAMACHY (2003-2011, closed circles) measurements and AM3 BASE simulations (orange lines). Triangles indicate trends in $NO_x$ emissions (normalized to 2000) from Lamarque et al. (2010) with annual interpolation after 2000 to RCP8.5 (red) versus RCP4.5 (blue). (c-d) Differences in annual mean SCIAMACHY $VCD_{trop}$ $NO_2$ from 2003-2005 to 2009-2011. The red boxes denote the regions where emissions vary over time in the IAVASIA simulation (Table 1). Satellite $NO_2$ data are from www.temis.nl, with retrieval technique described in Boersma et al.(2004).

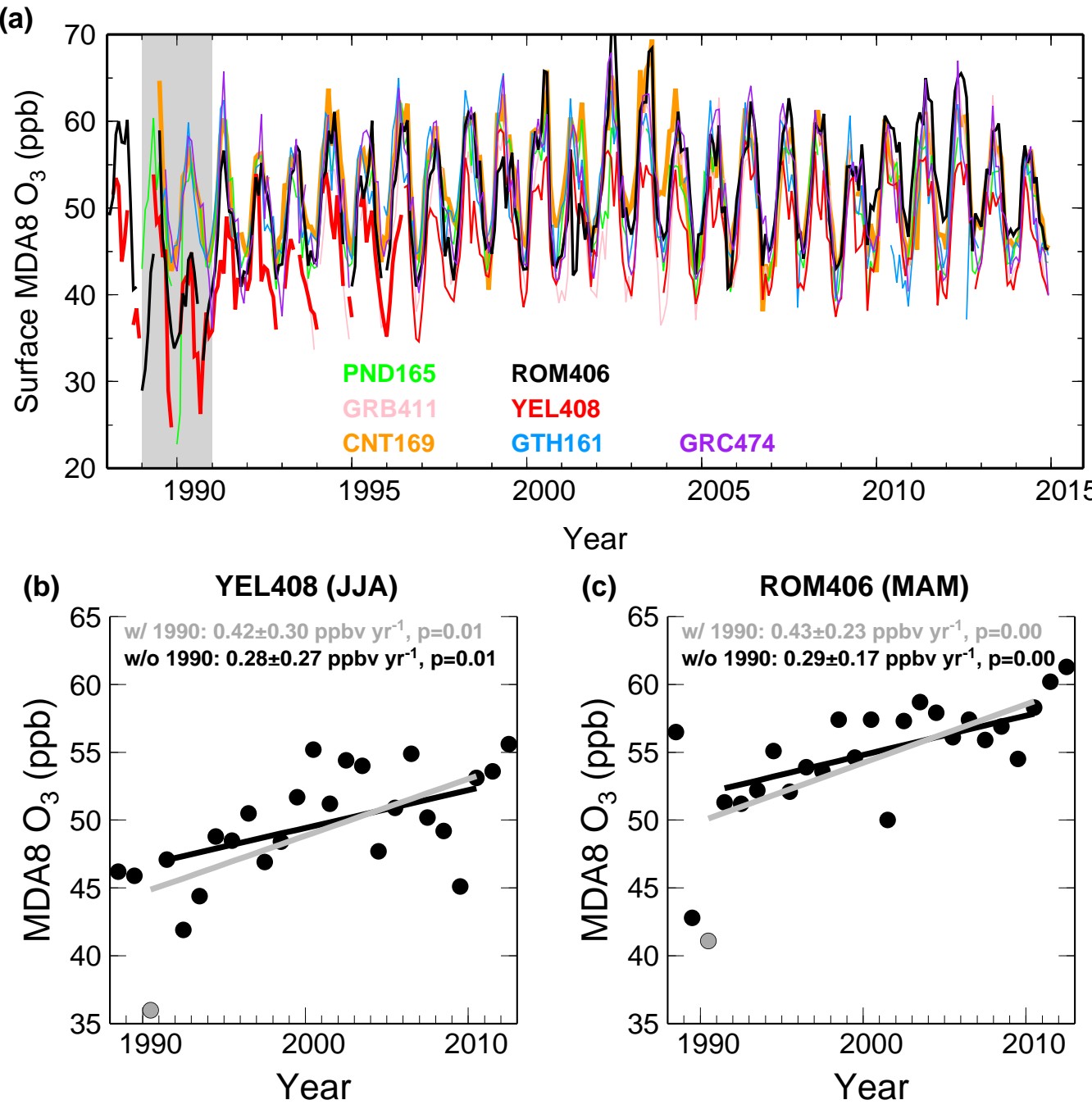

Figure 2. **Measurement uncertainties**. (a) Comparison of observed monthly mean MDA8 $O_3$ at WUS CASTNet sites. All sites have more than 90% data availability in every month shown. The grey shading denotes the period when data at Yellowstone (red) and Rocky Mountain (black) were inconsistent with the other sites. (b-c) The 1990-2010 trends of median JJA MDA8 $O_3$ at Yellowstone and median MAM MDA8 $O_3$ at Rocky Mountain with and without data in 1990.

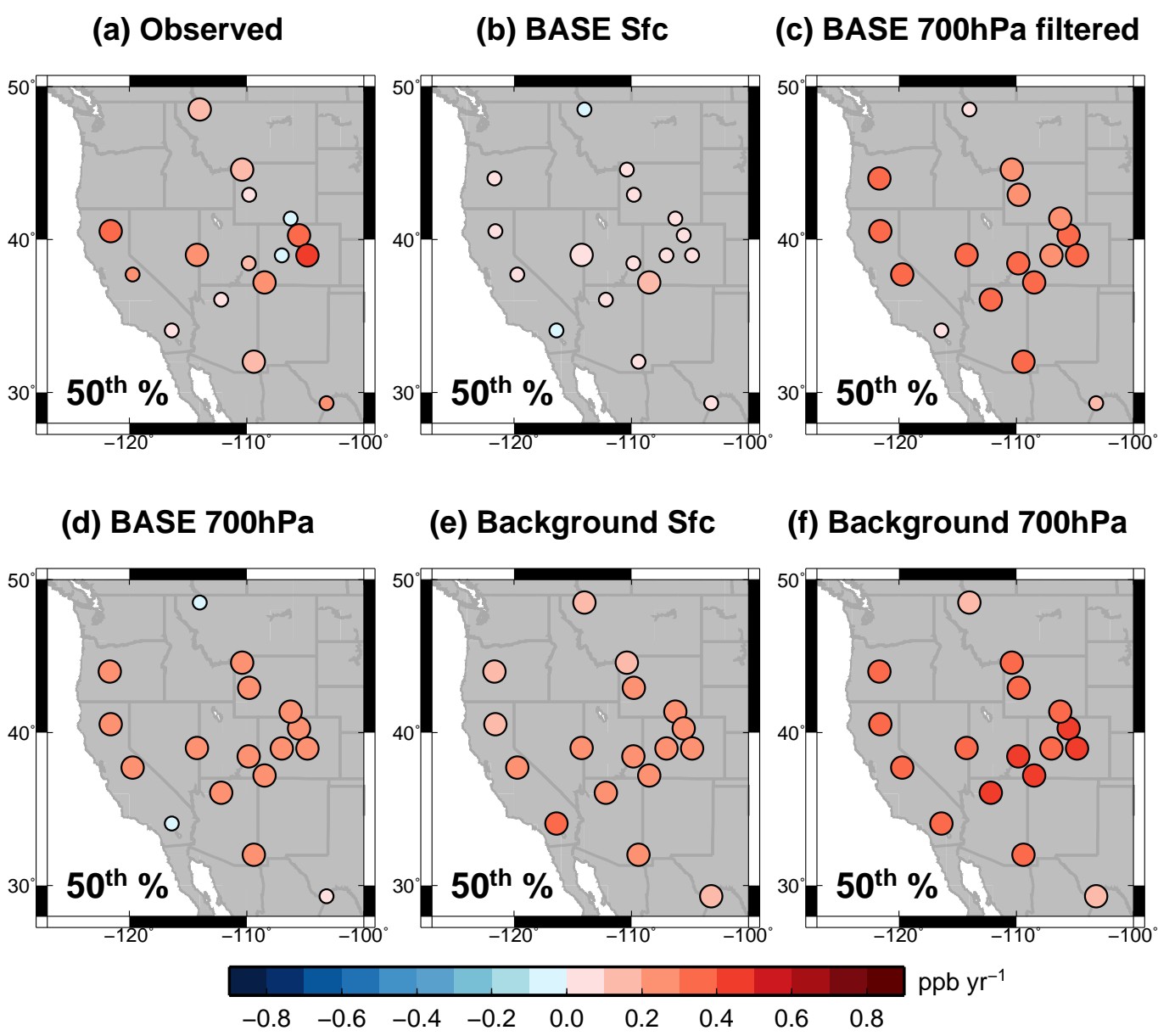

Figure 3. **Influence of baseline sampling.** Median spring MDA8 O$_3$ trends over 1988-2014 at WUS sites from: (a) Observations; (b) BASE model sampled at the surface; (c) BASE sampled at 700 hPa and filtered to remove the influence from fresh local pollution (see Sect. 2.4); (d) BASE sampled at 700 hPa without filtering; and (e-f) Background (with North American anthropogenic emissions shut off) sampled at the surface versus at 700 hPa. Note that three low-elevation (<1.5 km) sites Joshua Tree, Big Bend and Glacier National Parks are always sampled at the surface. Larger circles indicate sites with statistically significant trends (p<0.05).

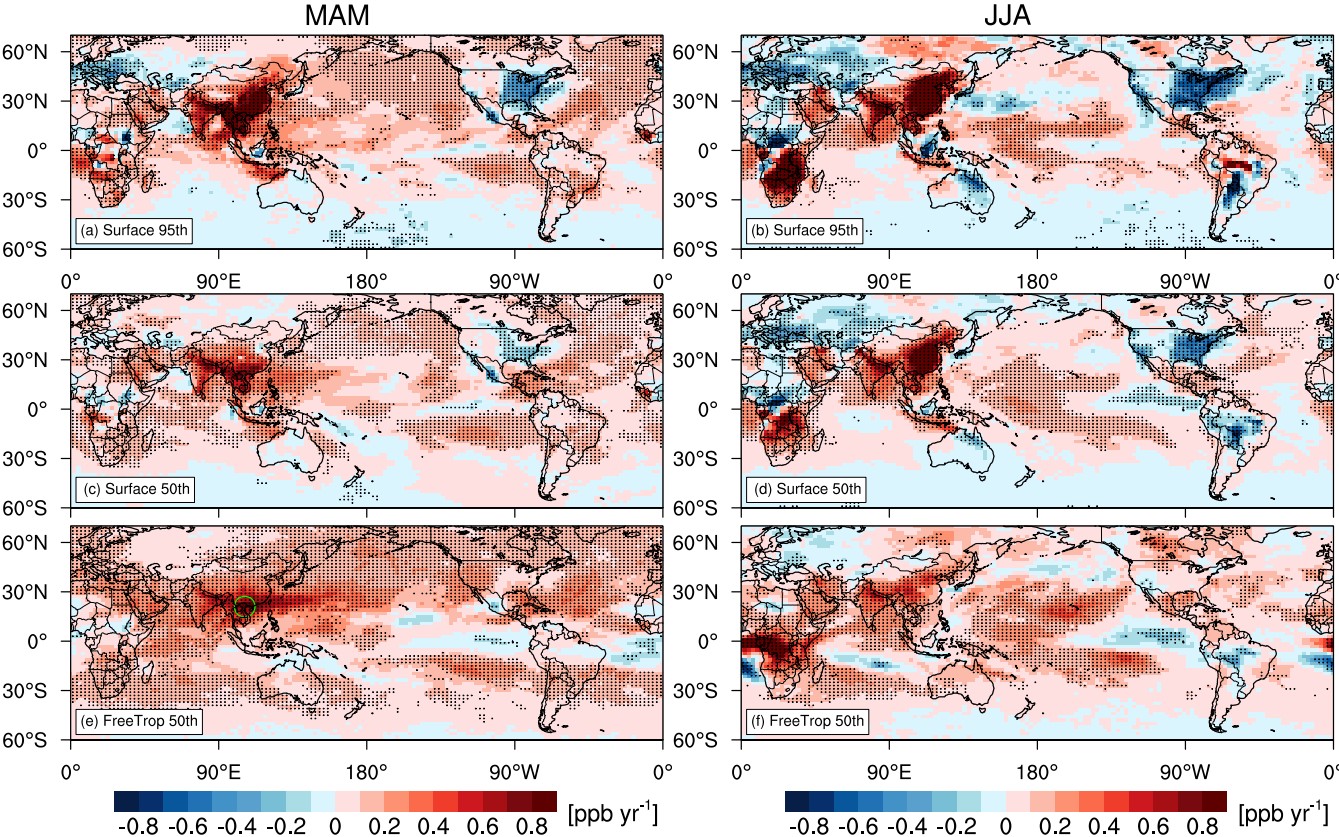

Figure 4. Global distribution of MDA8 $O_3$ trends from AM3 BASE over 1988-2014 for boreal spring (left) and summer (right) for the 95th percentile at the surface (a-b), median at the surface (c-d), and median in the free troposphere (700 hPa; e-f). Stippling indicates areas where the trend is statistically significant (p<0.05). The color scale is designed to resolve regional features rather than extreme values and saturates. The range of trends is -1 to +2.5 ppb $yr^{-1}$.

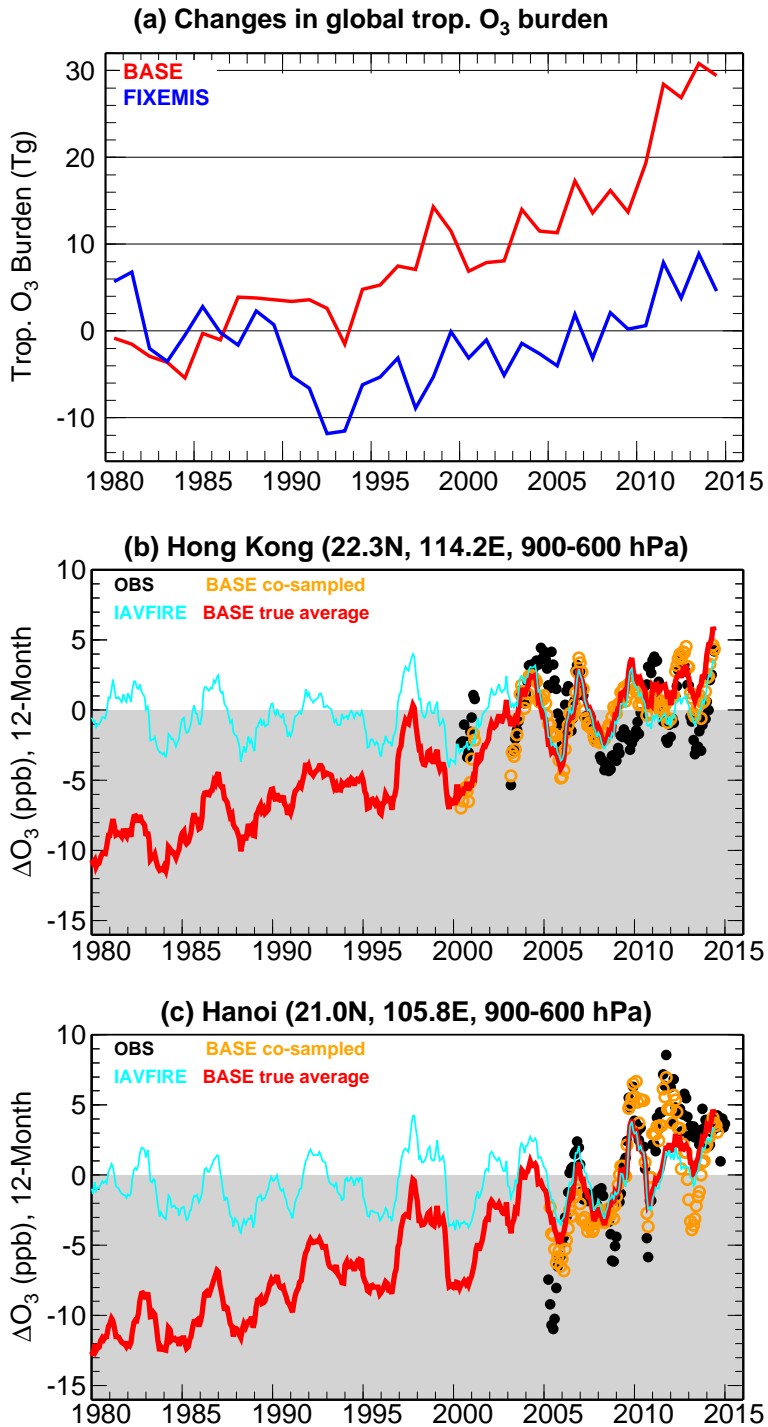

Figure 5. (a) Time series of changes in global tropospheric $O_3$ burden relative to the 1981-1990 mean from BASE and FIXEMIS simulations (Table 1). (b) Time series of 12-month running mean anomalies (relative to the 2005-2014 mean) of $O_3$ averaged over 900-600 hPa at Hong Kong from: the averages of ozonesonde samples (black circles) and BASE model co-sampled on sonde launch days (orange circles) versus the true average from BASE and IAVFIRE with continuous daily sampling (solid lines). (c) Same as (b) but for Hanoi.

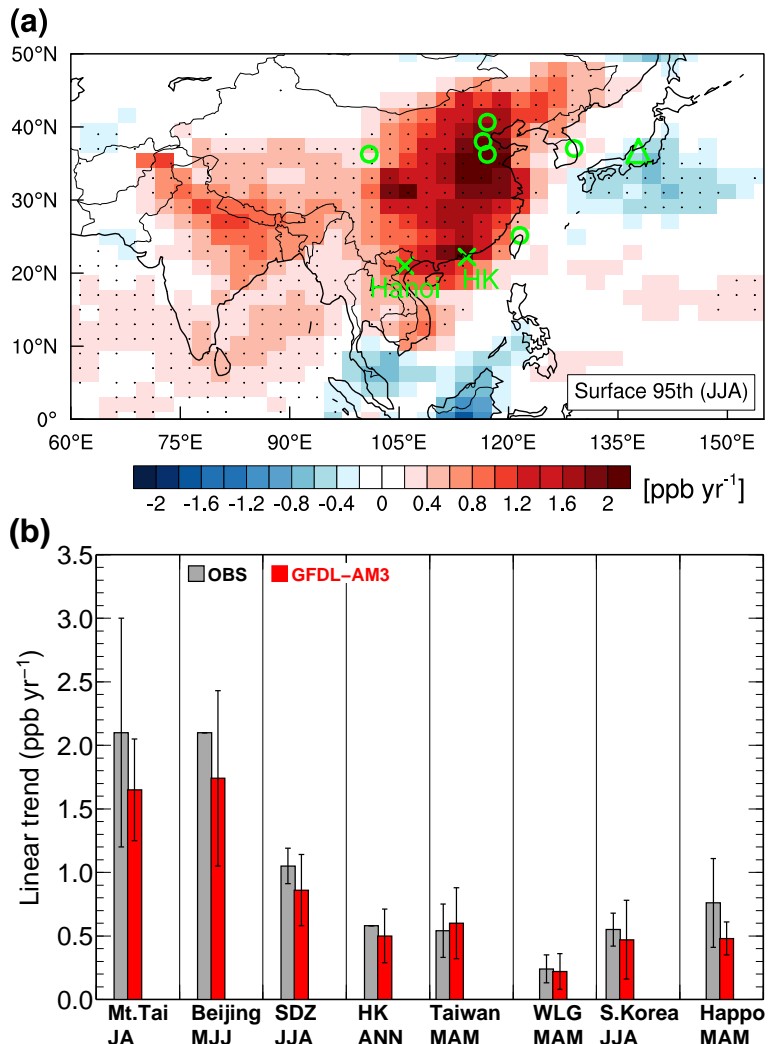

Figure 6. **Surface O$_3$ trends in Asia. (a)** Observation sites superimposed on a map of the 95$^{th}$ percentile summer MDA8 O$_3$ trends over 1995-2014 from AM3 BASE. **(b)** Comparison of median O$_3$ trends from AM3 (1995-2014) with observations (see text for periods): in Central Eastern China at Mt. Tai (July-August, Sun et al. 2016), Beijing (May-June-July, Ding et al. 2008) and Shangdianzi (SDZ) (JJA, Ma et al. 2016); in South China at Hong Kong (HK) (annual average, Wang et al. 2009) and Taiwan (MAM, Lin YK et al. 2010); at Mt. Waliguan (WLG) in western China (MAM, Xu et al. 2016); at South Korea (JJA, Lee et al. 2014) and Mt. Happo Japan (MAM, Tanimoto 2009). For Mt. Happo (triangle on map) AM3 is sampled at 700 hPa and filtered for the influence from Asian continental air - more representative of observed baseline conditions in spring.

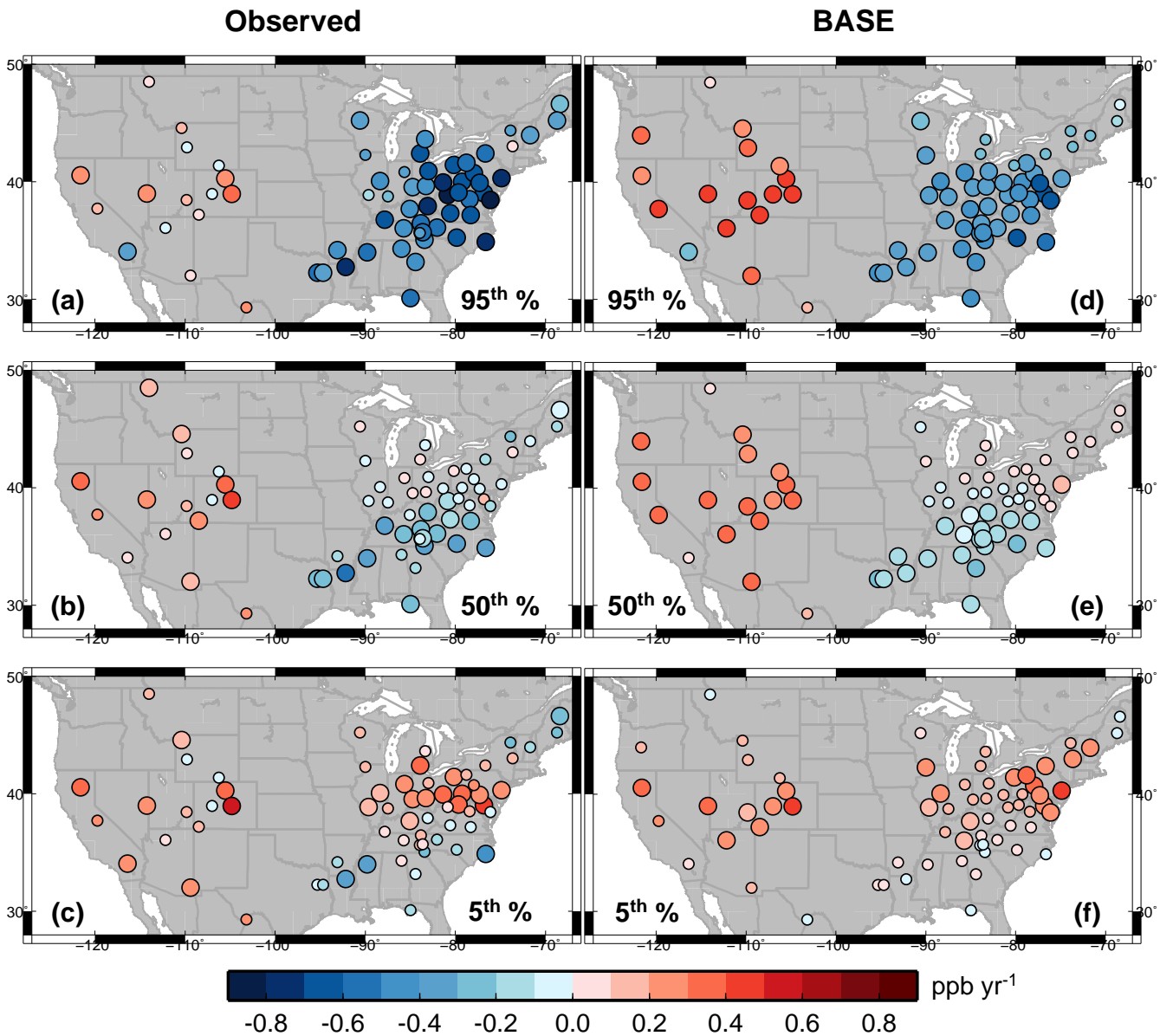

Figure 7. Linear trends in spring (MAM) MDA8 O$_3$ over 1988-2014 at US rural sites for the 95th, 50th, and 5th percentiles as observed (left) and simulated (right) in AM3 BASE. Larger circles indicate sites with statistically significant trends (p<0.05). For WUS high-elevation sites, the model is sampled at 700 hPa and filtered to remove local influence (see text in Sect. 2.4).

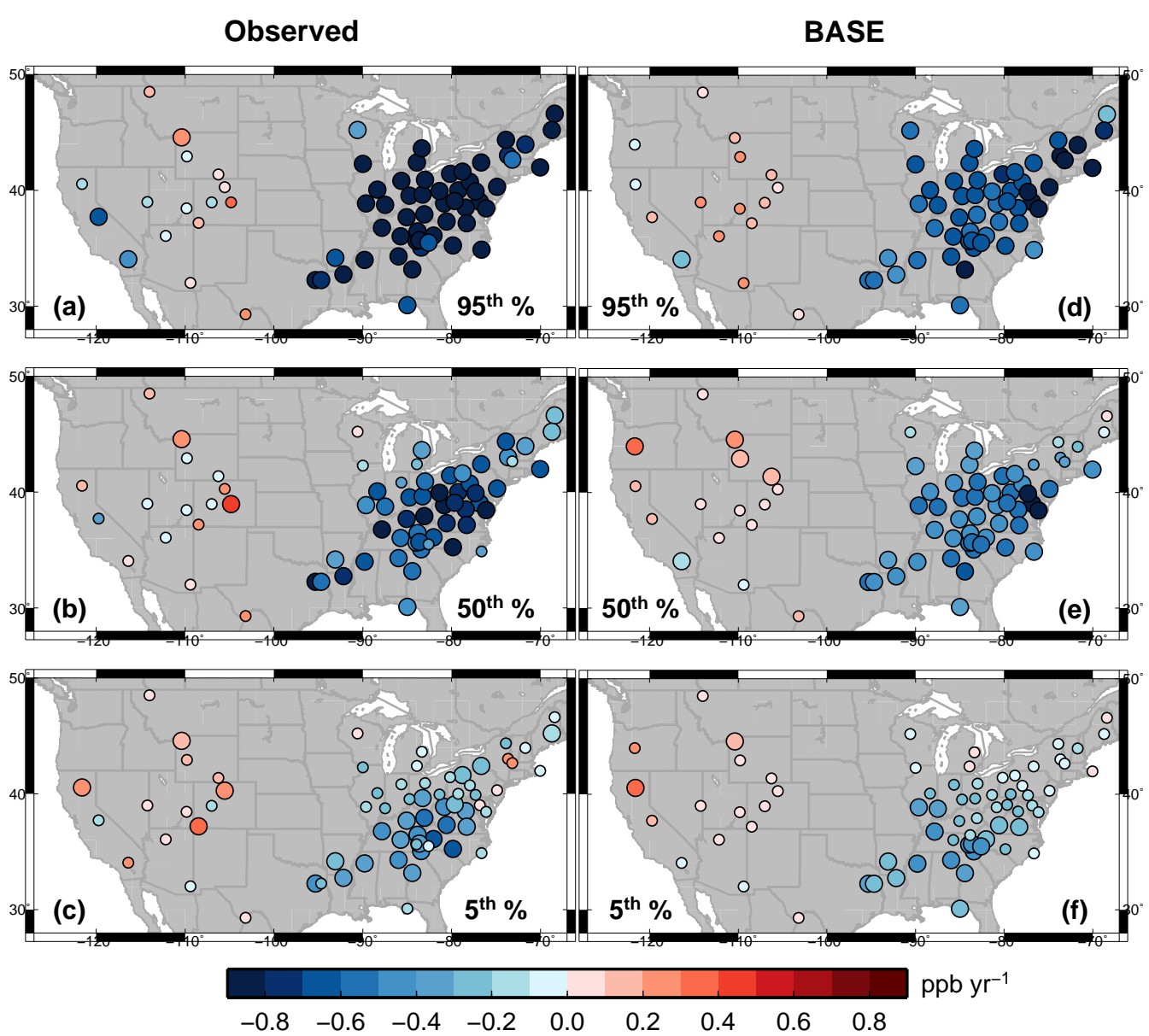

Figure 8. As in Figure 7, but for summer (JJA). Note that the color scale saturates at ±0.8.

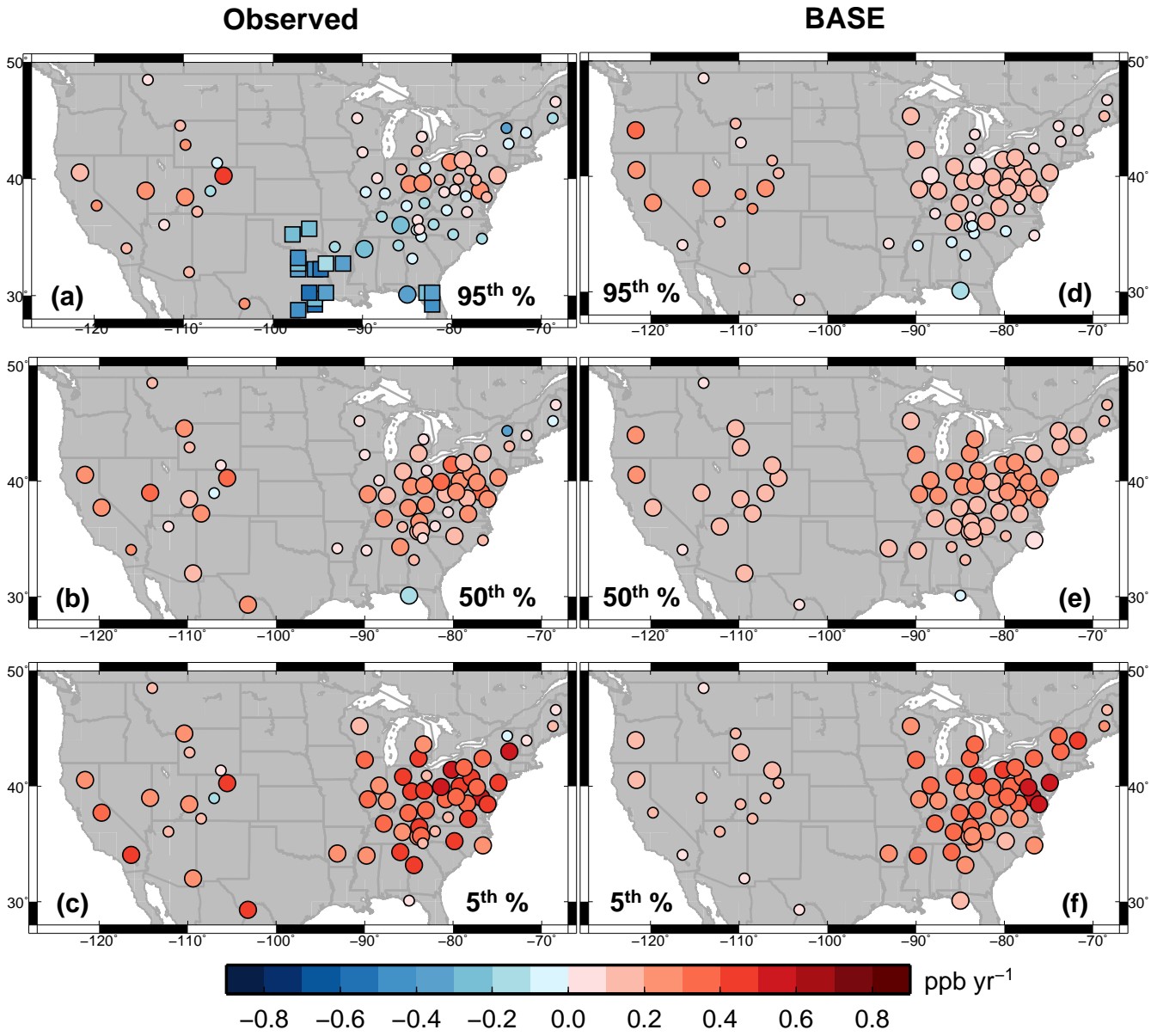

Figure 9. As in Figure 7, but for winter (DJF). Large squares in **(a)** denote AQS sites with significant O$_3$ decreases.

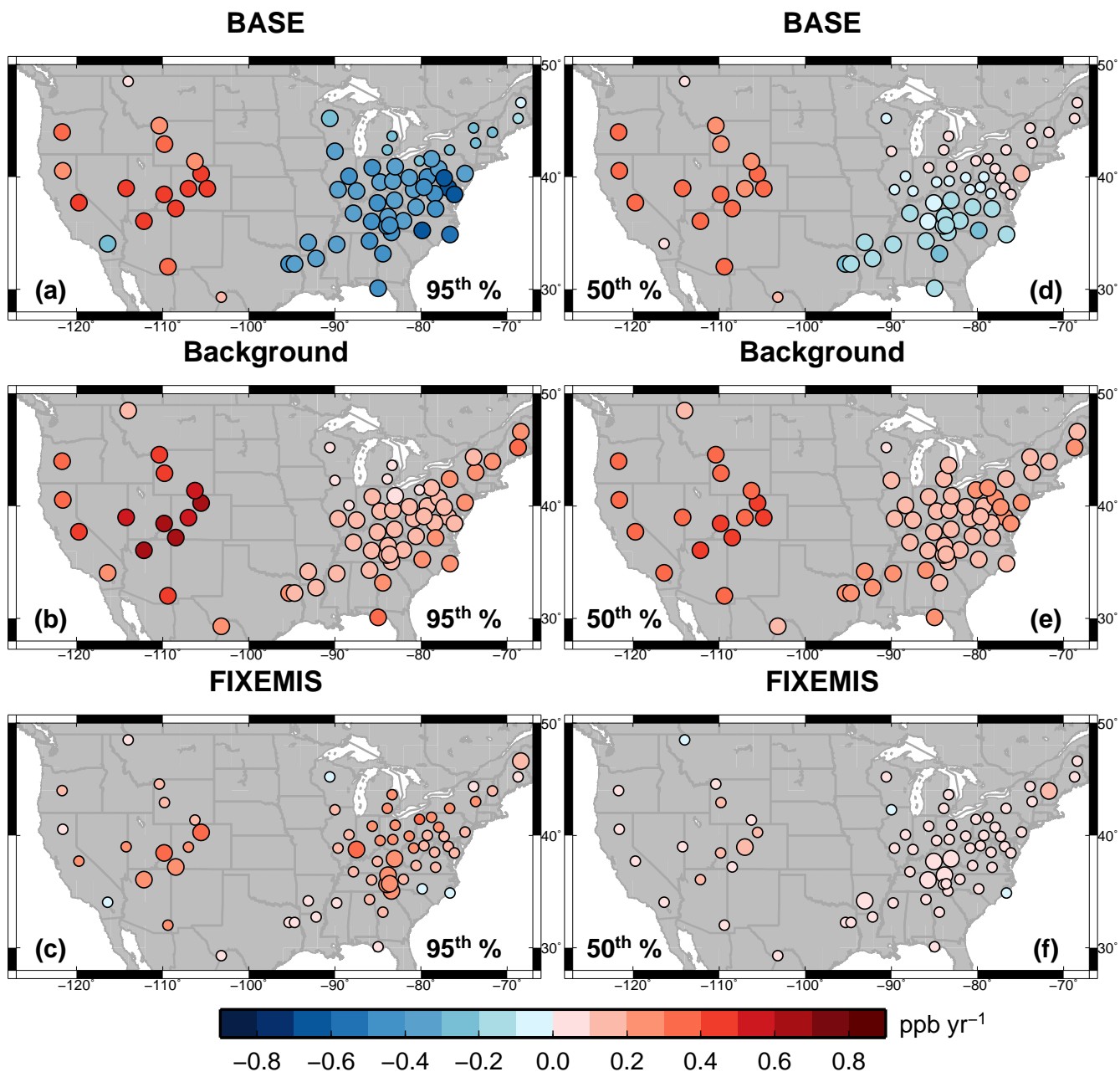

Figure 10. Linear trends in the 95th (left) and 50th (right) percentile springtime MDA8 O₃ over 1988-2014 at US rural sites from BASE (top), Background (middle) and FIXEMIS simulations (bottom). Larger circles indicate sites with statistically significant trends (p<0.05). Top panels are repeated from Fig.7d,e. Note that the 95th (50th) percentile is sampled separately from the Background and FIXEMIS simulations without depending on the times when the BASE simulation is experiencing the 95th (50th) percentile days.

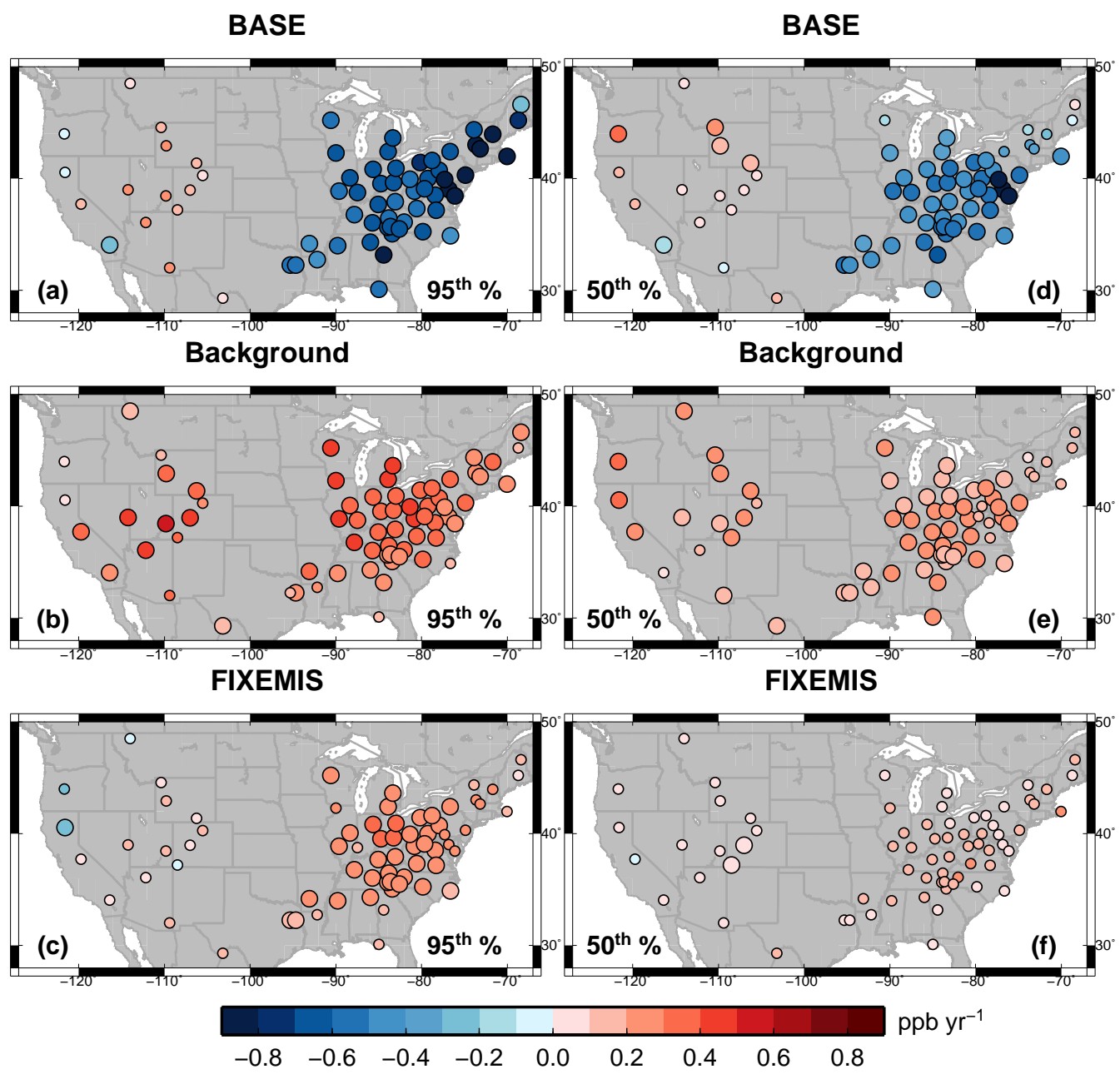

Figure 11. As in Figure 10, but for summer. Top panels are repeated from Fig. 8d,e.

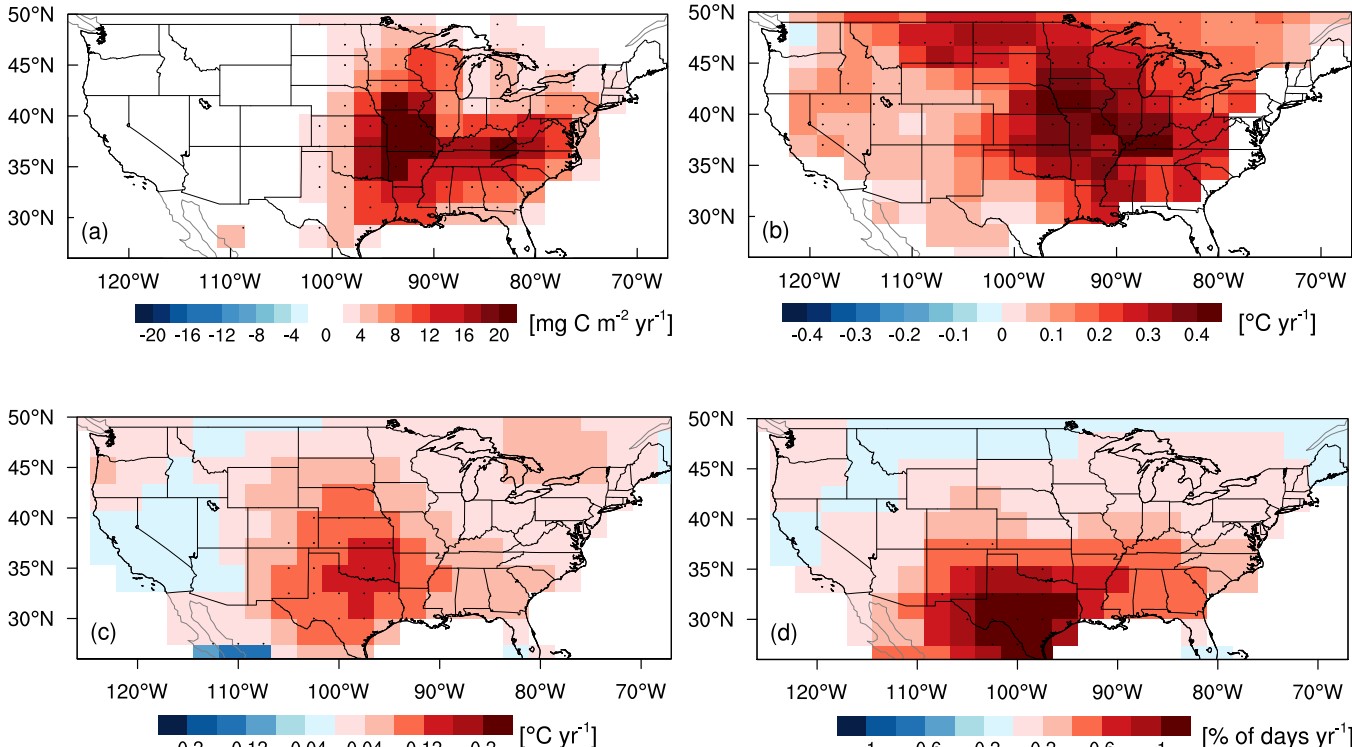

Figure 12. The 1990-2012 trends in: (a) model JJA total biogenic isoprene emissions; (b) model $90^{th}$ percentile JJA daily maximum temperature; (c) the warmest daily maximum temperature and (d) the frequency of warm days (i.e., those above the $90^{th}$ percentile for the base period 1961-90) for August obtained from GHCNDEX dataset (Donat et al., 2013; available at http://www.climdex.org/view_download.html). Stippling denotes areas where the change is statistically significant ($p<0.05$). Note that the trends are calculated for the 1990-2012 period, instead of 1988-2014, to avoid the influence from hot extremes in 1988 and cold conditions in 2014 (Sect. 6). When these years are included, the trends in (c) and (d) are swamped by the anomalies. The trends in (a) and (b) are similar between 1990-2012 and 1988-2014.

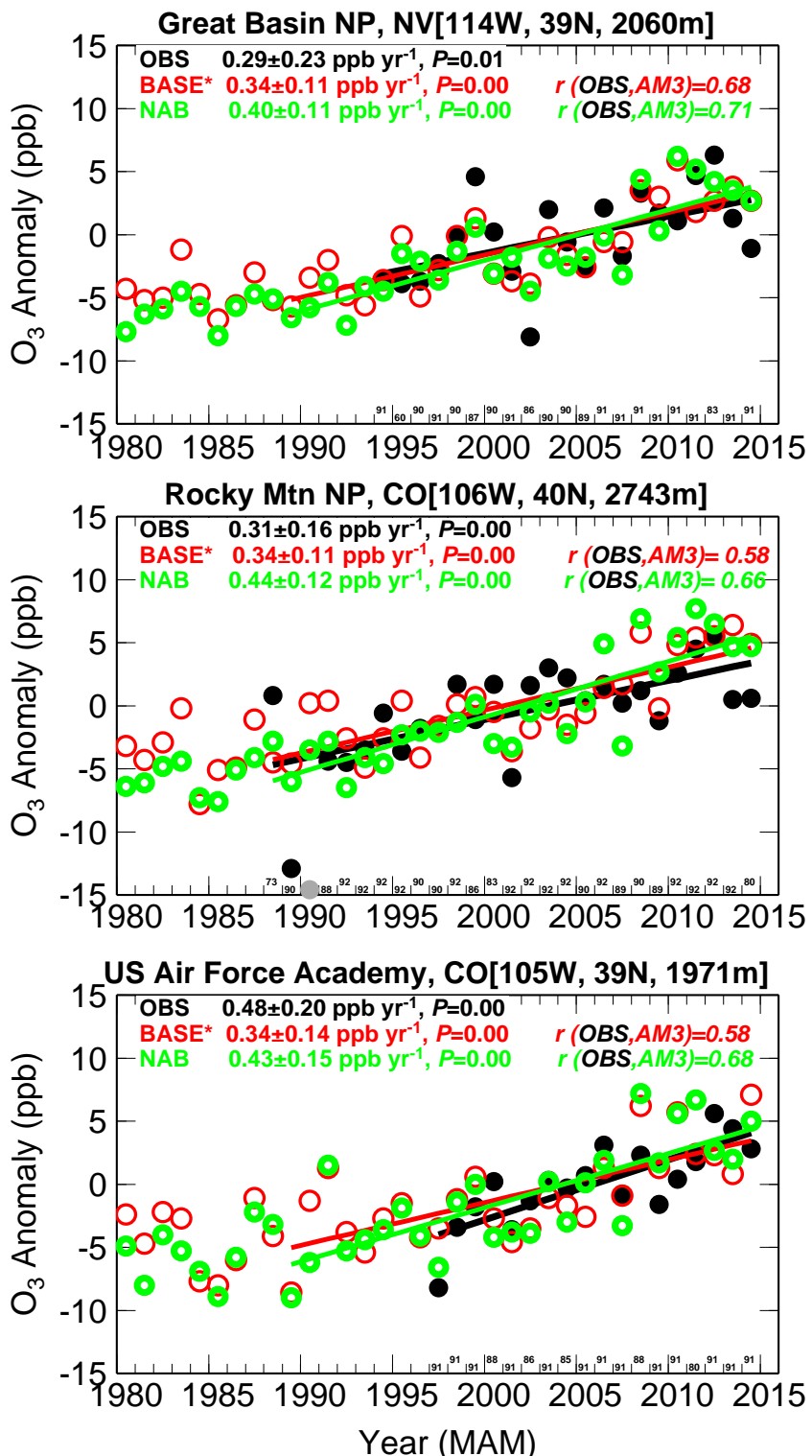

Figure 13a. Time series of median spring MDA8 $O_3$ anomalies (relative to the 1995-2014 mean) at Great Basin, Rocky Mountain, and US Air Force Academy as observed (black) and simulated in AM3 BASE filtered for baseline conditions (red, see Sect.2.4) and in Background with North American anthropogenic emissions zeroed out (NAB; green). Presented on the top of the graph are statistics from the linear fit and correlations between observations and simulations. Numbers on the bottom of the graph denote the sample size of observations for each year. Grey dots indicate uncertain observations that are removed from the linear fit (see Sect. 2.3).

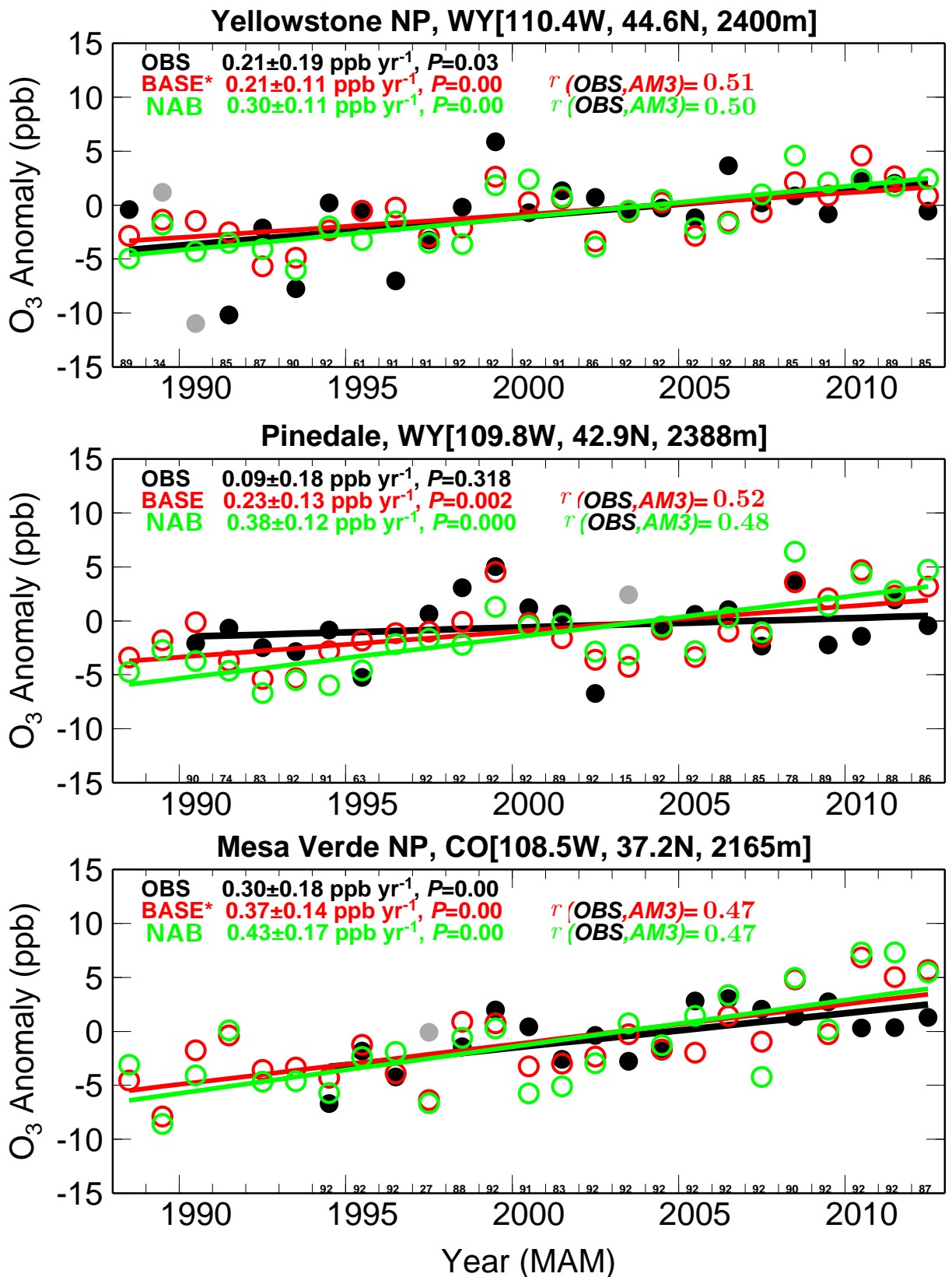

Figure 13b. Same as Figure 13a, but for Yellowstone, Pinedale, and Mesa Verde over the period 1988-2012.

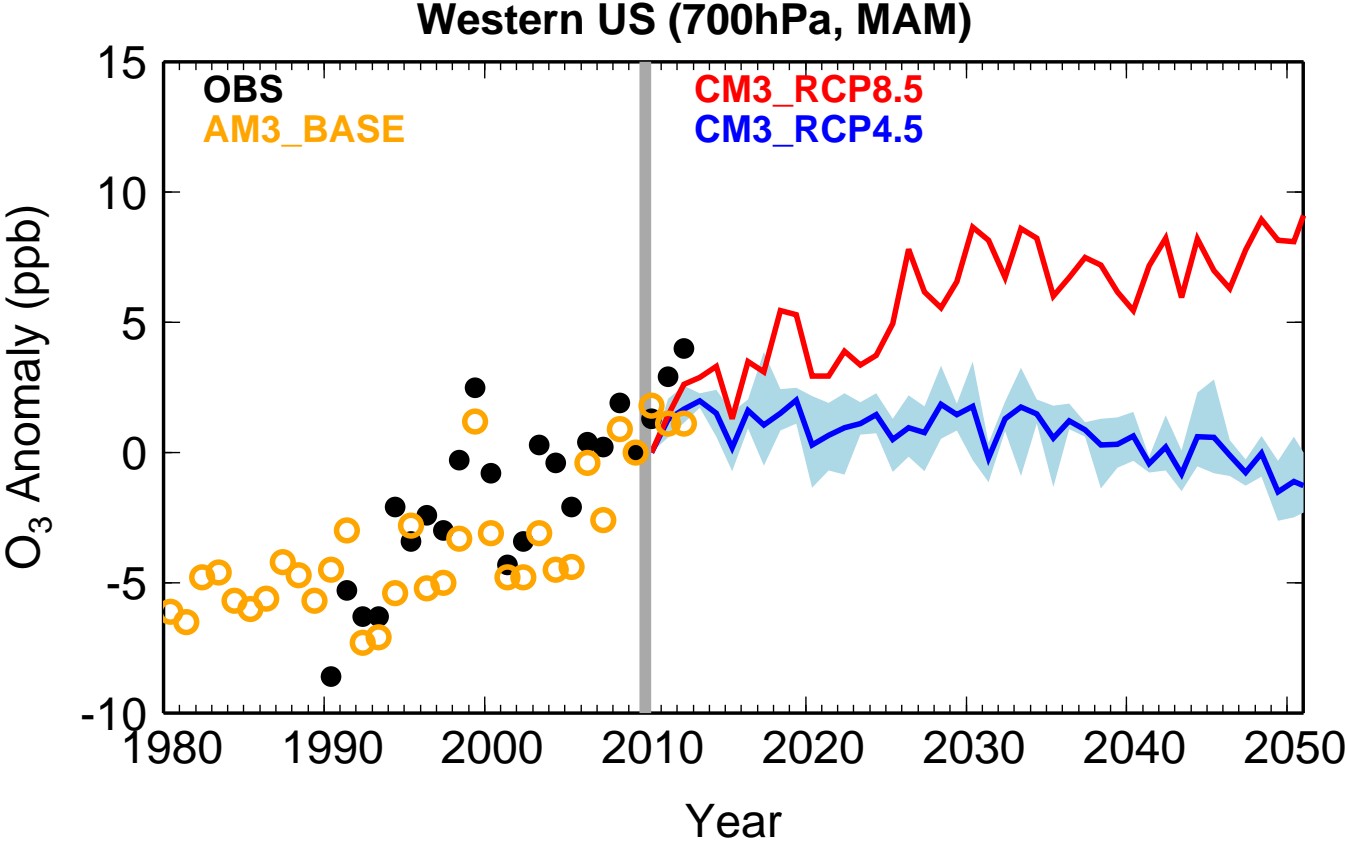

Figure 14. **Future projections.** Time series of median springtime $O_3$ changes relative to 2010 in GFDL AM3 hindcast (orange circles) and CM3 future simulations for RCP8.5 (red) versus RCP4.5 (blue; shading represents the range of three ensemble members), sampled at 700 hPa over the WUS (35-45N,120-105W). Black circles indicate observed changes averaged from Lassen, Great Basin, and Rocky Mountain National Parks.

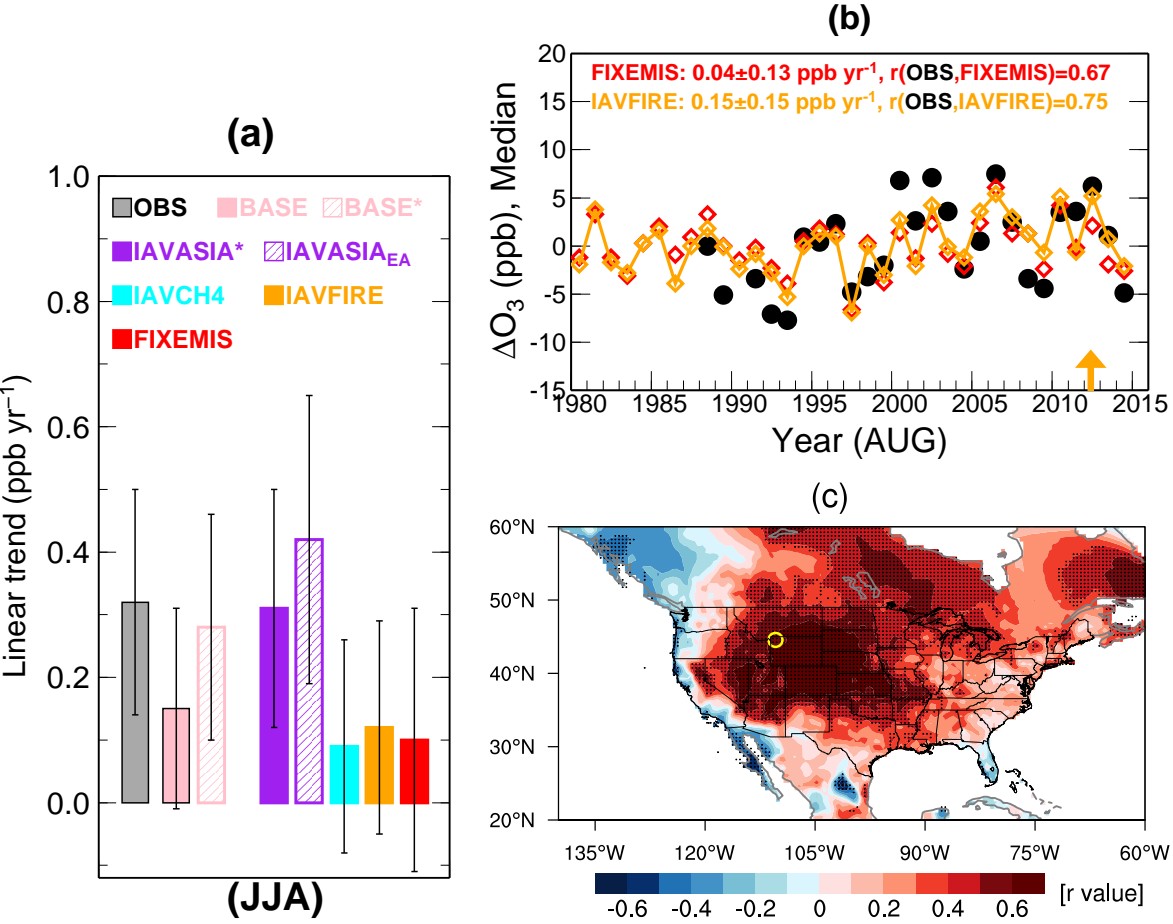

Figure 15. **Summertime O$_3$ in Yellowstone National Park.** (a) Median JJA MDA8 O$_3$ trends over 1988-2012 at Yellowstone from observations (black) and simulations sampled at 700 hPa for BASE without filtering (pink), BASE filtered for baseline conditions (hatched pink), IAVASIA (solid purple, baseline), IAVASIA filtered for Asian influence (EACOt≥67th, hatched purple), IAVCH$_4$ (cyan), IAVFIRE (orange) and FIXEMIS (red). (b) Time series of anomalies in August median MDA8 O$_3$ at Yellowstone as observed (black) and simulated by the model sampled at the surface, with constant (red) and time-varying wildfire emissions (orange). Trends over 1988-2014 are reported. (c) Interannual correlations of JJA mean MDA8 O$_3$ observed at Yellowstone with JJA mean daily maximum temperature from observations (Harris et al., 2014).

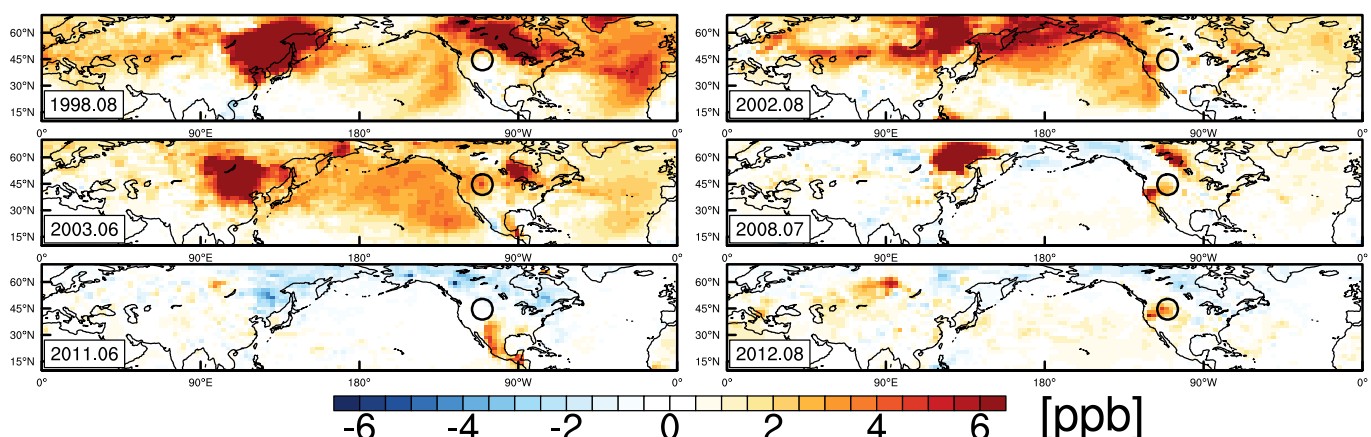

**IAVFIRE - FIXEMIS: Surface MDA8 O3 Anomaly**

Figure 16. Surface MDA8 $O_3$ enhancements from wildfire emissions for individual months in the years with large biomass burning in boreal regions (1998, 2002, 2003) and over the WUS (2008, 2011, 2012), as diagnosed by the differences between IAVFIRE and FIXEMIS. The black circle denotes the location of Yellowstone National Park.

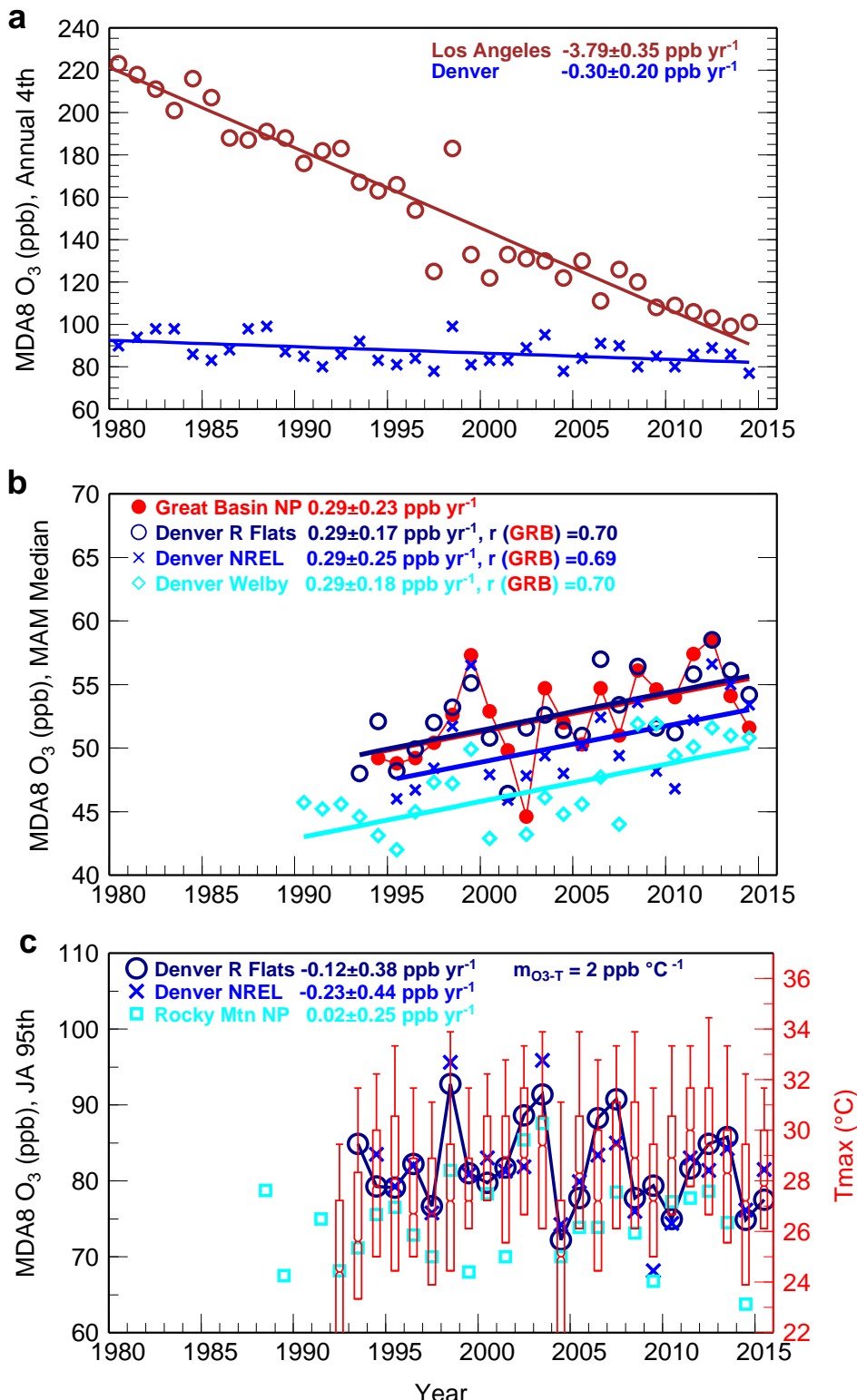

Figure 17. **Surface O$_3$ trends in Denver. (a)** Comparison of observed trends in annual 4$^{th}$ highest MDA8 O$_3$ at Crestline Los Angeles (brown) and in Denver (blue, computed from all monitors available in Denver non-attainment counties). **(b)** Time series of observed median MAM MDA8 O$_3$ at Great Basin National Park (red), in comparison with three monitors in Denver. **(c)** Time series of observed 95$^{th}$ percentile July-August MDA8 O$_3$ in Denver, together with statistics (25th, 50th, 75th, 95th) of observed July-August daily maximum temperature at Rocky Flats (red, right axis).

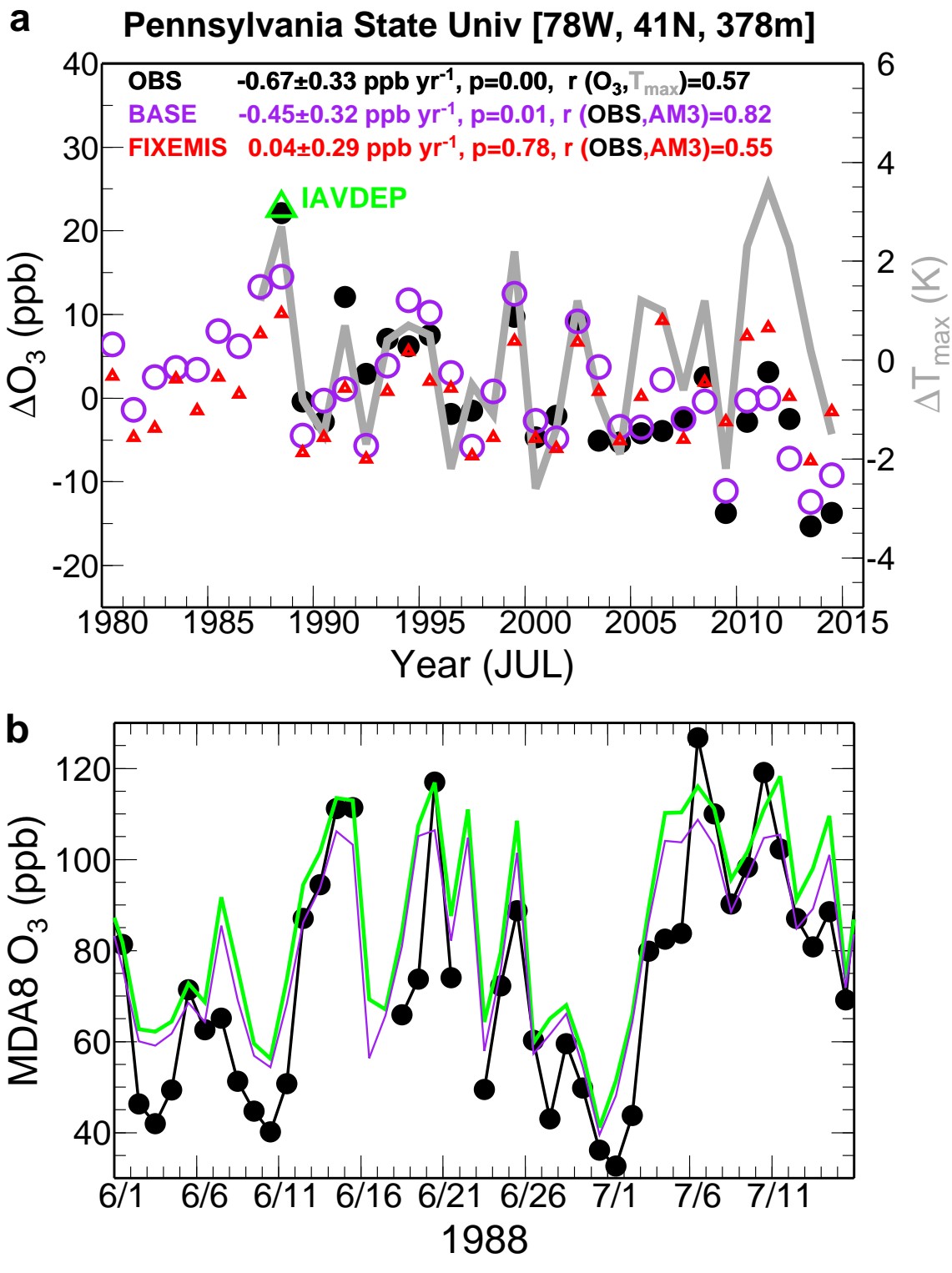

Figure 18. **(a)** Time series of July mean MDA8 $O_3$ anomalies (relative to 1988-2014) at the Pennsylvania State University (PSU) CASTNET site as observed (black) and simulated by the GFDL-AM3 model with time-varying (purple) and constant anthropogenic emissions (red), along with observed anomalies in July mean daily max temperature (gray lines; right axis). The green triangle denotes the 1988 $O_3$ anomaly from a sensitivity simulation using BASE emissions but with 35% decreases in $V_{d,O3}$. **(b)** Time series of daily MDA8 $O_3$ at PSU from June 1 to July 16 in 1988 as observed (black) and simulated by the BASE model (purple) and the sensitivity simulation with 35% decreases in $V_{d,O3}$ (green).

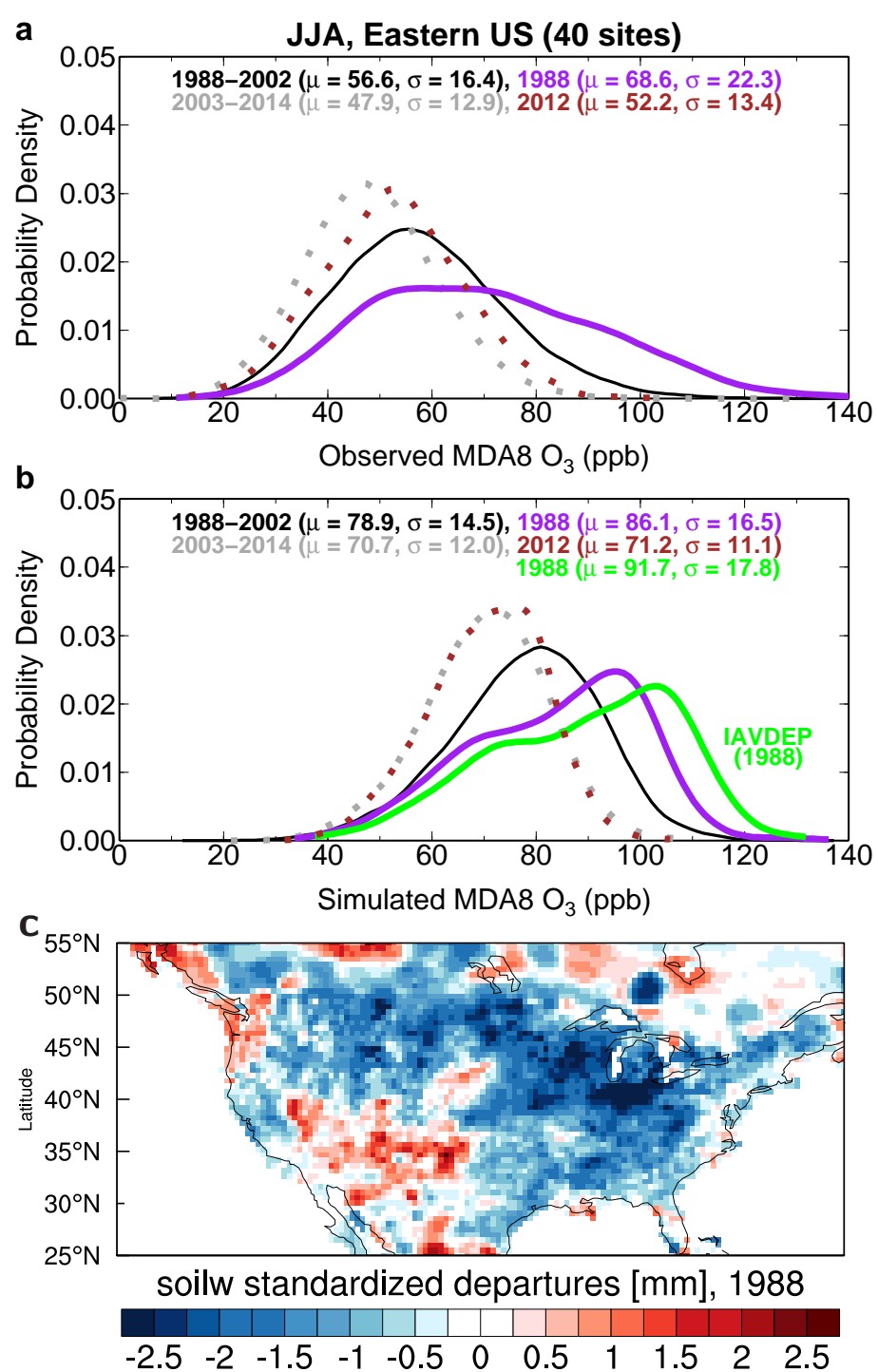

Figure 19. **(a)** Comparisons of probability distributions of summertime MDA8 $O_3$ from 40 EUS CASTNet sites for the pre-$NO_x$ SIP Call (1988-2002; solid black) versus post-$NO_x$ SIP Call (2003-2014; dashed gray) periods and during the extreme heat waves of 1988 (solid purple) versus 2012 (dashed brown). The median ($\mu$) and standard deviation ($\sigma$) are shown (ppb). **(b)** Same as **(a)** but from AM3 BASE. Also shown is the $O_3$ distribution in 1988 from a sensitivity simulation with 35% decreases in $V_{d,O3}$ in drought areas (green). **(c)** Standardized soil moisture departures for JJA 1988 (calculated by dividing anomalies by the 1979-2010 climatological standard deviation, using data from NOAA Climate Prediction Center).

**Figure 20. Summary of US surface O$_3$ trends and drivers.** Changes in decadal mean MDA8 O$_3$ from 1981-1990 to 2003-2012 simulated in a suite of GFDL-AM3 experiments for spring and summer for the western (32N-46N and 123W-102W), Northeast (37N-45N and 90W-65W) and Southeast (30N-36N and 95W-77W) US domains. Observations are not shown because limited data are available during 1981-1990. Experiments are color-coded with the error bars indicating the range of the mean change at the 95% confidence level. Filled circles represent the changes under Background (green) and IAVASIA (purple) when filtered for Asian influence (EACOt$\geq$67$^{th}$%), while other results are from the unfiltered models. The text near the bottom of the plot provides the change in NO$_x$ emissions over the same period for each region.