# Peer review of "US surface ozone trends and extremes from 1980-2014: Quantifying the roles of rising Asian emissions, domestic controls, wildfires, and climate"

_Atmospheric Chemistry and Physics, 2016_

## Referee Comment (RC1) · Anonymous Referee #1 · 12 Dec 2016

The manuscript on US surface ozone trends and extremes by Lin et al. is clearly one of the best modelling studies I have read in my career. It covers an important scientific topic with political relevance and provides an in-depth analysis of US surface ozone and its drivers to the extent that this can be achieved with a global model. It contains a careful and insightful analysis of observations and model results including a well-designed set of sensitivity experiments to attribute ozone trends and variability to various factors. The text is well structured and very well written. All arguments are clearly presented and justified; there is an adequate recognition of previous work. The figures are also very well designed and clear and readable. This would have almost

been the first manuscript which I would recommend to "publish as is", except that I do have a few very minor comments and suggestions how the text could be even further improved. In short, it was a real pleasure to review this manuscript.

Introduction: start with at least one general sentence about ozone being an important air pollutant which has been of relevance to the US for a long time

Page 2, lines 7-10: explicitly mention methane here (part of climate effects?)

Page 2, lines 33/34: this result is based on a previous study with the same model. Don't state it as undisputed fact. Please write "Previous model simulations indicate . . ." or similar.

Page 3, line 2: not only precursor trends, but also inter-annual (meteorological) variability make this difficult if not impossible

Page 3, line 14: you may also want to mention that models have difficulties in simulating the seasonal cycle at baseline sites correctly (see recent papers by Parrish et al., Derwent et al.)

Section 2: please provide at least one general statement about the GFDL model with a reference to the model description paper before describing the experiments.

Page 4, line 22: please provide a reference to the dry deposition climatology

Page 5, line 22: awkward grammar: "a number of studies (Hiboll)."

Page 6, lines 7-10: statement misleading: there are thousands of long-term monitoring sites from AQS and several hundred "rural" stations. Add "selected"?

Page 6, lines 15-17: Please state if trend was derived from daily MDA8 values or monthly values and how you tested for the appropriateness of a linear trend model.

Page 9, line 8 vs. Caption Figure 6: Lee et al. once cited as 2013, and once as 2014.

Page 10, line 11: Please add a quantitative summary statement how well the Asian

trends are captured. Figure 6 indicates within 10-20%, Mt. Happo is within 37%.

Page 11, lines 9-20: I recall from earlier discussions on USNE surface ozone that a large change occurred around 2001 when NOx scrubbers in power plants were activated. Is this worth mentioning here? Could this have an impact on the observed trends and/or the relation between spring and summer trends?

Page 13, line 4 vs. 20 ff: perhaps the rising isoprene discussion could be merged in one place? It is slightly confusing to see this in two places.

Page 14, lines 11ff: Figure caption (Figure 13) uses "NAB" as abbreviation for "Background" run. This should be made consistent (also the font of "NAB" in the legend differs from the other legend entries).

Page 15, line 1: Does the statement "can explain 50-65%..." assume linear additivity of the factors controlling surface ozone? Would the impacts be the same if you applied linear regression on the differences between the model simulations (instead of subtracting the linear trend estimates from each other)? Perhaps, Table 2 would be easier to digest if the individual contributions were listed (i.e. the differences) instead of the regression results themselves?

Page 15, line 38: please add a note how Asian emissions will decrease after 2030 according to RCP8.5. For example, will they reach year 2000 or year 1990 levels?

Page 16, lines 33/34: "consistent with the seasonality of pollution transport from Asia." Isn't this also the influence of the Asian summer monsoon in July/August which reduces surface ozone over Asia itself?

Page 20, lines 22-27: if possible, the argument about dry deposition influencing the high end of ozone distributions during the 1988 heatwave should be substantiated by an additional (1-year or only summer months) model simulation where dry deposition could be turned off (or reduced).

Page 21, lines 1-2: how about "plume chemistry" as another explanation for the overall

bias? There are strong NOx gradients also in the horizontal, and ozone production efficiency is higher in the medium-NOx range than in the high NOx range.

Conclusions: the conclusions are more a summary than real conclusions. I suggest to shorten this summary of results and instead try to go one step further in assessing the possible consequences of this study. For example: even though methane hasn't played a major role in the past, will it become more important in the future if, as suggested by the RCPs, Asian NOx emissions will decrease again? Or: what do we expect from future NOx emissions in the NEUS? In relation to climate change: could there be a greater role of biogenic VOC and would this lead to more or less severe ozone episodes?

Figure 20: why are the observed trends not included in this figure?

---

## Referee Comment (RC2) · Anonymous Referee #2 · 30 Dec 2016

This paper uses modeling in conjunction with observations to assess the causes of surface ozone trends in the United States, and applies some novel approaches to this important problem. The analysis is robust and the paper is generally well-written. I have listed some specific comments below to improve the clarity of some parts of the text.

Page 1 Line 32: Clarify that this is future springtime O3

Page 1 Lines 34-35: Do you mean that the onset of isoprene emissions is earlier in the Southeast than other regions, or that it became earlier over time?

Section 2.1: What time period is the model run for?

P6 Line 18: Why not adjust for sample autocorrelation?

P6 Line 35: Is it only 1990 that has anomalously low values at some sites, or several of the early years? See, for example, the discussion in Strode et al. [2015].

P7 Line 27: What is the justification for picking 700 hPa?

P7 Line 35: Is BASE the same as AM3_BASE? If so, please use one or the other consistently.

P11 Line 31: Since a number of studies have examined trends for slightly different time periods (for example, Cooper et al [2012]), it would be helpful to summarize here how your results for trends through 2014 compare with those trends, and what effect the inclusion of recent years has on the trends.

P13 Line 23: How does the GHCNDEX relate to the meteorology used to drive the model? Why not calculate the change in max temperature etc. using the same met fields that drive the model?

P20 Line 16: This is a significant bias, and should be discussed earlier in the paper.

Fig. 8 caption: What does "colorbar saturates at -0.8" mean?

---

## Author Comment (AC1) · 21 Jan 2017

**Response to Anonymous Referee #1**

The manuscript on US surface ozone trends and extremes by Lin et al. is clearly one of the best modelling studies I have read in my career. It covers an important scientific topic with political relevance and provides an in-depth analysis of US surface ozone and its drivers to the extent that this can be achieved with a global model. It contains a careful and insightful analysis of observations and model results including a well-designed set of sensitivity experiments to attribute ozone trends and variability to various factors. The text is well structured and very well written. All arguments are clearly presented and justified; there is an adequate recognition of previous work. The figures are also very well designed and clear and readable. This would have almost been the first manuscript which I would recommend to "publish as is", except that I do have a few very minor comments and suggestions how the text could be even further improved. In short, it was a real pleasure to review this manuscript.

**RE: We truly appreciate the reviewer for carefully reading the manuscript and for favorable comments and insightful suggestions. Below we include a point-by-point response (in bold blue) to the reviewer, responding to** _their comments (in italic)_ **and explaining the changes made to the manuscript.**

_Introduction: start with at least one general sentence about ozone being an important air pollutant which has been of relevance to the US for a long time._
**RE: Good suggestion! We now begin with this sentence:**

**"Within the United States, ground-level $O_3$ has been recognized since the 1940s and 1950s as an air pollutant detrimental to public health."**

_Page 2, lines 7-10: explicitly mention methane here (part of climate effects?)_
**RE: Done. "There are concerns that rising Asian emissions and global methane …"**

_Page 2, lines 33/34: this result is based on a previous study with the same model. Don't state it as undisputed fact. Please write "Previous model simulations indicate . . ." or similar._
**RE: We now say:**

**"Model simulations indicate that import of Asian pollution enhances mean WUS surface $O_3$ in spring by ~5 ppb (Zhang et al., 2008; Lin et al., 2012b), and occasionally contributes 8-15 ppb during springtime pollution episodes observed at rural sites (Lin et al., 2012b) as supported by in situ aerosol composition analysis (VanCuren and Gustin, 2015)"**

_Page 3, line 2: not only precursor trends, but also inter-annual (meteorological) vari- ability make this difficult if not impossible_
**RE: Good point! We now say:**

**"Discerning directly the effect of climate change on air quality from long-term observation records of $O_3$ would be ideal, but concurrent trends in precursor emissions and large internal variability in regional climate impede such an effort."**

*Page 3, line 14: you may also want to mention that models have difficulties in simulating the seasonal cycle at baseline sites correctly (see recent papers by Parrish et al., Derwent et al.)*
**RE: We did not make change here because the focus of this paper is on long-term trends. Adding discussions on the seasonal cycle somewhat interrupts the overall flow of that paragraph.**

*Section 2: please provide at least one general statement about the GFDL model with a reference to the model description paper before describing the experiments.*
*Page 4, line 22: please provide a reference to the dry deposition climatology*
**RE: Done. Please see the revised manuscript.**

*Page 5, line 22: awkward grammar: "a number of studies (Hiboll)."*
**RE: Changed to "… by a few recent studies (e.g., *Hilboll et al.*, 2013)"**

*Page 6, lines 7-10: statement misleading: there are thousands of long-term monitoring sites from AQS and several hundred "rural" stations. Add "selected"?*
**RE: Add "selected".**

*Page 6, lines 15-17: Please state if trend was derived from daily MDA8 values or monthly values and how you tested for the appropriateness of a linear trend model.*
**RE: This is clarified in Section 2.3.**

**"The trend is calculated separately for the 5th, 50th and 95th percentiles of daily MDA8 $O_3$ for each season through ordinary linear least-square regression. Statistics are derived for the slope of the linear regression in units of ppb yr-1, the range of the slope with a 95% confidence limit (not adjusted for sample autocorrelation), and the p-value indicating the statistical significance of the trend based on a two-tailed t-test."**

*Page 9, line 8 vs. Caption Figure 6: Lee et al. once cited as 2013, and once as 2014.*
**RE: Nice catch! Revised.**

*Page 10, line 11: Please add a quantitative summary statement how well the Asian trends are captured. Figure 6 indicates within 10-20%, Mt. Happo is within 37%.*
**RE: Good suggestion! We now stated "We conclude that GFDL-AM3 captures 65-90% of the observed $O_3$ increases in Asia, lending confidence in its application to assess the global impacts of rising Asian emissions."**

*Page 11, lines 9-20: I recall from earlier discussions on USNE surface ozone that a large change occurred around 2001 when NOx scrubbers in power plants were activated. Is this worth*

*mentioning here? Could this have an impact on the observed trends and/or the relation between spring and summer trends?*

**RE: We now mention this in the revised manuscript:**

**"Many northeast states in the late 1990s and early 2000s did not turn on power plant NO$_x$ emission controls until the O$_3$ season (May-September), which may also contribute to observed differences between spring and summer O$_3$ trends."**

*Page 13, line 4 vs. 20 ff: perhaps the rising isoprene discussion could be merged in one place? It is slightly confusing to see this in two places.*

**RE: We have moved that sentence down to the next paragraph.**

*Page 14, lines 11ff: Figure caption (Figure 13) uses "NAB" as abbreviation for "Background" run. This should be made consistent (also the font of "NAB" in the legend differs from the other legend entries).*

**RE: The "NAB" abbreviation is only used in Figure 13 because of limited space. We have used the term "Background" throughout the text in the manuscript.**

*Page 15, line 1: Does the statement "can explain 50-65%..." assume linear additivity of the factors controlling surface ozone? Would the impacts be the same if you applied linear regression on the differences between the model simulations (instead of sub- tracting the linear trend estimates from each other)? Perhaps, Table 2 would be easier to digest if the individual contributions were listed (i.e. the differences) instead of the regression results themselves?*

**RE: As suggested by the reviewer, we apply linear regression on the differences between the model simulations and find no significant change from the impacts calculated by subtracting the linear trends in Table 2. Thus, no change is made in the manuscript.**

*Page 15, line 38: please add a note how Asian emissions will decrease after 2030 according to RCP8.5. For example, will they reach year 2000 or year 1990 levels?*

**RE: Done.**

**"Under the RCP8.5 scenario, Chinese NO$_x$ emissions are projected to peak in 2020-2030, reflecting an increase of ~50% from 2010 (Fig.1a), followed by a sharp decrease reaching 1990 levels by 2050."**

*Page 16, lines 33/34: "consistent with the seasonality of pollution transport from Asia." Isn't this also the influence of the Asian summer monsoon in July/August which reduces surface ozone over Asia itself?*

**RE: We now say:**

**"The stronger increase measured in June than in July-August is consistent with the Asian summer monsoon producing surface O$_3$ minimum in July-August (e.g., Lin et al., 2009), as well as the seasonality of intercontinental pollution transport."**

*Page 20, lines 22-27: if possible, the argument about dry deposition influencing the high end of ozone distributions during the 1988 heatwave should be substantiated by an additional (1-year or only summer months) model simulation where dry deposition could be turned off (or reduced).*

**RE: Thanks for the suggestion! We have conducted a sensitivity simulation for 1988 with 35% decreases in $O_3$ deposition velocities over drought areas. The results are shown in Figs. 18 and 19 (copied below) and discussed in Section 6 (please see tracked changes in the revised manuscript).**

[Figure]

Figure 18. (a) Time series of July mean MDA8 $O_3$ anomalies (relative to 1988-2014) at the Pennsylvania State University (PSU) CASTNET site as observed (black) and simulated by the GFDL-AM3 model with time-varying (purple) and constant anthropogenic emissions (red), along with observed anomalies in July mean daily max temperature (gray lines; right axis). The green triangle denotes the 1988 $O_3$ anomaly from a sensitivity simulation using BASE emissions but with 35% decreases in $V_{d,O3}$. (b) Time series of daily MDA8 $O_3$ at PSU from June 1 to July 16 in 1988 as observed (black) and simulated by the BASE model (purple) and the sensitivity simulation with 35% decreases in $V_{d,O3}$ (green).

[Figure]

Figure 19. (a) Comparisons of probability distributions of summertime MDA8 O$_3$ from 40 EUS CASTNet sites for the pre-NO$_x$ SIP Call (1988-2002; solid black) versus post-NO$_x$ SIP Call (2003-2014; dashed gray) periods and during the extreme heat waves of 1988 (solid purple) versus 2012 (dashed brown). The median ($\mu$) and standard deviation ($\sigma$) are shown (ppb). (b) Same as (a) but from AM3 BASE. Also shown is the O$_3$ distribution in 1988 from a sensitivity simulation with 35% decreases in V$_{d,O3}$ in drought areas (green). (c) Standardized soil moisture departures for JJA 1988 (calculated by dividing anomalies by the 1979-2010 climatological standard deviation, using data from NOAA Climate Prediction Center).

*Page 21, lines 1-2: how about "plume chemistry" as another explanation for the overall bias? There are strong NO$_x$ gradients also in the horizontal, and ozone production efficiency is higher in the medium-NO$_x$ range than in the high NO$_x$ range.*

**RE: We don't think model limitation in resolving plume chemistry is a major explanation for the bias. Travis et al. (2016) used a 0.25°x0.25° model and found**

**20 ppb biases similar to our 2-degree model before adjustment of NO$_x$ emissions. No changes are made in the manuscript.**

*Figure 20: why are the observed trends not included in this figure?*
**RE: Because this figure shows decadal mean changes from 1981-1990 to 2003-2013. There are only limited observations available during 1981-1990. We have clarified this in the caption of Fig.20.**

*Conclusions: the conclusions are more a summary than real conclusions. I suggest to shorten this summary of results and instead try to go one step further in assessing the possible consequences of this study. For example: even though methane hasn't played a major role in the past, will it become more important in the future if, as suggested by the RCPs, Asian NOx emissions will decrease again? Or: what do we expect from future NOx emissions in the NEUS? In relation to climate change: could there be a greater role of biogenic VOC and would this lead to more or less severe ozone episodes?*

**RE: Good suggestion. We have changed the title of Section 7 to "Conclusion and Recommendations". Now we explicitly discuss the implications of our work; (1) on the common model biases on baseline ozone trends and recommendations for future multi-model analysis for the Chemistry-Climate Model Initiative, (2) the growing importance of rising global methane and NOx emissions in South Asian countries, where ozone production is more efficient, after Chinese emissions continue to decline in the coming decades, (3) the benefits of future NO$_x$ emission controls on O$_3$ reductions in the Southeast US, and (4) uncertainty in model treatment of land-biosphere couplings for projecting pollution extremes in a warming climate. Please see tracked changes in Section 7 of the revised manuscript.**

---

## Author Response (AR1)

**Dear Dr. Duncan (co-Editor),**

**We thank the two anonymous reviewers for helpful comments on the manuscript. We have revised the manuscript to reflect their suggestions. Below we include a point-by-point response (in bold blue) to the reviewers, responding to *their comments (in italic)* and explaining the changes made to the manuscript.**

**Best regards,**
**Meiyun Lin**
**(on behalf of the authors)**

**Response to Anonymous Referee #1**

The manuscript on US surface ozone trends and extremes by Lin et al. is clearly one of the best modelling studies I have read in my career. It covers an important scientific topic with political relevance and provides an in-depth analysis of US surface ozone and its drivers to the extent that this can be achieved with a global model. It contains a careful and insightful analysis of observations and model results including a well-designed set of sensitivity experiments to attribute ozone trends and variability to various factors. The text is well structured and very well written. All arguments are clearly presented and justified; there is an adequate recognition of previous work. The figures are also very well designed and clear and readable. This would have almost been the first manuscript which I would recommend to "publish as is", except that I do have a few very minor comments and suggestions how the text could be even further improved. In short, it was a real pleasure to review this manuscript.

**RE: We truly appreciate the reviewer for carefully reading the manuscript and for favorable comments and insightful suggestions.**

*Introduction: start with at least one general sentence about ozone being an important air pollutant which has been of relevance to the US for a long time.*
**RE: Good suggestion! We now begin with this sentence:**

**"Within the United States, ground-level $O_3$ has been recognized since the 1940s and 1950s as an air pollutant detrimental to public health."**

*Page 2, lines 7-10: explicitly mention methane here (part of climate effects?)*
**RE: Done. "There are concerns that rising Asian emissions and global methane …"**

*Page 2, lines 33/34: this result is based on a previous study with the same model. Don't state it as undisputed fact. Please write "Previous model simulations indicate . . ." or similar.*
**RE: We now say:**

**"Previous model simulations indicate that import of Asian pollution enhances mean WUS surface $O_3$ in spring by ~5 ppb (Zhang et al., 2008; Lin et al., 2012b),**

**and occasionally contributes 8-15 ppb during springtime pollution episodes observed at rural sites (Lin et al., 2012b)."**

*Page 3, line 2: not only precursor trends, but also inter-annual (meteorological) vari- ability make this difficult if not impossible*
**RE: Good point! We now say:**

**"Discerning directly the effect of climate change on air quality from long-term observation records of O3 would be ideal, but concurrent trends in precursor emissions and large internal climate variability on regional scales impede such an effort."**

*Page 3, line 14: you may also want to mention that models have difficulties in simulating the seasonal cycle at baseline sites correctly (see recent papers by Parrish et al., Derwent et al.)*
**RE: We did not make change here because the focus of this paper is on long-term trends. Adding discussions on the seasonal cycle somewhat interrupt the overall flow of that paragraph.**

*Section 2: please provide at least one general statement about the GFDL model with a reference to the model description paper before describing the experiments.*
*Page 4, line 22: please provide a reference to the dry deposition climatology*
**RE: Done. We slightly reorganize the first paragraph of Section 2 and include additional information on a sensitivity simulation for 1988 with decreased O3 dry deposition velocities due to droughts simulated by the GFDL Land Model 3 (see also Section 6, Figures 18 and 19).**

*Page 5, line 22: awkward grammar: "a number of studies (Hiboll)."*
**RE: Changed to "… by a few recent studies (e.g., *Hilboll et al.*, 2013)"**

*Page 6, lines 7-10: statement misleading: there are thousands of long-term monitoring sites from AQS and several hundred "rural" stations. Add "selected"?*
**RE: Add "selected".**

*Page 6, lines 15-17: Please state if trend was derived from daily MDA8 values or monthly values and how you tested for the appropriateness of a linear trend model.*
**RE: This is clarified in Section 2.3.**

**"The trend is calculated separately for the 5th, 50th and 95th percentiles of daily MDA8 O3 for each season through ordinary linear least-square regression. Statistics are derived for the slope of the linear regression in units of ppb yr-1, the range of the slope with a 95% confidence limit (not adjusted for sample autocorrelation), and the p-value indicating the statistical significance of the trend based on a two-tailed t test."**

*Page 9, line 8 vs. Caption Figure 6: Lee et al. once cited as 2013, and once as 2014.*
**RE: Nice catch! Revised.**

*Page 10, line 11: Please add a quantitative summary statement how well the Asian trends are captured. Figure 6 indicates within 10-20%, Mt. Happo is within 37%.*
**RE: Good suggestion! We now stated "We conclude that GFDL-AM3 captures 65-90% of the observed $O_3$ increases in Asia, lending confidence in its application to assess the global impacts of rising Asian emissions."**

*Page 11, lines 9-20: I recall from earlier discussions on USNE surface ozone that a large change occurred around 2001 when NOx scrubbers in power plants were activated. Is this worth mentioning here? Could this have an impact on the observed trends and/or the relation between spring and summer trends?*
**RE: We now mention this in the revised manuscript:**

**"Many northeast states in the late 1990s and early 2000s did not turn on power plant $NO_x$ emission controls until the $O_3$ season (May-September), which may also contribute to observed differences between spring and summer $O_3$ trends."**

*Page 13, line 4 vs. 20 ff: perhaps the rising isoprene discussion could be merged in one place? It is slightly confusing to see this in two places.*
**RE: We have moved that sentence down to the next paragraph.**

*Page 14, lines 11ff: Figure caption (Figure 13) uses "NAB" as abbreviation for "Background" run. This should be made consistent (also the font of "NAB" in the legend differs from the other legend entries).*
**RE: The "NAB" abbreviation is only used in Figure 13 because of limited space. We have used the term "Background" throughout the text in the manuscript.**

*Page 15, line 1: Does the statement "can explain 50-65%..." assume linear additivity of the factors controlling surface ozone? Would the impacts be the same if you applied linear regression on the differences between the model simulations (instead of sub- tracting the linear trend estimates from each other)? Perhaps, Table 2 would be easier to digest if the individual contributions were listed (i.e. the differences) instead of the regression results themselves?*
**RE: As suggested by the reviewer, we apply linear regression on the differences between the model simulations and find no significant change from the impacts calculated by subtracting the linear trends in Table 2. Thus, no change is made in the manuscript.**

*Page 15, line 38: please add a note how Asian emissions will decrease after 2030 according to RCP8.5. For example, will they reach year 2000 or year 1990 levels?*
**RE: Done.**

**"Under the RCP8.5 scenario, Chinese NO$_x$ emissions are projected to peak in 2020-2030, reflecting an increase of ~50% from 2010 (Fig.1a), followed by a sharp decrease reaching 1990 levels by 2050."**

*Page 16, lines 33/34: "consistent with the seasonality of pollution transport from Asia." Isn't this also the influence of the Asian summer monsoon in July/August which reduces surface ozone over Asia itself?*
**RE: We now say:**

**"The stronger increase measured in June than in July-August is consistent with the influence of the Asian summer monsoon producing surface O$_3$ minimum in July-August in Asia (e.g., Lin et al., 2009), as well as the seasonality of intercontinental pollution transport."**

*Page 20, lines 22-27: if possible, the argument about dry deposition influencing the high end of ozone distributions during the 1988 heatwave should be substantiated by an additional (1-year or only summer months) model simulation where dry deposition could be turned off (or reduced).*

**RE: Thanks for the suggestion! We have conducted a sensitivity simulation for 1988 with reduced O$_3$ deposition velocities as simulated by the GFDL Land Model 3 driven by reanalysis meteorology. This simulation is briefly described in Section 2.1. The results are shown in Figs. 18 and 19 and discussed in Section 6 (please see tracked changes in the revised manuscript).**

*Page 21, lines 1-2: how about "plume chemistry" as another explanation for the overall bias? There are strong NO$_x$ gradients also in the horizontal, and ozone production efficiency is higher in the medium-NO$_x$ range than in the high NO$_x$ range.*
**RE: We don't think model limitation in resolving plume chemistry is a major explanation for the bias. Travis et al. (2016) used a 0.25°x0.25° model and found 20 ppb biases similar to our 2-degree model before adjustment of NO$_x$ emissions. No changes are made in the manuscript.**

*Figure 20: why are the observed trends not included in this figure?*
**RE: Because this figure shows decadal mean changes from 1981-1990 to 2003-2013. There are only limited observations available during 1981-1990. We have clarified this in the caption of Fig.20.**

*Conclusions: the conclusions are more a summary than real conclusions. I suggest to shorten this summary of results and instead try to go one step further in assessing the possible consequences of this study. For example: even though methane hasn't played a major role in the past, will it become more important in the future if, as suggested by the RCPs, Asian NOx emissions will decrease again? Or: what do we expect from future NOx emissions in the NEUS? In relation to climate change: could there be a greater role of biogenic VOC and would this lead to more or less severe ozone episodes?*

RE: Good suggestion. We have changed the title of Section 7 to "Conclusion and Recommendations". Now we explicitly discuss the implications of this study: (1) recommendations for future multi-model analysis for the IGAC/Chemistry-Climate Model Initiative (CCMI), (2) the growing importance of rising global methane and NOx emissions in the tropical Asian countries, where ozone production is more efficient, in the coming decades, (3) the benefits of future NOx emission controls on $O_3$ reductions in the Southeast US, and (4) uncertainties in the model treatment of land-biosphere couplings and their impacts on pollution extremes in a warming climate. Please see tracked changes in Section 7 of the revised manuscript.

**Response to Anonymous Referee #2**

**We thank the reviewer for positive comments on the manuscript. Below we include a point-by-point response (in bold blue) to the reviewer, responding to** _their comments (in italic)_ **and explaining the changes made to the manuscript.**

_This paper uses modeling in conjunction with observations to assess the causes of surface ozone trends in the United States, and applies some novel approaches to this important problem. The analysis is robust and the paper is generally well-written. I have listed some specific comments below to improve the clarity of some parts of the text._

_Page 1 Line 32: Clarify that this is future springtime O3_

**RE: Changed to: "Mean springtime $O_3$ above the WUS is projected to increase by ~10 ppb from 2010 to 2030 under the RCP8.5 global change scenario."**

_Page 1 Lines 34-35: Do you mean that the onset of isoprene emissions is earlier in the Southeast than other regions, or that it became earlier over time?_

**RE: We now say "The $O_3$ decreases driven by NOx emission controls were most pronounced in the Southeast, where the** _seasonal onset_ **of biogenic isoprene emissions and $NO_x$-sensitive O3 production occurs earlier than in the Northeast."**

_Section 2.1: What time period is the model run for?_
**RE: please see Table 1.**

_P6 Line 18: Why not adjust for sample autocorrelation?_
**RE:  We do not adjust the confidence limit for sample autocorrelation to enable a directly comparison with the trends reported in the published literature (e.g., Cooper et al., 2012; Parrish et al., 2014).**

_P6 Line 35: Is it only 1990 that has anomalously low values at some sites, or several of the early years? See, for example, the discussion in Strode et al. [2015]._
**RE: Here we are talking about the cross-site consistency on the anomalies. The other years, such as 1992-1993, also have low-anomalies, but they are consistent across the sites and reflect the influence from the Mount Pinatubo eruption as discussed in Lin et al. (Nature Communications, 2015).**

_P7 Line 27: What is the justification for picking 700 hPa?_
**RE: The level is representative of free tropospheric air since we want to limit the excessive influence from surface pollution in the model.**

_P7 Line 35: Is BASE the same as AM3_BASE? If so, please use one or the other consistently._
**RE: Yes, they are the same. We have avoided using AM3_BASE in the revised manuscript.**

*P11 Line 31: Since a number of studies have examined trends for slightly different time periods (for example, Cooper et al [2012]), it would be helpful to summarize here how your results for trends through 2014 compare with those trends, and what effect the inclusion of recent years has on the trends.*

**RE: Good suggestion!! We now include additional discussions in Section 4.1 regarding the difference in the trends reported in this work compared to prior studies. Copied below:**

**"The north-to-south gradient in springtime $O_3$ trends over the EUS reflects the earlier seasonal transition from $NO_x$-saturated to $NO_x$-sensitive $O_3$ production regimes in the Southeast, where plentiful radiation in spring enhances $HO_x$ supply and biogenic isoprene emissions are turned on earlier than the Northeast. The different response of springtime $O_3$ to $NO_x$ controls in the Southeast vs. Northeast noticed in this work is not present in prior analyses for shorter time periods (1990-2010 in *Cooper et al*. 2012 and 1998-2013 in *Simon et al*. 2015). We find 72% of the Southeast sites experiencing significant median $O_3$ decreases in spring over 1988-2014, while *Cooper et al*. found only 8%. Sites with significant 95th percentile springtime $O_3$ decreases in the EUS are also more common in our study (85% compared to 43% in *Cooper et al*.). For the 5th percentile, 45% of the Northeast sites in our analysis have significant spring $O_3$ increases, whereas only 15% in *Cooper et al*.**

**"Compared to the 1990-2010 trends reported in *Cooper et al*., the EUS summer $O_3$ decreases reported here with additional data to 2014 are 33% stronger."**

*P13 Line 23: How does the GHCNDEX relate to the meteorology used to drive the model? Why not calculate the change in max temperature etc. using the same met fields that drive the model?*

**RE: Note that the model is nudged to NCEP U and V but not temperature (as described in Section 2.1). The simulated change in Tmax is shown in Fig.12b. GHCNDEX represents observations, with input data from the Global Historical Climatology Network (GHCN) Daily station data.**

*P20 Line 16: This is a significant bias, and should be discussed earlier in the paper.*

**RE: We have mentioned the mean model biases in Section 4.2 when referring to Figs. S4 and S5. The high model biases in EUS surface ozone is well known and common across the models. The discussion fits better in Section 6, which focuses on EUS.**

*Fig. 8 caption: What does "colorbar saturates at -0.8" mean?*

**RE: Changed to "The color scale saturates at ± 0.8". It means that there are values outside of the -0.8 to +0.8 range.**

[revised manuscript text omitted]

We first evaluate the annual trends of $O_3$ over 900-600 hPa at Hanoi (21°N, 106°E) and Hong Kong (22°N, 114°E) ozonesonde sites in Southeast Asia (**Fig.5b-5c**), where our model indicates the greatest $O_3$ increases (**Fig.4e**). The ozonesonde frequency is 4 profiles per month at Hong Kong and only 1-2 profiles per month at Hanoi. To determine the representativeness of $O_3$ trends derived from these sparse measurements, we compare observations and model results co-sampled on sonde launch days with the 'true average' determined from $O_3$ fields archived every three hours from the model, as in our prior work for WUS sites (Lin et al., 2015a; Lin et al., 2015b). The trends are generally consistent across the sonde data, model co-sampled and 'true average' results for Hong Kong, with a total increase of ~15% from 2005 to 2014. However, sampling deficiencies may influence the trends derived from ozonesondes at Hanoi recently reported by Zhang Y. et al. (2016). 
[revised manuscript text omitted]

| | | |
|---|---|---|
| **Page 1: [1] Deleted** | **Meiyun Lin** | **12/20/16 10:59:00 AM** |

| | | |
|---|---|---|
| **Page 15: [2] Deleted** | **Meiyun Lin** | **1/20/17 4:20:00 PM** |

from

| | | |
|---|---|---|
| **Page 15: [2] Deleted** | **Meiyun Lin** | **1/20/17 4:20:00 PM** |

from

| | | |
|---|---|---|
| **Page 15: [2] Deleted** | **Meiyun Lin** | **1/20/17 4:20:00 PM** |

from

| | | |
|---|---|---|
| **Page 15: [2] Deleted** | **Meiyun Lin** | **1/20/17 4:20:00 PM** |

from

| | | |
|---|---|---|
| **Page 15: [2] Deleted** | **Meiyun Lin** | **1/20/17 4:20:00 PM** |

from

| | | |
|---|---|---|
| **Page 15: [2] Deleted** | **Meiyun Lin** | **1/20/17 4:20:00 PM** |

from

| | | |
|---|---|---|
| **Page 15: [3] Deleted** | **Meiyun Lin** | **1/20/17 4:22:00 PM** |

total

| | | |
|---|---|---|
| **Page 15: [3] Deleted** | **Meiyun Lin** | **1/20/17 4:22:00 PM** |

total

| | | |
|---|---|---|
| **Page 15: [3] Deleted** | **Meiyun Lin** | **1/20/17 4:22:00 PM** |

total

| | | |
|---|---|---|
| **Page 15: [3] Deleted** | **Meiyun Lin** | **1/20/17 4:22:00 PM** |

total

| | | |
|---|---|---|
| **Page 15: [3] Deleted** | **Meiyun Lin** | **1/20/17 4:22:00 PM** |

total

| | | |
|---|---|---|
| **Page 15: [4] Deleted** | **Meiyun Lin** | **1/20/17 4:24:00 PM** |

| | | |
|---|---|---|
| **Page 15: [4] Deleted** | **Meiyun Lin** | **1/20/17 4:24:00 PM** |

| | | |
|---|---|---|
| **Page 15: [4] Deleted** | **Meiyun Lin** | **1/20/17 4:24:00 PM** |

| | | |
|---|---|---|
| **Page 15: [4] Deleted** | **Meiyun Lin** | **1/20/17 4:24:00 PM** |

| | | |
|---|---|---|
| **Page 15: [4] Deleted** | **Meiyun Lin** | **1/20/17 4:24:00 PM** |

| | | |
|---|---|---|
| **Page 15: [4] Deleted** | **Meiyun Lin** | **1/20/17 4:24:00 PM** |

| Page 15: [4] Deleted | Meiyun Lin | 1/20/17 4:24:00 PM |
| --- | --- | --- |

| Page 15: [4] Deleted | Meiyun Lin | 1/20/17 4:24:00 PM |
| --- | --- | --- |

| Page 15: [4] Deleted | Meiyun Lin | 1/20/17 4:24:00 PM |
| --- | --- | --- |

| Page 15: [5] Deleted | Meiyun Lin | 12/21/16 5:35:00 PM |
| --- | --- | --- |

zone

| Page 15: [5] Deleted | Meiyun Lin | 12/21/16 5:35:00 PM |
| --- | --- | --- |

zone

| Page 15: [5] Deleted | Meiyun Lin | 12/21/16 5:35:00 PM |
| --- | --- | --- |

zone

| Page 15: [5] Deleted | Meiyun Lin | 12/21/16 5:35:00 PM |
| --- | --- | --- |

zone

| Page 15: [6] Deleted | Meiyun Lin | 1/7/17 6:45:00 PM |
| --- | --- | --- |

year

| Page 15: [6] Deleted | Meiyun Lin | 1/7/17 6:45:00 PM |
| --- | --- | --- |

year

| Page 15: [7] Deleted | Meiyun Lin | 12/12/16 3:11:00 PM |
| --- | --- | --- |

and

| Page 15: [7] Deleted | Meiyun Lin | 12/12/16 3:11:00 PM |
| --- | --- | --- |

and

| Page 15: [7] Deleted | Meiyun Lin | 12/12/16 3:11:00 PM |
| --- | --- | --- |

and

| Page 15: [7] Deleted | Meiyun Lin | 12/12/16 3:11:00 PM |
| --- | --- | --- |

and

| Page 15: [7] Deleted | Meiyun Lin | 12/12/16 3:11:00 PM |
| --- | --- | --- |

and

| Page 20: [8] Deleted | Meiyun Lin | 12/20/16 12:44:00 PM |
| --- | --- | --- |

| Page 21: [9] Deleted | Meiyun Lin | 12/22/16 12:54:00 PM |
| --- | --- | --- |

(Figure 20 about here: 2003-2012 minus 1981-1990)

| Page 22: [10] Deleted | Meiyun Lin | 1/6/17 3:19:00 PM |
| --- | --- | --- |

rising

| Page 22: [10] Deleted | Meiyun Lin | 1/6/17 3:19:00 PM |
|---|---|---|

rising

| Page 22: [10] Deleted | Meiyun Lin | 1/6/17 3:19:00 PM |
|---|---|---|

rising

| Page 22: [10] Deleted | Meiyun Lin | 1/6/17 3:19:00 PM |
|---|---|---|

rising

| Page 22: [10] Deleted | Meiyun Lin | 1/6/17 3:19:00 PM |
|---|---|---|

rising

| Page 22: [10] Deleted | Meiyun Lin | 1/6/17 3:19:00 PM |
|---|---|---|

rising

| Page 22: [10] Deleted | Meiyun Lin | 1/6/17 3:19:00 PM |
|---|---|---|

rising

| Page 22: [10] Deleted | Meiyun Lin | 1/6/17 3:19:00 PM |
|---|---|---|

rising

| Page 22: [10] Deleted | Meiyun Lin | 1/6/17 3:19:00 PM |
|---|---|---|

rising

| Page 22: [10] Deleted | Meiyun Lin | 1/6/17 3:19:00 PM |
|---|---|---|

rising

| Page 22: [10] Deleted | Meiyun Lin | 1/6/17 3:19:00 PM |
|---|---|---|

rising

| Page 22: [10] Deleted | Meiyun Lin | 1/6/17 3:19:00 PM |
|---|---|---|

rising

| Page 22: [11] Deleted | Meiyun Lin | 1/21/17 11:12:00 AM |
|---|---|---|

domestic

| Page 22: [11] Deleted | Meiyun Lin | 1/21/17 11:12:00 AM |
|---|---|---|

domestic

| Page 22: [11] Deleted | Meiyun Lin | 1/21/17 11:12:00 AM |
|---|---|---|

domestic

| Page 22: [11] Deleted | Meiyun Lin | 1/21/17 11:12:00 AM |
|---|---|---|

domestic

| Page 22: [11] Deleted | Meiyun Lin | 1/21/17 11:12:00 AM |
|---|---|---|

domestic

| Page 22: [11] Deleted | Meiyun Lin | 1/21/17 11:12:00 AM |
|---|---|---|

domestic

| Page 22: [11] Deleted | Meiyun Lin | 1/21/17 11:12:00 AM |
|---|---|---|

domestic

| Page 22: [11] Deleted | Meiyun Lin | 1/21/17 11:12:00 AM |
|---|---|---|

domestic

| Page 22: [11] Deleted | Meiyun Lin | 1/21/17 11:12:00 AM |
|---|---|---|

domestic

| Page 22: [12] Formatted | Meiyun Lin | 1/6/17 3:11:00 PM |
|---|---|---|

Subscript

| Page 22: [12] Formatted | Meiyun Lin | 1/6/17 3:11:00 PM |
|---|---|---|

Subscript

| Page 22: [12] Formatted | Meiyun Lin | 1/6/17 3:11:00 PM |
|---|---|---|

Subscript

| Page 22: [12] Formatted | Meiyun Lin | 1/6/17 3:11:00 PM |
|---|---|---|

Subscript

| Page 22: [13] Deleted | Meiyun Lin | 1/6/17 3:46:00 PM |
|---|---|---|

rising

| Page 22: [13] Deleted | Meiyun Lin | 1/6/17 3:46:00 PM |
|---|---|---|

rising

| Page 22: [13] Deleted | Meiyun Lin | 1/6/17 3:46:00 PM |
|---|---|---|

rising

| Page 22: [13] Deleted | Meiyun Lin | 1/6/17 3:46:00 PM |
|---|---|---|

rising

| Page 22: [13] Deleted | Meiyun Lin | 1/6/17 3:46:00 PM |
|---|---|---|

rising

| Page 22: [13] Deleted | Meiyun Lin | 1/6/17 3:46:00 PM |
|---|---|---|

rising

| Page 22: [13] Deleted | Meiyun Lin | 1/6/17 3:46:00 PM |
|---|---|---|

rising

| Page 22: [13] Deleted | Meiyun Lin | 1/6/17 3:46:00 PM |
|---|---|---|

rising

| Page 22: [13] Deleted | Meiyun Lin | 1/6/17 3:46:00 PM |
|---|---|---|

rising

| Page 22: [13] Deleted | Meiyun Lin | 1/6/17 3:46:00 PM |
|---|---|---|

rising

| Page 22: [13] Deleted | Meiyun Lin | 1/6/17 3:46:00 PM |
|---|---|---|

rising

| Page 22: [13] Deleted | Meiyun Lin | 1/6/17 3:46:00 PM |
|---|---|---|

rising

| Page 22: [13] Deleted | Meiyun Lin | 1/6/17 3:46:00 PM |
|---|---|---|

rising

rRegional $NO_x$ controls also alleviated the $O_3$ buildup during the recent heat waves of 2011 and 2012 relative to earlier heat waves (**Figs. 18 and 19**). Despite high mean state biases, the model captures the salient features of observed $O_3$ trends over the EUS, including the largest summertime decreases in the 95[th] percentile, the north-to-south gradient in springtime $O_3$ trends, as well as wintertime increases in the 5[th] and 50[th] percentiles. The model also captures enhancements in monthly mean $O_3$ due to large-scale heat waves.

Font:Not Bold

Font:Not Bold

**Table 1** Summary of forcings and emissions used in AM3 hindcasts and CM3 projections

| Experiment | Time Periods | Meteorology | Radiative forcings | CH$_4$ (chemistry) | Anthropogenic emissions | Fire Emissions |
|---|---|---|---|---|---|---|
| BASE | 1979-2014 | Nudged to NCEP | Historical | Historical | Historical | Historical |
| Background | 1979-2014 | as BASE | Historical | Historical | Zeroed out in N. America; As BASE elsewhere | Historical |
| FIXEMIS | 1979-2014 | as BASE | Historical | 2000 | Constant* | Constant* |
| IAVFIRE | 1979-2014 | as BASE | Historical | 2000 | Constant* | Historical |
| IAVASIA | 1979-2012[+] | as BASE | Historical | 2000 | Varying in Asia as BASE; as in FIXEMIS elsewhere | Constant* |
| IAVCH$_4$ | 1979-2012[+] | as BASE | Historical | Historical | Constant* | Constant* |
| CM3_RCP4.5 | 2005-2050 | Free running | RCP4.5 | RCP4.5 | RCP4.5 | RCP4.5 |
| CM3_RCP8.5 | 2005-2050 | Free running | RCP8.5 | RCP8.5 | RCP8.5 | RCP8.5 |

*Averaged over the whole 1970-2010 period.

+Note that the IAVASIA and IAVCH$_4$ simulations only extend to 2012.

**Table 2.** Summary of linear trends in spring MDA8 $O_3$ for 1988 to 2012 (ppb $yr^{-1}$) observed at seven western U.S. sites and as simulated in the AM3 experiments. 
[revised manuscript text omitted]